# Zoology: Measuring and Improving Recall in Efficient Language Models

**Simran Arora**\*, **Sabri Eyuboglu**,\* **Aman Timalsina, Isys Johnson, Michael Poli,**
**James Zou, Atri Rudra, Christopher Ré**
Department of Computer Science, Stanford University
Stanford University
{simran, eyuboglu, poli, jamesz, chrismre}@cs.stanford.edu
{isysjohn, atri}@buffalo.edu
{atimalsi}@purdue.edu

## ABSTRACT

Attention-free language models that combine *gating* and *convolutions* are growing in popularity due to their efficiency and increasingly competitive performance. To better understand these architectures, we pretrain a suite of 17 attention and *gated-convolution* language models, finding that SoTA gated-convolution architectures still underperform attention by up to 2.1 perplexity points on the Pile. In fine-grained analysis, we find 82% of the gap is explained by each model's ability to recall information that is previously mentioned in-context, *e.g. Hakuna Matata means no worries Hakuna Matata it means no → ??*. On this task, termed *associative recall*, we find that attention outperforms gated-convolutions by a large margin: a 70M parameter attention model outperforms a 1.4 billion parameter gated-convolution model on associative recall. This is surprising because prior work shows gated convolutions can perfectly solve synthetic tests for AR capability. To close the gap between synthetics and real language, we develop a new formalization of the task called multi-query associative recall (MQAR) that better reflects actual language. We perform an empirical and theoretical study of MQAR that elucidates differences in the parameter-efficiency of attention and gated-convolution recall. Informed by our analysis, we evaluate simple convolution-attention hybrids and show that hybrids with input-dependent sparse attention patterns can close 97.4% of the gap to attention, while maintaining sub-quadratic scaling. Code is at: https://github.com/HazyResearch/zoology.

## 1 INTRODUCTION

Two advances – gating and long convolutions – have catalyzed a wave of excitement around *gated-convolution* language models (Fu et al., 2023a; Ma et al., 2022; Wang et al., 2022; Poli et al., 2023a, inter alia.). These architectures combine gating (*i.e.* element-wise multiplication) with *long* convolutional filters (*i.e.* the length of the sequence) to enable interactions between distant tokens (Dauphin et al., 2017; Gu et al., 2021). Recent work suggests these models, which exhibit better asymptotic scaling in input sequence length than attention, can match attention in language modeling quality.

We pretrain and evaluate 17 language models across 4 scales (70M - 1.4Bn) and 5 architectures on the same data and infrastructure setup. Surprisingly, we find that there is still a perplexity gap of up to 2.1 points between state-of-the-art convolution-based architectures and strong Transformer baselines in language modeling on the Pile (Table 1). Through fine-grained analysis, we find a single, simple capability is responsible for much of the gap: recalling information seen in-context. Consider the example below, where some tokens can be predicted by recalling an earlier association:

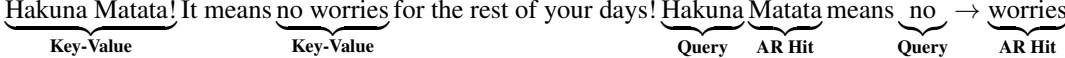

We find that errors on "AR Hits" (*e.g. worries* above) account for 82% of the perplexity gap to attention on average, despite only representing $6.4\%$ of all tokens in the Pile dataset. A 70M parameter Transformer can predict AR Hits better than a 1.4Bn parameter Hyena gated convolution model

---

\*Equal contribution, Random ordering by coin toss.

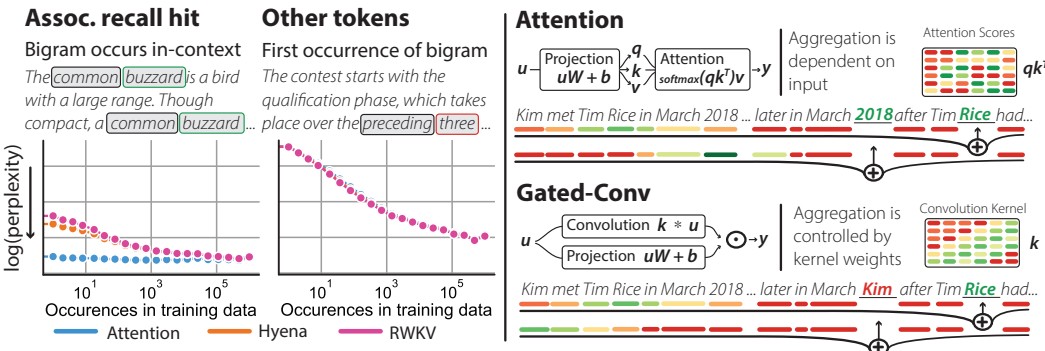

Figure 1: **The associative recall gap.** We stratify Pile validation data for models from each architecture class by whether or not the predicted token is a previously seen bigram in the example context. We plot validation perplexity versus the bigram's frequency in the training data. We can clearly see that the gap is localized to examples where bigrams occur, and are seen rarely during in training.

($20\times$ larger) (Table 1, Table 6). The AR gap persists at the 7Bn parameter scale when comparing RWKV and Llama-2 (Appendix G).[1]

This task, *associative recall* (AR), has a long history in machine learning (Graves et al., 2014; Ba et al., 2016, inter alia.) (Appendix A.1). Prior work argues that a model's ability to perform AR is predictive of in-context learning quality (Elhage et al., 2021; Olsson et al., 2022). As a result, AR has been adopted as a tool in designing new architectures and prior work has shown that gated convolution architectures match attention on synthetic AR tasks used as proxies for real language modeling (Fu et al., 2023a; Poli et al., 2023a; Lutati et al., 2023). For this reason, the downstream AR perplexity gaps are surprising.

Through measuring recall on real data, we learn the key disparity is that prior synthetic formulations assume there is *one query* per input, at a *fixed position* in the sequence, where tokens come from a small vocabulary size (*e.g.* $|V| < 50$, less than model dimension). Yet, language modeling often requires performing *multiple recalls* (*e.g.* for both "Hakuna Matata" and "no worries" above, in a single forward pass), at *varying positions*, with tokens from a large vocabulary (larger than model dimension). We thus propose the study of *multi-query* AR (MQAR). Compared to the prior AR formulations, MQAR better captures the persisting quality gaps on synthetic and real world data (Section 4). However, it is not clear why MQAR elucidates the gap.

**We formalize the the MQAR gap across the architecture classes.** Gated-convolutions process variable sequences using *fixed* filters defined by the model weights rather than as functions of the input data (See Figure 1). We find it is inefficient for gated-convolutions to perform the variable distance token-to-token interactions (e.g. at a distance of 10 tokens for *Hakuna Matata* and 9 for *no worries*) in a parameter and FLOPs efficient way compared to attention. Attention achieves input-dependence since it computes all token-to-token interactions when determining how to mix information in the sequence. We formally describe the limitation of gated convolutions via theory and experiments in the rest of the paper.

We first introduce a simple operator, BASECONV, that can provably simulate the class of architectures built from gating and convolution primitives. This includes architectures such as H3, Hyena, RWKV, and RetNet. We show with theory and experiments that the model dimension for BASECONV (and thus the aforementioned architectures) to solve MQAR grows with the input sequence length (Theorem 4.4) while attention can solve MQAR with model dimension independent of sequence length (Proposition 4.3, Section 4.3).[2] In practice, gated-convolutions appear to encode approximate solutions to AR that support only a subset of token-interaction distances (which affects the ability to identify matching keys given MQAR queries). While theoretically analyzing deep learning architectures is challenging (Hahn, 2020; Merrill et al., 2022; Keles et al., 2023), the fact that gating and convolutions are polynomial operations facilitates our precise analysis.

---

[1]We measure AR on real data using a simple heuristic: $n$-gram tokens that are repeated in context.

[2]Note that the *runtime* of Attention is quadratic, but gated convolutions is near linear.

| Model | Param (M) | TFLOPs | Overall | AR Hits | Other Tokens | % of gap due to AR Hits |
|---|---|---|---|---|---|---|
| | | | | **Slices** | | |
| Attention | 125 | 2.46 | **11.01** (2.40) | **2.16 (0.77)** | 12.45 (2.52) | — |
| Long Conv | 128 | 1.74 | 16.98 (2.83) | 25.62 (3.24) | 16.46 (2.80) | 40.1% |
| H3 | 168 | 2.55 | 12.06 (2.49) | 6.75 (1.91) | 12.60 (2.53) | 88.4% |
| Hyena | 158 | 2.41 | 11.60 (2.45) | 5.00 (1.61) | 12.28 (2.51) | 100.0% |
| RWKV | 169 | 2.08 | 11.64 (2.45) | 5.70 (1.74) | 12.29 (2.51) | 100.0% |
| Attention | 360 | 6.23 | **9.44** (2.25) | **1.98 (0.69)** | 10.62 (2.36) | — |
| Long Conv | 360 | 4.08 | 13.13 (2.57) | 13.27 (2.59) | 13.12 (2.57) | 40.5% |
| H3 | 357 | 4.85 | 10.38 (2.34) | 4.81 (1.57) | 11.00 (2.40) | 65.8% |
| Hyena | 358 | 5.03 | 10.07 (2.31) | 3.83 (1.34) | 10.75 (2.38) | 98.2% |
| RWKV | 351 | 4.31 | 9.79 (2.28) | 3.82 (1.34) | 10.51 (2.35) | 100.0% |

Table 1: **Language modeling validation perplexity on the Pile.** After pretraining on 10B tokens of Pile data, we report log perplexity with negative log-likelihood in parentheses. We report overall scores, and for the AR vs. non-AR token slices defined in Section 3. FLOPs are computed for inputs of 2048 tokens based on the equations in Appendix C. A table with additional results is in Table 4.

**We show that input-dependent sequence mixing is important to solve MQAR efficiently.** The scaling for gated convolutions with input-independent convolutions is undesirable so we next ask which architectural choices close the gap. We show that data-dependent sequence mixing helps an architecture solve MQAR efficiently (Theorem 4.5). The model needs to adapt the sequence mixing weights based on the token-interaction distances required for each new example.

Several architectural modifications could satisfy the input-dependence property. Based on our analysis, we evaluate *minimal modifications* that only add input-dependent operations on exact-match repeated bigram tokens (our heuristic for measuring tokens that require recall). Note that language models often need to perform recall between fuzzier substrings (e.g. synonymous bigrams) or higher-dimensional concepts. However, we show that simply inserting input-dependent operator — e.g., a convolution filter that shifts the sequence based on the bigram positions or sparse attention placed only on repeated bigram positions — to the BASECONV architecture at $< 10\%$ of layers suffices to outperform the Transformer baseline on Pile language modeling (Section 5) and succeed on MQAR synthetic tasks (Section 4.3). Moreover, this closes $> 80\%$ of the MQAR perplexity gap on the Pile validation data. Finally, we prototype solutions that *learn* the positions at which to use input-dependent operations, validating that they also close the gap to attention on the Pile.

In this work, we analyze the increasingly-popular convolution-based architectures and identify fundamental limitations. We hope our analysis informs the design of future efficient architectures.

## 2 BACKGROUND AND PRELIMINARIES

In this section, we describe the setting, introduce notation, and discuss important related work. See Appendix A for a broader discussion of related work.

**Language modeling.** We study auto-regressive language models trained on the task of next token prediction (Gao et al., 2020). Given a sequence of $N$ tokens $\boldsymbol{x} = \{x_0, ..., x_{N-1}\}$ drawn from a vocabulary $C$, the model outputs a probability distribution over $C$ for each of $x_i$ given the preceding tokens $\mathrm{P}(x_i|x_0, ..., x_{i-1})$. The language models in this work share the same high-level architecture. First, each token $x_i$ in the input is embedded in $d$-dimensional space yielding a matrix $\boldsymbol{u} \in \mathbb{R}^{N \times d}$. Next, $\boldsymbol{u}$ is passed through a stack of $L$ layers, with layer $\ell$ outputting $\boldsymbol{u}^\ell \in \mathbb{R}^{N \times d}$. Finally, the embeddings $\boldsymbol{u}^L$ output by the last layer are mapped back to logits over $C$ with a linear projection. Each layer transforms $\boldsymbol{u}$ with a *sequence mixer* (*e.g.* attention) followed by a *state mixer* (*e.g.* MLP). Unless specified, our models adhere to the implementation details of the LLaMA architecture (except the sequence mixer, which we vary throughout) (Touvron et al., 2023).

**Sequence mixers.** Our work evaluates how the choice of sequence mixer affects the quality and behavior of language models. Most sequence mixers aggregate the token embeddings in a sequence via a weighted sum. For example, $\boldsymbol{y}[i, :] = \sum_{j=0}^{N-1} \omega(i, j) \boldsymbol{u}[j, :])$, where $\omega$ is a function outputting scalar weights. We study the differences between two classes of sequence mixers, discussed next.

*Attention.* (Vaswani et al., 2017) We review attention, the *de facto* language model sequence mixer. An attention layer is parameterized by three learnable projection matrices $\mathbf{Q}, \mathbf{K}, \mathbf{V} \in \mathbb{R}^{N \times d}$. To compute the output $\boldsymbol{y}$ given inputs $\boldsymbol{u}$, attention applies the projections to the input: $\boldsymbol{q} = \mathbf{Q}\mathbf{u}$, $\boldsymbol{k} = \mathbf{K}\mathbf{u}$, $\boldsymbol{v} = \mathbf{V}\mathbf{u}$. The projected embeddings are aggregated according to: $\boldsymbol{y} = \text{softmax}(\frac{1}{\sqrt{d}}\boldsymbol{q}\boldsymbol{k}^\top)\boldsymbol{v}$ (shown in Fig. 1) in $\mathcal{O}(N^2 d)$ time, which is expensive for long sequences (large $N$).

*Gated-convolutions.* A more efficient alternative to attention is the convolution, which is defined as $\boldsymbol{y}[i,:] = \sum_{j=0}^{N-1} \mathbf{k}[j,:] \odot \boldsymbol{u}[i-j,:]$ where the kernel $\mathbf{k} \in \mathbb{R}^{N \times d}$ is a learnable weight matrix. Convolutions can be computed in time $O(Nd \log N)$ using the Fast Fourier Transform (FFT) and the convolution theorem: $\boldsymbol{y} = \mathbf{u} * \mathbf{k} = \text{FFT}^{-1}(\text{FFT}(\mathbf{u}) \odot \text{FFT}(\mathbf{k}))$ (Cooley and Tukey, 1965). While purely convolutional architectures match or outperform attention in certain domains (*e.g.* vision (Tay et al., 2022; Gu et al., 2021), audio (Goel et al., 2022), and time-series (Zhang et al., 2023)), they trail by a large margin on language (Fu et al., 2023a).

Recent work has closed much of this gap by combining convolutions with *gating* (*i.e.* elementwise multiplication of the input with a transformed version of itself Dauphin et al. (2017)). Some state-of-the-art sub-quadratic language models claim attention-level quality (Fu et al., 2023a; Wang et al., 2022; Poli et al., 2023a; Peng et al., 2023; Zhai et al., 2021; Fu et al., 2023b, inter alia). Though they may appear different on the surface, these sub-quadratic architectures can all be expressed in terms of convolutions and gating (see Appendix H.2 for detailed descriptions). This work analyzes the differences between attention and the broad class of *gated convolution* mixers.

## 3 IDENTIFYING THE ASSOCIATIVE RECALL PROBLEM

In this section, we measure the perplexity gap between gated convolutions and attention and show that single skill termed *associative recall* accounts for 82% of the gap on average. This is surprising because prior work shows gated convolutions solve a synthetic version of associative recall perfectly. Informed by our analysis, we define a new synthetic formulation that better reflects real data. This task facilitates our analysis of why the gap occurs (Section 4) and how to fix it (Section 5).

### 3.1 FINE-GRAINED ANALYSIS OF DOWNSTREAM QUALITY

**Perplexity Gap** We pretrain a suite of large language models with different sequence mixers across 3 scales (70M-360M) for 10B tokens on the standard Pile language modeling setting using the EleutherAI GPT-NeoX training infrastructure (Gao et al., 2020; Andonian et al., 2023). In the main paper, we compare attention to three state-of-the-art gated convolution sequence mixers: H3, Hyena, and RWKV (Fu et al., 2023a; Poli et al., 2023a; Peng et al., 2023).[3] We further include a pure long-convolution model to underscore the importance of gating. In Appendix F, we include results on additional sequence mixers (Hasani et al., 2022; Sun et al., 2023). We use a strong Transformer attention baseline with rotary embeddings and SwiGLU MLPs following the Llama architecture (Touvron et al., 2023). We also use this strong training recipe when training the attention-free sequence mixers. For experimental details and hyperparameters, see Appendix C.

Across scales, we find that attention outperforms the gated convolutions. The minimum quality gaps are $+2.14$, $+0.59$, $+0.35$ PPL at 70M, 160M, and 360M parameter scales, respectively. We report overall test perplexity in Table 1. Though these gaps are relatively small on average, models may still perform very differently on different subsets of data (Eyuboglu et al., 2022).

**Associative Recall Perplexity** To better understand the differences between attention and gated convolutions, we perform a fine-grained analysis of next token predictions and observe that convolution-based models struggle to recall associations previously seen in context. For example, in Fig. 1, the model must recall the association between "*Tim*" and the last name "*Rice*". Following a long line of prior work, we call this skill *associative recall* (AR) (Willshaw et al., 1969; Hopfield, 1982) (see Appendix A for an extended discussion of AR's history in machine learning). In Appendix D.1.1, we provide annotated Pile examples to demonstrate the phenomenon qualitatively.

*Quantifying AR performance.* It is challenging to derive a quantitative measure of associative recall performance on the Pile because we don't know which next token predictions in raw text require recall. We propose a simple heuristic to identify these tokens, which we refer to as *AR Hits*, that enables us to scale our analysis to over 10 million tokens of Pile validation data. An AR Hit is

---

[3]RWKV is commonly referred to as an RNN. We show it can be viewed as a convolution (Appendix H.2.2).

the last token of an $n$-gram repeated in context (*e.g.* the second occurence of *"Rice"* in Fig. 1). However, some common $n$-grams (*e.g. "of the"*) could have been memorized during training, so we factor in the frequency with which an $n$-grams appeared in the training data. We stratify Pile tokens into two slices based on this heuristic and report perplexity on each in Table 1:

1. **AR Hits**: (6.4% of tokens) Tokens in the final position of a bigram (a pair of consecutive tokens) which previously appeared in context, but $\leq 1250\times$ during training.

2. **Other tokens:** (93.6% of tokens) Tokens in the final position of a bigram which did not previously appear in context or it appeared $> 1,250$ times during training.

In Figure 1, we visualize these slices by plotting log-perplexity against the frequency with which bigrams appear during training. Strikingly, the gap between attention and gated convolutions is the largest on AR hits with the fewest occurrences. On the other tokens, there is no gap.

In Table 1, we also compute the percentage of the difference in perplexity between attention and each model that is due to AR tokens: $\frac{\Delta \log(\phi_{\text{AR}}) \cdot |T_{\text{AR}}|}{\Delta \log(\phi) \cdot |T|}$, where $\phi$ is the perplexity and $T$ is the set of tokens in the test set. This quantity can also be interpreted as the fraction of the overall gap that would close if a model matched attention on the AR slice. We find that the AR slice accounts for 82% of the average quality gap between the gated convolutions and attention.

To evaluate the AR capacity of larger models, we train two 1.4 billion parameter attention and Hyena models for 50 billion tokens on the Pile and repeat this analysis (see Table 6). Strikingly, a 70 million parameter attention model is a full perplexity point better in the AR slice than this 1.4B Hyena model that is $20\times$ its size (2.41 vs. 3.43 ppl.). In Appendix G, we also evaluate open-source RWKV and attention models trained up to 7 billion parameters. We show that RWKV's performance degrades sharply relative to attention as we increase the number of recall-queries in an example Appendix G.

### 3.2 FORMALIZING THE PROBLEM: MULTI-QUERY ASSOCIATIVE RECALL

This gap in associative recall perplexity is very surprising because prior work shows gated-convolutions can perfectly solve a formalized version of the task (Fu et al., 2023a; Poli et al., 2023a; Olsson et al., 2022). In this synthetic task, the input $x$ contains a sequence of bigrams representing *key-value* pairs from a random dictionary followed by a *single* query token. For example, the correct output for the input below would be 3:

$$\underbrace{A\ 4\ B\ 3\ C\ 6\ E\ 2\ F\ 1\ C\ 6\ G\ 8}_{\textbf{Key-Value}} \rightarrow \underbrace{B\ ?}_{\textbf{Query}}$$

Gated convolutions (*e.g.* H3, Hyena, RWKV) can solve this task perfectly for most sequence lengths.

These conclusions are inconsistent with our findings on the Pile, as described above, so we ask how this formulation of AR differs from the way AR manifests in real language. We identify a major difference. In real world inputs, the language model often needs to perform multiple associative recalls in a single forward pass, at varying positions in the sequence (*e.g. "Tim Rice"* and *"March 2018"* in Fig. 1. We refer to this as Multi-Query AR (MQAR), and the formal definition follows: [4]

**Definition 3.1** (Multi-Query-AR (MQAR)). We are given an input sequence $x = \{x_0, \ldots, x_{N-1}\}$ where each $x_i \in C$ is a token drawn from a vocabulary of size $c = |C|$. The task is to check, for every query $1 \leq i < N$, whether there exists a $0 \leq j < i$ such that $u_i \equiv u_j$. If so, output $u_{j+1}$.

For example, the correct output for input below would be 4, 6, 1, 2, 3:

$$\underbrace{A\ 4\ B\ 3\ C\ 6\ F\ 1\ E\ 2}_{\textbf{Key-Value}} \rightarrow A\ ?\ C\ ?\ \underbrace{F\ ?}_{\textbf{Query}}\ E\ ?\ B\ ?$$

In Section 4, we use MQAR to explain the quality gap between gated convolutions and attention.

## 4 EXPLAINING THE ASSOCIATIVE RECALL PROBLEM

In this section, we provide an explanation for the gap in associative recall performance by analyzing the formal MQAR task theoretically and empirically. In Section 4.1, we define a simple gated-convolution architecture, called BASECONV, which we show can simulate a broad class of

---

[4]In Appendix H.7.1 we define a formal, general form of this definition, used in the theoretical analysis.

architectures built from gating and convolutions. This allows us to make general statements that apply to popular gated-convolution architectures like Hyena, RWKV, or H3. In Section 4.2, we show that there exist theoretical solutions to MQAR that could in principle be learned by BASECONV, and we analyze their complexity in terms of model width and depth. In Section 4.3, we use experiments on synthetic data to show that solving MQAR with BASECONV (and other gated-convolution architectures) requires model dimension to scale linearly with the sequence length. In contrast, attention solves MQAR consistently in our experiments with model dimension scaling independently of sequence length. These empirical scaling laws provide a potential explanation for the AR gap and, alongside our theoretical analysis, point to the potential solutions discussed in Section 5.

## 4.1 BASECONV: A MINIMAL GATED CONVOLUTION OPERATOR

In this section, we define our minimal gated-convolution architecture, called BASECONV. Given a function, we would like to know the most efficient model (*e.g.* parameters, FLOPs) that can represent the solution. In this work, we show we can precisely reason about this question for representing *polynomial functions* with gated convolutions as gating and convolutions are both polynomial operations. The standard model defining computational complexity for polynomials is by the size of the smallest arithmetic circuit that can compute the polynomial. We define the BASECONV gated convolution operator which is exciting because (1) it is universal in that it that can simulate any arithmetic circuit $\mathcal{C}$ (with only a poly-log blowup in the corresponding parameters) and (2) it is simple to implement efficiently (19 lines of pure PyTorch including imports, see Appendix B).

**Definition 4.1** (BASECONV Operator). Given an input $\boldsymbol{u} \in \mathbb{R}^{N \times d}$, the BASECONV operator for layer $\ell$ is defined as:

$$\boldsymbol{y} := \underbrace{\left(\boldsymbol{u} \cdot \boldsymbol{W}^\ell + \boldsymbol{b}_1^\ell\right)}_{\text{Linear Projection}} \odot \underbrace{\left(\boldsymbol{h}^\ell * \boldsymbol{u} + \boldsymbol{b}_2^\ell\right)}_{\text{Convolution}} \tag{1}$$

where the layer is parameterized by learnable filters $\boldsymbol{h} \in \mathbb{R}^{N \times d}$, a linear projection $\boldsymbol{W}^\ell \in \mathbb{R}^{d \times d}$, and 'bias' matrices $\boldsymbol{b}_1, \boldsymbol{b}_2 \in \mathbb{R}^{N \times d}$. The $\odot$ is component-wise product and convolution of two matrices is computed as convolution of the corresponding columns.

In our experiments, each BASECONV layer uses $\tilde{\mathcal{O}}(Nd + d^2)$ parameters[5] and can be computed in $\tilde{\mathcal{O}}(Nd^2)$ operations. For our theoretical results, we can assume the weight matrix $\boldsymbol{W}^\ell$ is restricted to a class of matrices that support near-linear time matrix multiplication (*e.g.* Kaleidoscope matrices, see Definition H.3). Under this assumption, BASECONV uses $\tilde{\mathcal{O}}(Nd)$ parameters and $\tilde{\mathcal{O}}(Nd)$ FLOPs (Proposition H.6). We now state the equivalency result between arithmetic circuits and BASECONV a "canonical" representation of arithmetic circuits (Theorem H.21 in Appendix H.5):

**Theorem 4.2** (Equivalency to Arithmetic Circuits). *For an arithmetic circuit $\mathcal{C}$ of size $s$ and depth $\Delta$ that takes $\boldsymbol{u} \in \mathbb{R}^{N \times d}$ as input, there exists an equivalent BASECONV operator that uses $\tilde{\mathcal{O}}(s\Delta)$ parameters and $\tilde{\mathcal{O}}(\Delta)$ layers.*[6]

In other words, any gated convolution model with small number of layers can be simulated by BASECONV with only a (poly)logarithmic blowup in parameters and layers. We note that arithmetic circuits are a very well studied computation model in computational complexity Bürgisser et al. (1996). Many well-known efficient algorithms on matrices (e.g. the FFT or the current best known matrix-matrix multiplication algorithm) in fact give small arithmetic circuits. However, arithmetic circuits are inherently discrete objects – we cannot learn them via gradient descent. Theorem 4.2 shows that (up to poly-log loss in parameters), we can *instead* learn over BASECONV models. This result generalizes a similar result from Dao et al. (2020) for the special class of linear functions: we generalize the earlier result to the class of *all* polynomials.

For specific gated convolution layers, we can get rid of the poly-logarithmic factor blowup–we observe in the appendix that BASECONV and Hyena models can simulate each other with only a small constant blowup in parameters (Proposition H.12 in Appendix H.5).

---

[5]We use $\tilde{\mathcal{O}}(\cdot)$ to hide poly-log factors.

[6]The formal statement in the Appendix has a sharper version of this result in terms of the circuit 'width'.

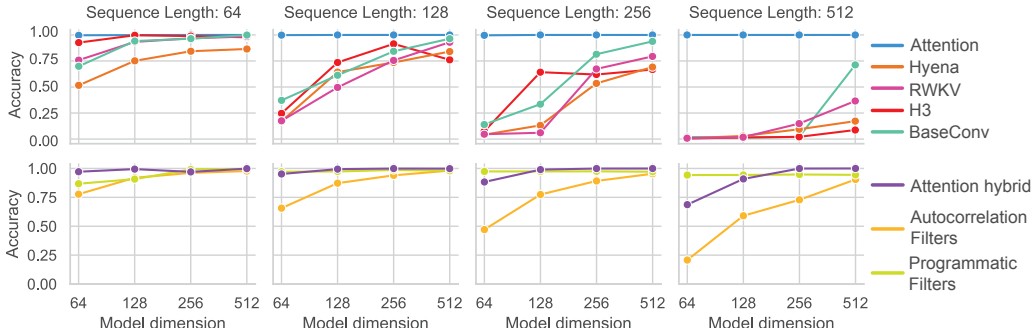

Figure 2: The x-axis is the model dimension and the y-axis is accuracy on MQAR. **(Top, Claim 1) Gated convolutions and attention.** We observe the gated convolutions require larger dimensionality than attention to solve the task. **(Bottom, Claim 2) Input-dependent filters.** Models with input-dependent aggregation achieve improved scaling over the gated-convolutions.

## 4.2 THEORETICAL ANALYSIS OF GATED CONVOLUTION CAPACITY AND RECALL

In this section, we provide theoretical MQAR solutions that could in principle be learned by each architecture and analyze their complexity in terms of model width and depth. First, we note that attention solves MQAR with parameters independent of sequence length (Proposition H.27).

**Proposition 4.3** (Attention). *Given an input $u \in \{0, 1\}^{N \times 3c}$, Attention (even without using softmax) solves MQAR for $u$ using $\mathcal{O}(c^2)$ parameters, $\mathcal{O}(Nc^2 + N^2c)$ time complexity and $\mathcal{O}(1)$ layers.*

It is natural to wonder then if *all* pairwise comparisons among tokens are necessary to solve MQAR. Indeed, in the RAM setting, a sequential algorithm can simply utilize $N$ logarithmic insertion and membership queries to solve MQAR in subquadratic time. Unfortunately, any model attempting to emulate this would require $\Omega(N)$ layers. Instead, we observe that we can parallelize this algorithm using dyadic intervals and achieve a depth of $\tilde{\mathcal{O}}(1)$ (Proposition H.30). We then convert this algorithm into an arithmetic circuit and apply Theorem 4.2 to derive an equivalent BASECONV model. This allows us to prove new upper bounds for BASECONV models applied to MQAR, which improves upon the quadratic time complexity of attention to near-linear runtime at the cost of using poly-log layers (Theorem H.37 in Appendix H.7).

**Theorem 4.4** (Input-Independent Filters[7]). *Given an input $u \in \{0, 1\}^{N \times \mathcal{O}(\log c)}$ to MQAR (where we assume that distinct tokens are embedded into distinct vectors in $\{0, 1\}^{\mathcal{O}(\log c)}$), there exists a BASECONV operator that solves MQAR for $u$ using $\tilde{\mathcal{O}}(N \log c)$ parameters as well as time complexity and $\tilde{\mathcal{O}}(1)$ layers.*

Nevertheless, the poly-logarithmic number of layers in the above result is undesirable in practice. But, we show that using *input-dependent* convolution filters, one can get constant many layers (for a sub-class of inputs). Towards that end, we define the interaction distance between a query $q_i$ and the matching key $k_j$ as $i - j$. This then allows us to present the corresponding upper bound for data-dependent mixing (Theorem H.38 in Appendix H.8).

**Theorem 4.5** (Input-Dependent Filters). *Given an input $u \in \{0, 1\}^{N \times c}$ to MQAR (where we assume that the tokens are embedded as one-hot encoding in $\{0, 1\}^c$ and there exists at most $t$ distinct interaction distances),[8] there exists a BASECONV operator that uses input-dependent kernels to solve the above case of MQAR using $\mathcal{O}(t \cdot Nc)$ parameters and $\mathcal{O}(1)$ layers.*

## 4.3 EMPIRICAL ANALYSIS OF GATED CONVOLUTION CAPACITY AND ASSOCIATIVE RECALL

In this section, we measure empirically how model dimension must scale in order for different sequence mixers to solve MQAR.

**Setup** We train and evaluate models on a synthetic MQAR with vocabulary size $8,192$, across model dimension and sequence length ($64 - 512$). Appendix E provides further details on the con-

---

[7]Meaning the filter is defined as a function of the model parameters, independent of the input.

[8]Note that the interaction distances can be arbitrary: there is just a bounded number of distinct distances.

struction of this synthetic task. Following Olsson et al. (2022), we train two layer models with a Transformer backbone that interleaves sequence mixing and state mixing (MLPs). For each architecture, we sweep four learning rates from $\log(-4)$ to $\log(-2)\}$ and report maximum test accuracy.

Our results, which are summarized in Figure 2, support two main claims:

**Claim 1 (Gated-convolutions and attention).** *Gated-convolution models with two layers require model dimension to scale at least linearly in sequence length in order to solve associative recall, while attention models can solve it with near-constant dimensionality.* We compare attention and BASECONV as well as three popular instantiations of gated-convolution architectures: RWKV, H3, and Hyena (Peng et al., 2023; Fu et al., 2023c; Poli et al., 2023a). In the top row of Fig. 2, attention solves MQAR perfectly at all sequence lengths using a constant model dimension of 64. In contrast, MQAR does not achieve accuracy $> 0.9$ unless $d \geq N$.

**Claim 2 (Input-dependent filters).** *Using input-dependent filters in gated-convolution models can close some of the gap to attention.* In Theorem 4.5, we show that BASECONV with input-dependent filters could solve MQAR with improved scaling. In this solution, we construct a filter that spikes at position $j$ if matching keys are separated by $j$ tokens. We evaluate two approaches for constructing this filter: (1) programatically (*i.e.* hard-coded comparisons between token ids) or (2) with autocorrelation, which could learn to perform fuzzy-matches (see Appendix H.8). In the bottom row of Fig. 2, we see that BASECONV with programmatic input-dependent filters achieves near-constant scaling in model dimension and that BASECONV with autocorrelation input-dependent filters achieves improved scaling over BASECONV with input-independent filters.

These input-dependent filters cannot easily be made to satisfy causality and using an $O(N \log N)$ filter per gap could be expensive if each gap applies only to a small number of bigrams. A simpler and perhaps more efficient way to solve the problem would be to introduce a small amount of attention to an otherwise BASECONV model (Fu et al., 2023a). As the first natural baseline, we evaluate an attention hybrid on synthetic MQAR and show that it achieves improved scaling in Figure 2. Next, we put these empirical and theoretical insights into practice on Pile language modeling.

## 5 CLOSING THE ASSOCIATIVE RECALL GAP

In this section, we evaluate hybrid BASECONV-Attention models that leverage different sparsity patterns. We show that hybrids with input-dependent sparsity patterns can close most of the gap to attention, while maintaining sub-quadratic scaling. We also show that BASECONV hybrids can outperform attention-only models by up to a full perplexity point, all while being dramatically simpler to implement and analyze than prior hybrids (Fu et al., 2023a) (see implementation in Appendix B. We describe the architectures in Section 5 and results on Pile language modeling in Section 5.

**Sparse BASECONV-Attention Hybrids** We evaluate hybrids composed primarily of BASECONV layers and three attention layers (6.3% of layers at 354M parameters and 10% at 168M parameters). We augment the attention layers with operators that *selectively* apply attention to some tokens based on a selection function $f : \mathbb{R}^{N \times d} \to \{0, 1\}^N$. They take as input $\boldsymbol{u} \in \mathbb{R}^{N \times d}$ and output $\boldsymbol{y} \in \mathbb{R}^{N \times d}$:

$$\boldsymbol{y}[i, :] = \text{softmax}(\frac{1}{\sqrt{d}}\boldsymbol{q}[i, :]\boldsymbol{k}^{\top})\boldsymbol{v} \cdot f(\boldsymbol{u})[i] \tag{2}$$

where $\boldsymbol{q}, \boldsymbol{k}, \boldsymbol{v}$ are query, key, and value projections, as in attention. We evaluate four choices for $f$:

(1) *Full attention.* First, we evaluate the performance of full attention hybrids. These hybrids correspond to fixing $f(\boldsymbol{u})[i] = 1$ for all $i$ in Eq. (2). Prior work has shown full attention hybrids to be effective, but they do not explain why supplementing gated convolutions with attention is necessary (Fu et al., 2023a). Based on our findings in Section 3, we hypothesize that sparse attention applied only to associative recall hits may suffice.

(2) *Random selection.* As a control, we evaluate sparse attention randomly applied to tokens. This corresponds to a stochastic selection function where $f(\boldsymbol{u})[i]$ is drawn from a Bernoulli. Many prior works have employed sparse attention patterns like this one, which are independent of the input (Zaheer et al., 2020; Child et al., 2019a; Beltagy et al., 2020, inter alia.).

(3) *Programmatic selection.* Next, to evaluate our hypothesis that attention is needed for associative recall, we prototype a programmatic selection function that selects only those tokens that might be

| Model | Param (M) | TFLOPs | Overall | Slices | |
| --- | --- | --- | --- | --- | --- |
| | | | | AR Hits | Other Tokens |
| Attention | 125 | 2.46 | 11.01 (2.40) | 2.16 (0.77) | 12.45 (2.52) |
| BASECONV | 168 | 2.46 | 12.90 (2.56) | 8.68 (2.16) | 13.29 (2.59) |
| + Random selection | 162 | 2.44 | 12.13 (2.50) | 5.33 (1.67) | 12.83 (2.55) |
| + Programmatic selection | 162 | 2.44 | 11.06 (2.40) | 2.70 (0.99) | 12.19 (2.50) |
| + Learned selection | 166 | 2.44 | 11.06 (2.40) | 3.20 (1.16) | 12.14 (2.50) |
| + Full attention | 166 | 2.58 | 9.57 (2.26) | 2.01 (0.70) | 10.76 (2.38) |
| Attention | 360 | 6.23 | 9.44 (2.25) | 1.98 (0.69) | 10.62 (2.36) |
| BASECONV | 354 | 4.81 | 11.01 (2.40) | 5.98 (1.79) | 11.52 (2.44) |
| + Random selection | 365 | 5.06 | 12.94 (2.56) | 6.17 (1.82) | 13.62 (2.61) |
| + Programmatic selection | 365 | 5.06 | 9.54 (2.26) | 2.35 (0.86) | 10.50 (2.35) |
| + Learned selection | 351 | 5.06 | 9.59 (2.26) | 2.61 (0.96) | 10.58 (2.36) |
| + Full attention | 351 | 5.10 | 8.59 (2.15) | 1.95 (0.67) | 8.91 (2.19) |

Table 2: **Language model perplexity on slices of the PILE.** We evaluate Hyena and BASECONV with Hybridization and Selective look-up at 160 and 355M parameters.

associative recall hits. Specifically, $f(\boldsymbol{x}[i,:])$ is 1 if the token $x_i$ previously occurred in the sequence. In practice, we compare raw token ids, not token embeddings.

$$f(\boldsymbol{x})[i] = \begin{cases} 1 & \text{if there exists} \quad j < i \quad \text{such that} \quad x_i = x_j \\ 0 & \text{otherwise} \end{cases} \tag{3}$$

(4) *Learned selection.* Finally, we prototype a learned selection function $f(\boldsymbol{u})[i] = \sigma(\boldsymbol{u}[i,:] \cdot \boldsymbol{W})$ parameterized as a simple linear layer with sigmoid activation. We fix a hyperparameter $k$ and select the top-$k$ tokens with the highest score in each batch. This allows us to compute attention in $\mathcal{O}(ndk)$ time. During training, we add small amount of Gaussian noise to the top-$k$ calculation to encourage exploration and use an auxiliary loss to encourage sparse selection: $\ell_f(\mathbf{u}) = \frac{1}{N} \max(0, \sum_{i=1}^{N} f(\boldsymbol{u})[i] - k)$. This approach is most similar to the recently proposed SeqBoat architecture (Ren et al., 2023). We discuss the differences in Appendix A.

**Downstream Evaluations** We evaluate the prototypes on Pile language modeling. We take BASECONV architectures at the 150M and 360M parameter scales and add input-dependent selection to three layers (details in Appendix C.2). Results are in Table 2. We validate that the prototypes close the overall and AR quality gaps to attention when added to the BASECONV backbone.

At 360M parameters, BASECONV with just 3 attention layers can outperform the Transformer, while requiring fewer FLOPs. Our hybrid attention-BASECONV models outperform attention only models by 0.85 perplexity points while enabling an 18% reduction in total FLOPs vs attention. However, this uses full quadratic attention. We next show that sparse attention localized to potential AR tokens, is also sufficient to close the gap, validating our insights on the role of input-dependence for MQAR. At 360M parameters, programmatic selection closes 85% of the gap between pure BASECONV and attention on the AR slice, in contrast to the random selection control. Learned selection closes 72% a of the gap using just $k = 256$ (sub-quadratic) attention positions per example.

## 6 DISCUSSION AND CONCLUSION

We present an extensive analysis of gated convolution architectures in light of their recent popularity. We identify a persisting quality gap between efficient convolution and inefficient attention based architectures, largely due to a single failure mode associative recall. We design a new multi-query associative recall (MQAR) analysis tool, which correlates with downstream AR quality. We theoretically and empirically explain the gap is due to insufficient data-dependent mixing in gated convolutions and we show minimal architectures that close the gap on the Pile.

Our results in analyzing gated convolutions go beyond the conventional wisdom that attention is the "*right*" model. A significant amount of work focuses on improving the efficiency of attention (Dao et al., 2022; Dao, 2023; Katharopoulos et al., 2020a) and theoretically studying the exact power of attention (Hahn, 2020; Merrill et al., 2022; Keles et al., 2023). Attention is often used as the goalpost for what is needed downstream. We hope our contributions highlight the value of MQAR, and more broadly tasks tied to real language modeling, as a proxy to study.

## ACKNOWLEDGMENTS

We thank Tri Dao, Daniel Fu, Neel Guha, Stefano Massaroli, Eric Nguyen, and Michael Zhang for helpful feedback and discussion during this work. We are grateful to Together Computer for making this work possible. We gratefully acknowledge the support of DARPA under Nos. FA86501827865 (SDH) and FA86501827882 (ASED); NIH under No. U54EB020405 (Mobilize), NSF under Nos. CCF1763315 (Beyond Sparsity), CCF1563078 (Volume to Velocity), and 1937301 (RTML); ONR under No. N000141712266 (Unifying Weak Supervision); the Moore Foundation, NXP, Xilinx, LETI-CEA, Intel, IBM, Microsoft, NEC, Toshiba, TSMC, ARM, Hitachi, BASF, Accenture, Ericsson, Qualcomm, Analog Devices, the Okawa Foundation, American Family Insurance, Google Cloud, Microsoft Azure, Swiss Re, Brown Institute for Media Innovation, Department of Defense (DoD) through the National Defense Science and Engineering Graduate Fellowship (NDSEG) Program, Fannie and John Hertz Foundation, National Science Foundation Graduate Research Fellowship Program, Texas Instruments Stanford Graduate Fellowship in Science and Engineering, and members of the Stanford DAWN project: Teradata, Facebook, Google, Ant Financial, NEC, VMWare, and Infosys. The U.S. Government is authorized to reproduce and distribute reprints for Governmental purposes notwithstanding any copyright notation thereon. Any opinions, findings, and conclusions or recommendations expressed in this material are those of the authors and do not necessarily reflect the views, policies, or endorsements, either expressed or implied, of DARPA, NIH, ONR, or the U.S. Government. AR's work is supported by NSF grant# CCF-2247014. IJ's work is supported by an NSF Graduate Fellowship.

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

APPENDIX

The appendix includes the following content:

1. Appendix A provides an extended discussion of related work and concepts.

2. Appendix B provides a code implementation of the BASECONV architecture.

3. Appendix C gives details for the experiments, including model architectures and hyperparameters.

4. Appendix D provides additional analysis of how MQAR appears in real data across a variety of language distributions.

5. Appendix E provides a formal definition of the MQAR problem, synthetic construction procedure, and experimental details.

6. Appendix F provides additional synthetic experiments and analysis.

7. Appendix G provides experiments and analysis for how the MQAR gap changes as we scale the gated convolution and attention architectures.s

8. Appendix H gives proofs and additional discussion for the theoretical analysis in our work.

## A    EXTENDED RELATED WORK

We are inspired by and build on prior work in mechanistic interpretability (Appendix A.1), efficient architecture design (Appendix A.2), and input-dependent architectures (Appendix A.3).

### A.1    MECHANISTIC INTERPRETABILITY, SYNTHETIC LANGUAGES, AND RECALL

Work in mechanistic interpretability aims to decompose the capabilities of a neural network into human-understandable algorithms that can be attributed to specific parameters in the model (Olah, 2022; Power et al., 2022; Elhage et al., 2021; Cammarata et al., 2020). Some of these works, use synthetic data to validate mechanistic interpretations of neural networks (Olsson et al., 2022). This relates to a broader line of work using synthetic languages to study language model architectures (Wang and Eisner, 2016; White and Cotterell, 2021; Allen-Zhu and Li, 2023; Ravfogel et al., 2019; Xie et al., 2021). *Mechanistic design* puts mechanistic interpretations to use in designing new architectures and learning algorithms. Several works in architecture research have used synthetic tasks to validate architecture designs (Kitaev et al., 2020).

Our work is focused on one particular synthetic task: *associative recall*. Motivated by psychological models of how humans associate and retrieve information, work in the early days of neural network research focused on developing systems capable of associative recall (Willshaw et al., 1969; Feldman et al., 1981; Hopfield, 1982). For example, Hopfield networks, proposed in 1982, are a recurrent neural network explicitly designed to support associative ("content-addressable") memory (Hopfield, 1982). More recently, several notable recurrent neural network mechanisms (*e.g.* neural turing machines, LSTMs, and RNNs with fast weights) were evaluated on a synthetic version of associative recall (Graves et al., 2014; Ba et al., 2016; Zhang and Zhou, 2017, inter alia.). These works use a formulation of associative recall very similar to the single-query associative recall in our work. Since the rise of large language models, several works have argued that LLMs ability to perform *in-context* learning is due, at least in part, to the associative recall capabilities of attention (Elhage et al., 2021; Olsson et al., 2022).

### A.2    EFFICIENT LANGUAGE MODELING ARCHITECTURES

We first briefly review the efficiency motivations for the recent excitement around gated convolution architectures. While attention requires compute that scales as $\mathcal{O}(N^2)$ in sequence length $N$, convolutions scale as $\mathcal{O}(N \log N)$ (Cooley and Tukey, 1965). Ideal, inference complexity is $\mathcal{O}(1)$ in sequence length, as provided by recurrent neural networks. State-space models can be computed either as a convolution or recurrence, to achieve both sub-quadratic training and constant inference complexity in sequence length (Gu et al., 2021). Architectures that use *implicit* convolutional filters (Poli et al., 2023a), can be converted to an SSM via a simple distillation step (Massaroli et al., 2023).

A.3    INPUT-DEPENDENCE IN SEQUENCE MODELING ARCHITECTURES

In an input-dependent sequence model, the way tokens are aggregated across the sequence is controlled by the data, not just the model parameters. We highlight several prior works related to our study that explore input-dependent sequence models.

- **Attention** (Vaswani et al., 2017) achieves input-dependent sequence mixng since the $Q, K$, and $V$ terms are achieved via linear combinations of the input $x$. For instance, $Q = xW_Q$ for some learned weight matrix $W_Q$.

- **Input-Independent Convolution Architectures**. Given the quadratic scaling of attention, Gu et al. (2021) propose S4, an architecture that use long convolutions to process the input. CKConv Romero et al. (2022) uses a convolution filter that is implicitly parametrized by an MLP of the relative positions of the observations in a time-series. A line of subsequent work improved upon S4 by changing the parametrization (Gupta et al., 2022; Gu et al., 2022; Mehta et al., 2022; Ma et al., 2022; Fu et al., 2023c) or changing the SISO convolutions to MIMO convolutions (Smith et al., 2023). The convolution filter in each of these architectures is input-independent. The linear convolution layer alone cannot perform MQAR (Fu et al., 2023a).

- **Input-Dependence via Gating**.    Since pure convolution architectures with input-independent filters struggle to perform associative recall, an important task in in-context learning (Olah, 2022), subsequent work proposed to introduce input-dependence by adding a *gating* or element-wise multiplication operation to the architecture, where both the inputs to the operator are defined in terms of the input (Fu et al., 2023a; Poli et al., 2023a). The multiplication is generally $y = \sigma(Wx) \odot x$. The filters remain input-independent. These architectures demonstrate promising results on the associative recall synthetics proposed in prior work Fu et al. (2023a); Poli et al. (2023a) and provide large downstream improvements over S4.

  Prior work suggests the models match attention in quality (Poli et al., 2023a; Fu et al., 2023b; Peng et al., 2023) and this has led to increasing use of these architectures (Nguyen et al., 2023; toe, 2023), however our results suggest there is still a sizable gap to attention. We show the gap is largely due to the models' recall abilities and capture this behavior in a novel synthetic task. We theoretically show that dimension scales in sequence length when solving MQAR with gated convolutions.

  Selective S4 Wang et al. (2023) learns which positions to mask prior to passing the input to the next long convolution (S4) layer. This resembles gating (which also controls which information flows forwards to the next layer).

- **Input-Dependent Convolution Architectures** Prior work introduces convolution architectures with input-dependent filters (Yang et al., 2020; Kosma et al., 2023). For instance, Yang et al. (2020) parametrizes the convolution filter as a linear combination of $n$ projections of the input. Liquid S4 Hasani et al. (2022) is also motivated by introducing input-dependence to S4. To accomplish this, the work proposes including correlation terms between the input tokens during state mixing. We evaluate Liquid S4 and find the architecture lags attention on the Pile by 4.4 perplexity points in Table 3.

| Model | Param (M) | Overall | Slices | | % of gap due to |
| | | | AR Hits | Other Tokens | |
| Attention | 125 | **12.37** (2.52) | **2.32 (0.84)** | 14.04 (2.64) | — |
| Liquid S4 | 145 | **16.80** (2.82) | **22.75 (3.12)** | 16.42 (2.80) | 52.6% |

Table 3: Validation perplexity on the Pile after pretraining on 5B tokens. We report log perplexity with negative log-likelihood in parentheses. AR vs. non-AR token slices are defined in Section 3.

- **Recurrent Architectures** The pure long-convolution architectures can also be computed as recurrences. Letting $s_i$ represent the hidden state at time $i$, $u_i$ be the input at time $i$, and $A, B, C$ be projection matrices, the recurrence computes:

$$s_{i+1} = f(As_i + Bu_i)$$

$$y_i = Cs_i$$

In a linear RNN like S4, the $A$, $B$, and $C$ matrices are input-independent and $f$ is an identify function. However, recent work proposes input-dependent RNNs (e.g. RetNet Sun et al. (2023)), where a subset of these matrices is input-dependent (e.g. $A = xW_a$). Note that linear attention mechanisms can also be expressed as input-dependent RNNs (Katharopoulos et al., 2020b). While these architectures improve over gated convolutions in MQAR synthetic experiments and downstream quality on the Pile (Appendix F), we observe and prove that the required RNN hidden state dimensionality grows with the number of key-value pairs that the model needs to recall in the sequence (Appendix H.6). In contrast, attention complexity does not scale with the number of key-value pairs to recall.

- **Hybrid Architectures** Finally, we can combine architectural components that provide different capabilities. Prior work proposes architectures that hybridize sub-quadratic layers and attention (Fu et al., 2023a; Ma et al., 2022; Ren et al., 2023), but does motivate this choice from a mechanistic design perspective. Further, H3 without attention Fu et al. (2023a) and MEGA (Ma et al., 2022), which uses blocked attention, underperform attention on language modeling.

  SeqBoat (Ren et al., 2023) shares closest resemblance to our learned selection module. SeqBoat introduces a different module that learns where to use attention however, their model can end up using full attention at a layer, providing quadratic scaling in the worst case. Different from SeqBoat, we use an auxiliary loss to encourage sparsity in the selected attention positions and select the top-$k$ positions to remain sub-quadratic.

Overall, our work finds that previously proposed sub-quadratic models do not efficiently solve the MQAR task.

## B  CODE LISTING

In this section, we include code listings for the BASECONV operator. We begin with the a standard BASECONV implementation that uses an explicitly-parameterized long convolution. Below, we also provide the implementation for BASECONVwith an implicitly-parameterized long convolution.

```python
import torch

def fft_conv(u: torch.Tensor, k: torch.Tensor):
    """
    Args:
        u (torch.Tensor): (batch_size, d_model, seq_len)
        k (torch.Tensor): (d_model, l_max)
    Return:
        y (torch.Tensor): (batch_size, d_model, seq_len)
    """
    seqlen = u.shape[-1]
    fft_size = 2 * seqlen
    k_f = torch.fft.rfft(k, n=fft_size) / fft_size
    u_f = torch.fft.rfft(u.to(dtype=k.dtype), n=fft_size)
    y = torch.fft.irfft(u_f * k_f, n=fft_size, norm="forward")[..., :
    seqlen]
    return y

class BaseConv(torch.nn.Module):

    def __init__(self, d_model: int, l_max: int, **kwargs):
        super().__init__()
        self.d_model, l_max = d_model, l_max,
        self.projection = torch.nn.Linear(self.d_model,  self.d_model)
        self.filter = torch.nn.Parameter(torch.randn(self.d_model, l_max)
    , requires_grad=True)

    def forward(self, u: torch.Tensor):
        """
        Args:
            u (torch.Tensor): (batch_size, d_model, seq_len)
```

```
30          Return:
31              y (torch.Tensor): (batch_size, d_model, seq_len)
32          """
33          u_conv = fft_conv(u.transpose(1, 2), self.filter).transpose(1, 2)
34          u_proj = self.projection(u)
35          y = u_conv * u_proj
36          return y + u
```

Listing 1: **Explicit BASECONV implementation.** Implementation of the BaseConv layer with explicitly-parameterized long convolutions. The implemtnation is 19 lines excluding comments and whitespace.

In some of our experiments, we interleave BASECONV layers that use explicit short convolution filters (like those in torch.nn.Conv1d) and with BASECONVlayers that use implicit long convolution filters (like those described in Poli et al. (2023a)).

Below we include the code for BASECONV with an implicit convolution.

```
1  class PositionalEmbedding(nn.Module):
2      def __init__(self, emb_dim: int, seq_len: int, **kwargs):
3          """Complex exponential positional embeddings for implicit long
   convolution filters."""
4          super().__init__()
5          t = torch.linspace(0, 1, seq_len)[None, :, None]  # 1, L, 1
6          bands = (emb_dim - 1) // 2
7          t_rescaled = torch.linspace(0, seq_len - 1, seq_len)[None, :,
   None]
8          w = 2 * math.pi * t_rescaled / seq_len  # 1, L, 1
9          f = torch.linspace(1e-4, bands - 1, bands)[None, None]
10         z = torch.exp(-1j * f * w)
11         z = torch.cat([t, z.real, z.imag], dim=-1)
12         self.z = nn.Parameter(z, requires_grad=False)
13
14     def forward(self, L):
15         return self.z[:, :L]
16
17 class BaseImplicitConv(nn.Module):
18     """
19     BaseConv with implicit filter parameterized by an MLP.
20
21     Args:
22         d_model (int): Number of expected features in input and output.
23         l_max (int): The maximum sequence length.
24         d_emb (int, optional): Dimension of the positional embeddings.
   Must be odd and $\geq$ to 3 (time, sine and cosine). Defaults to 3.
25         d_hidden (int, optional): The number of features in the hidden
   layer of the MLP. Defaults to 16.
26     """
27
28     def __init__(self, d_model: int, l_max: int, d_emb: int=3, d_hidden:
   int = 16,):
29         """
30         Long convolution with implicit filter parameterized by an MLP.
31         """
32         super().__init__()
33         self.pos_emb = PositionalEmbedding(d_emb, l_max)
34         self.filter_mlp = nn.Sequential(nn.Linear(d_emb, d_hidden), torch
   .nn.ReLU(), nn.Linear(d_hidden, d_model))
35         self.projection = torch.nn.Linear(d_model, d_model)
36
37
38     def forward(self, u: torch.Tensor, *args, **kwargs):
39         """
40         Args:
41             u (torch.Tensor): (batch_size, seq_len, d_model)
```

```
42          Return:
43              y (torch.Tensor): (batch_size, seq_len, d_model)
44          """
45          filter = self.filter_mlp(self.pos_emb(u.shape[1])).transpose(1,
     2)
46          u_conv = fft_conv(u.transpose(1, 2), filter).transpose(1, 2).to(
     dtype=u.dtype)
47          u_proj = self.projection(u)
48          y = u_conv * u_proj
49          return y + u
```

Listing 2: **Implicit BASECONV implementation.** Implementation of the BASECONV layer with implicitly-parameterized long convolutions. (The implementation is 34 lines excluding comments and whitespace).

## C    DOWNSTREAM EXPERIMENTAL DETAILS

We use A100 80GB Nvidia GPUs to run all experiments. We use the reference training infrastructure from https://github.com/EleutherAI/gpt-neox for all pretraining runs. The Pile data is tokenized using the GPT2BPETokenizer and all models see the data in the same order.

Below we provide details on the hyperparameters and settings for training each architecture studied in the paper on the real-world Pile data. In Appendix C.1 we summarize and justify our method for measuring the quality gap due to associative recall between the convolution architectures and attention. In Appendix C.2, we provide details on the pure gated convolution architectures, attention baseline, and hybrid architectures studied in the paper.

### C.1    MEASURING THE MQAR GAP ON REAL LANGUAGE DATA

Here we summarize our method for computing the amount of quality gap between the convolution and attention models that is ascribed to associative recall capability (e.g., in Table 1):

1. Given an input sequence, we identify recurring bigrams (i.e. bigrams that have already appeared in the sequence at a prior position). Since bigrams that appear frequently during training may be memorized by the model, rather than requiring the model to perform recall at inference-time, we only measure AR log-probabilities with respect to bigrams that are seen fewer than a threshold number of times during training. The threshold used in all the experiments in our submission is $1,250$ training occurrences in the 10B tokens of pretraining data.

2. We measure the log-probability assigned to the true bigram completion. This bigram completion is referred to as an AR Hit in our work. This protocol assumes that the model can produce the completion by recalling the prior occurrence of the bigram in the sequence.

3. For the model being evaluated $m$, and the attention model $M$, we measure the % of the quality gap between $m$ and $M$ ascribed to associative recall capability as follows. Let the average log-probability for all AR Hits across validation sequences be $l_H^m$ and $l_H^M$ for $m$ and $M$ respectively. Let the average log-probabilities of *all tokens* in the validation sequences be $l^m$ and $l^M$ respectively. Let $p_H$ be the proportion of AR Hit tokens in the validation data. As the final gap ascribed to AR, we report:

$$\min(\frac{(l_H^m - l_H^M)p_H}{l^m - l^M}, 1.0)$$

   Shown above, if $m$ is better than attention ($M$) overall and $M$ is better than $m$ at AR, we ascribe 100% of the gap to AR.

We briefly discuss two important decisions in this protocol. First, we only measure **explicit bigrams**, i.e. bigrams are identified based on token ids in the sequence. However, intuitively, models may also perform associative recall between related *concepts* produced by a contextual language model. For instance, language may contain bigrams in which one word is swapped by a synonym. As another example, a model may see a sentence such as "The iPhone is outside my budget so I instead purchased an Android phone. ... It was much _" and predict "cheaper" for the blank, recalling

that the sequence is discussing cost. Our work does not measure such fuzzy (more abstract) recall instances.

Next, we measure the gap based on **log-probabilities** rather than perplexity. This is simply because we want to make our metrics independent of the number of tokens in each of the slices of the validation set. Approximately 6.4% of validation tokens are AR Hits with the threshold set to consider bigrams seen less than $1,250\times$ during training.

## C.2 Gated Convolution Downstream Architectures

We evaluate over $4$ previously proposed architectures as well as BASECONV, the theoretically "canonical" representation for gated convolutions, introduced in Section 4, for a total of 14 training runs. Here we provide details on the hyperaparamters and configurations used for training each architecture. We also provide details on the FLOPs computation.

- **Attention** (Vaswani et al., 2017; Touvron et al., 2023) We train using the the specifications in Table 7. The parameters are sourced from the Transformer implementation in `https://github.com/EleutherAI/gpt-neox`.

- **Hyena** (Poli et al., 2023a) We train using the specifications in Table 11. The parameters are sourced from the Appendix of Poli et al. (2023a) and the implementation is sourced from the provided reference at `https://github.com/HazyResearch/safari`.

- **H3** (Fu et al., 2023a) We train using the specifications in Table 13. The hyperparameters and implementation use the reference at `https://github.com/HazyResearch/H3`.

- **RWKV** (Peng et al., 2023) We train using the specifications in Table 14. The parameters are sourced from the Appendix of Peng et al. (2023) and the details provided in the reference implementation at `https://github.com/BlinkDL/RWKV-LM`. We specifically evaluate RWKV-V4.

- **Pure long convolution** We train using the specifications in Table 16. We evaluate a simple long convolution based model with *no gating* as a reference point. While this is a generic architecture, we use the reference implementation and initialziations/regularizations from recent work Fu et al. (2023c). The implementation is provided at `https://github.com/HazyResearch/safari`.

- **BASECONV** We train using the specifications in Table 9. The implementation, amounting to 19 lines of PyTorch, is shown in Appendix B.

We provide the equations used to compute the FLOPs for each model, letting $D$ be the model width, $H$ the head dimension, $L$ the depth, $N$ the sequence length, $V$ the vocabulary size, and $B$ the batch size. FLOPs equations for different architecture types are provided in Table 8 (attention), Table 12 (Hyena and adapted for H3), **??** (pure long convolution), and Table 10 (BASECONV). We compute FLOPs for RWKV as in the Appendix of (Peng et al., 2023), based on the number of linear layer parameters, plus input and language modeling head FLOPs.

**Hybrid Architectures** In the main paper and appendix, we evaluate a series of architectures with a hybrid of gated convolution and non gated convolution layers. For these architectures, we use the same number of layers as the pure gated convolution architecture (as reported in Appendix C.2). The hybridized (non gated convolution) layers are inserted as replacements of the gated convolution layer. For each architecture, we simply evenly intersperse the two types of layers. We evaluate with the following replacement layers in this work:

- **Full attention** following the specification of attention in Appendix C.2.

- **Random selection** We evaluate sparse attention randomly applied to tokens as introduced in Section 5.

- **Programmatic selection** We apply sparse attention on AR hit tokens as introduced in Section 5. When processing the input sequence, we can causally determine if a bigram of raw token ids is repeated in the sequence to construct the attention pattern.

- **Learned selection** We *learn* the positions on which to use attention by introducing a simple linear layer with sigmoid activation to the architecture that is trained to output high scores

if attention should be applied to the token. We can take the top-$k$ positions to control the computational complexity of the layer. This approach is introduced in Section 5.

We use these protocols to validate that input-dependence suffices to address the associative recall gap and highlight that there are varied approaches to incorporating input-dependence in an architecture.

## D    EXTENDED DOWNSTREAM ANALYSIS OF MQAR

In this section, we provide additional analysis of how MQAR manifests in real language data. We extend our discussion of associative recall on the Pile Gao et al. (2020). We also perform analysis on sources in RedPajama including ArXiv and StackOverflow Together (2023).

### D.1    ADDITIONAL DISCUSSION OF MQAR IN THE PILE

The Pile is a widely popular language modeling corpus Gao et al. (2020).

1. First, in Appendix D.1.1, we provide several real examples of *how* associative recall occurs in the Pile to demonstrate it's role in language modeling. We color code the tokens to highlight differences in the next token predictions from Attention, RWKV and Hyena models.

2. Next, in Appendix D.1.2, we explain *where* associative recall hits tend to occur in sequences. This is useful to guide the design of sequence mixers — if associative recall tends to occur in specific places, the input-dependent sequence mixer may not need to compute all $N^2$ token-to-token interactions for every sequence.

### D.1.1    PILE EXAMPLES

For the examples below, the legend is: tokens are colored as both correct, both incorrect, Attention correct and Hyena incorrect, Attention incorrect and Hyena correct. For tokens where the models disagree, the predictions of each model are provided in parentheses.

> while lunch -ing at the Ma -ison Ber -gey b -ist -ro near his apartment : he
> had been mus -ing about `(rwkv= about, attn= on)` the ... `(723 tokens)` ...
> the young waitress -'s sigh at the Ma -ison Ber `(rwkv= Bl, attn= Ber)`
> -gey `(rwkv=-nd, attn=-gey)`

Example D.1: **Comparing next token predictions of a 350M parameter Attention model and a 350M parameter RWKV model.** In this example, the models need to perform associative recall to correctly predict "Bergey" in the 4-gram Ma-ison Ber-gey. The previous mention of the 4-gram was more than 700 tokens earlier in the passage.

> The second `(rwkv= first, attn= second)` section is all about
> Pixar Fest , and the `(rwkv= third, attn= the)` final section is
> all `(rwkv= about, attn= all)` about Pixar Pier . ... `(480 tokens)` ... -
> If `(rwkv=-Disney, attn=-If)` there wasn -âĠ -t enough Pixar at Disney-
> land , Pixar Fest `(rwkv= would, attn= Fest)` is `(rwkv= at, attn= is)`
> coming to the Disneyland Resort on April 13 , 2018 .

Example D.2: **Comparing next token predictions of a 350M parameter Attention model and a 350M parameter RWKV model.** In this example, the models need to perform associative recall to correctly predict "Fest" in the bigram Pixar Fest.

David Mus -a P -id -cock is an indigenous English re-
vert to `(rwkv=-ant, attn= to)` Islam `,(rwkv=,, attn= who)`
who formed the Islamic Party of Britain in 1989 , at
London Central Mosque `(rwkv= Mosque, attn= University)`
`.(rwkv=., attn= in)` ...(711 tokens)... -I have just found out that
David Mus `(rwkv= Cameron, attn= Mus)` -a P `(rwkv= (, attn= P)`
-id `(rwkv=-asha, attn=-id)` -cock `(rwkv=-ham, attn=-cock)`
is `(rwkv= (, attn= is)` speaking at a Yorkshire Forum debate today
in Bradford with

Example D.3: **Comparing next token predictions of a 350M parameter Attention model and a 350M parameter RWKV model.** In this example, the models need to perform associative recall to correctly predict a middle and last name. Note that the middle and last names span several tokens.

-The most common map elements `(hyena= in, attn= elements)` in
Sub -Space are prizes , or ” -g `(hyena=-points, attn=-g)` -reens
”`(hyena=-",, attn=")` ( -for their green color -). Pri -zes allow players to up-
grade their ships and gain `(hyena= other, attn= gain)` special weapons or
abilities `(hyena= abilities, attn= upgrades)` . While prizes are gener-
ally plent -ifully scattered throughout the map `(hyena= map, attn= game)` , the
upgrades or abilities `(hyena= bonuses, attn= abilities)` they award are
randomly selected by the zone . -Energy -Rather than dealing with ammunition
counts and hit points separately , Sub -Space combines both of these elements into
a single unit of measure : energy . Each ship is equipped with a certain amount of
energy , from which `(hyena= which, attn= the)` it must draw its health as
well as its weapons power . ...(1,526 tokens)... Speed Zone proved to be less
popular than the Jack -pot -/ -Running , Chaos , or ” -flag ” zone games and support
was discontinued shortly after Sub -Space `(hyena=-space, attn=-Space)`
went to retail .

Example D.4: **Comparing next token predictions of a 350M parameter Attention model and a 355M parameter Hyena model.** In this example, the models need to perform associative recall to correctly predict the name of the game "SubSpace". Note that both models correctly perform the recall when there is a short gap between tokens, but only Attention correctly performs recall when the gap is greater than 1,000 tokens.

Thus far , no systematic study has been published on is -
olation `(hyena=-olation, attn=-ot)` via sequential centrif -ug -ation , and
no systematic analysis is ...(491 tokens)... Fig . 1 `(hyena= 1, attn=-Â␣l)`
-is -olation via sequential `(hyena= centrif, attn= sequential)` centrif -ug
-ation .

Example D.5: **Comparing next token predictions of a 350M parameter Attention model and a 355M parameter Hyena model.** In this example, the models need to perform associative recall to correctly predict the technique "sequential centrifugation".

Miss -ouri `(hyena=-ouri, attn=-iss)` Southern has had 14 Major
League Baseball draft selections since the draft began in 1965 . ...(149
tokens)... Fred G . Hughes Stadium `(hyena= Stadium, attn= Field)`
( -opened in 1975 ) is named `(hyena= the, attn= named)`
after former `(hyena= the, attn= former)` J -op -lin Globe
publisher and `(hyena= and, attn= Fred)` Missouri South-
ern `(hyena= State, attn= Southern)` board of reg -ents member

Example D.6: **Comparing next token predictions of a 350M parameter Attention model and a 355M parameter Hyena model.** In this example, the models need to perform associative recall to correctly predict the name of the university "Missouri Southern".

### D.1.2    DISTRIBUTION OF MQAR HIT POSITIONS IN SEQUENCES

In Figure 3, we compute and plot the distances between AR hits and its prior bigram (key-value) occurrence in the sequence across the Pile training data. The distances follow a power law distribution where most AR hits are within 100 token positions from the prior bigram occurrence, and a long tail of AR hits requires long-range interactions.

**Implications**   This suggests that architectures which compute token-to-token interactions within windows, where the window size is less than the full sequence length, may suffice to handle MQAR in real data. For instance, sliding window attention may help with AR (Beltagy et al., 2020; Sun et al., 2023, inter alia.). Prior work leverages the observation that local attention captures most of the token dependencies to improve efficiency Sukhbaatar et al. (2019).

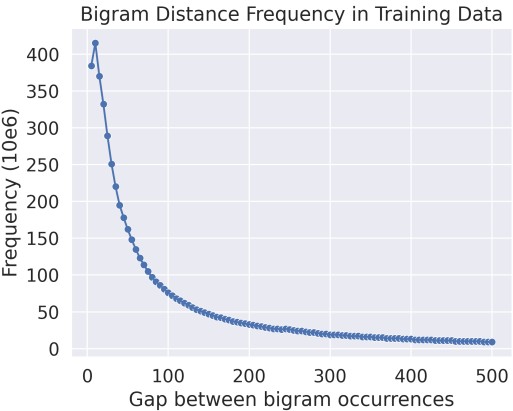

Figure 3: Across the Pile training data, we measure the distances between $n$-grams and their prior occurrences in the provided context. We plot the frequency across distances, finding it follows a power law distribution.

### D.2    ANALYZING MQAR ACROSS VARIED LANGUAGE DISTRIBUTIONS

We use the HuggingFace sample of the popular RedPajama language modeling corpus Together (2023) to understand how the prevalence of MQAR hits and and ways in which the hits appear vary across language distributions. We analyze text sequences from the following sources: ArXiv papers, Books, C4, Common Crawl, GitHub, Stack Exchange, and Wikipedia.

First we profile the prevalence of MQAR hits by computing the number of repeated bigram occurrences per sequence. Each sequence is $2,048$ tokens in length and exclude bigrams containing NLTK stop words or punctuation from the computation. The prevalence of hits and the distances between hit tokens and the prior bigram occurrence in context vary across distributions as shown in Figure 4 and 5.

Arxiv, Github, and Stack Exchange contain relatively structured or richly formatted langauge data. We observe that the prevalence of MQAR hits is relatively high in these three sources in Figure 4. We inspect why these documents contain many bigrams. In Arxiv, we find domain-specific terminology, groups of related citations, Latex commands, and mathematical equations are frequently repeated within the same document. In GitHub and Stack Exchange, we find function and variable names are frequently reused. Meanwhile, the C4, CC, Books, and Wikipedia bigrams are highly variable. The topics and documents in these corpora are relatively diverse and unstructured.

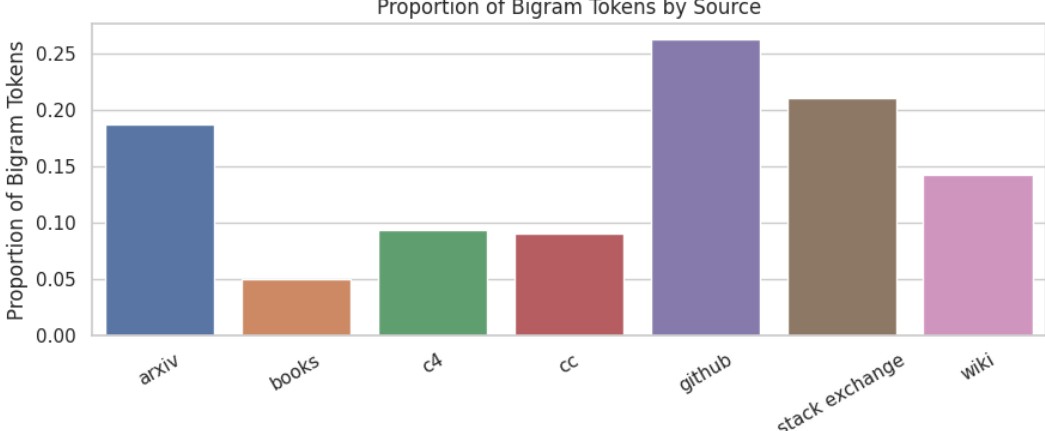

Figure 4: MQAR hits by sub-source of the RedPajama language corpus Together (2023). Hits are measured using the repeated bigrams in each sequence of length .

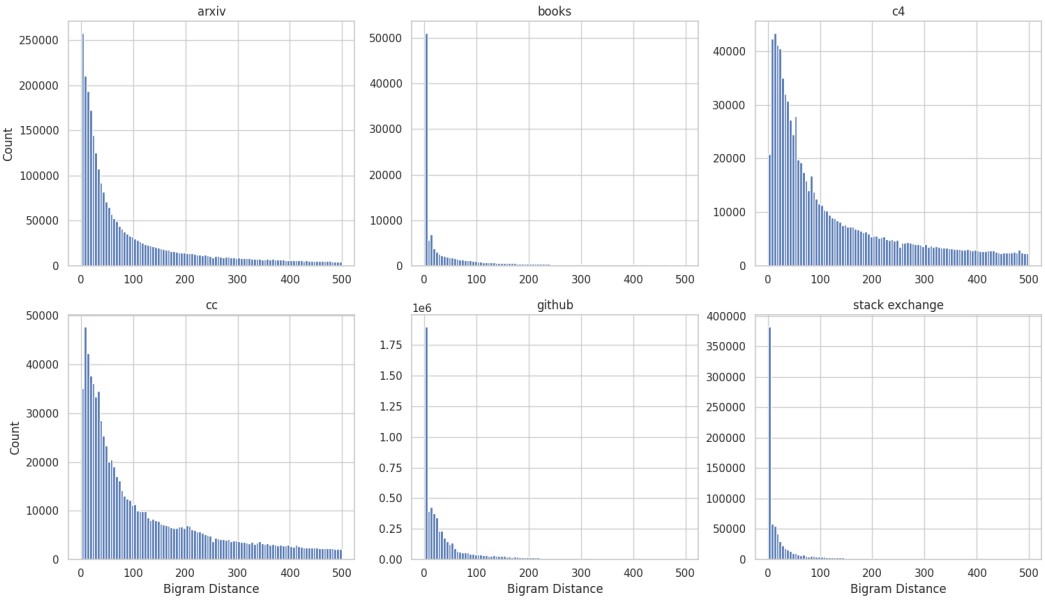

Figure 5: We measure the distance between $n$-grams and their prior occurrences in the provided context across sequences from each source. We plot the frequency across distances, finding it follows a power law distribution.

> If the placement $V$ of the hyperplanes is not generic, then the induced subdivision $\Sigma$ is not a triangulation. However, the above bijection is unaffected and, in particular, the fine type of a vertex of $CD_A$ can be read off the corresponding facet of $\Sigma$. We define the *crosscut complex* $ccut\Sigma \subseteq 2^{[n] \times [d]}$ to be the unique simplicial complex with the same vertices-in-facets incidences as the polyhedral complex $\Sigma$. The crosscut complex is a standard notion in combinatorial topology and can be defined in more generality (see Björner). The following observation is immediate from the definitions.
>
> **Proposition D.1.** *For $V$ an ordered sequence of $n$ points in $T^{d-1}$ let $A = A(V)$ be the corresponding tropical arrangement and let $\Sigma$ be the regular subdivision of $\Delta_{n-1} \times \Delta_{d-1}$ induced by $V$. Then the fine cotype ideal $fcI$ is Alexander dual to the Stanley-Reisner ideal of $ccut\Sigma$.*
>
> The crosscut complex encodes the information of which collections of vertices lie in a common face. Hence, the crosscut complex is a purely combinatorial object and does not see the affine structure of the underlying polyhedral complex.

Example D.7: ArXiv sequence where the initial bigram occurrence is highlighted in red and the repeated bigram with the MQAR hit is highlighted in blue.

```java
/**
 * Adds a directed edge to the graph from pt1 to pt2. Precondition: Both
 * GeographicPoints have already been added to the graph
 *
 * @param from The starting point of the edge
 * @param to The ending point of the edge
 * @param roadName The name of the road
 * @param roadType The type of the road
 * @param length The length of the road, in km
 * @throws IllegalArgumentException If the points have not already been
 * added as nodes to the graph, if any of the arguments is null, or if the
 * length is less than 0.
 */
public void addEdge(GeographicPoint from, GeographicPoint to,
        String roadName, String roadType, double length)
        throws IllegalArgumentException {

    if (from == null || to == null || roadName == null
            || roadType == null || length < 0 || !mapNodes.containsKey(from)
            || !mapNodes.containsKey(to)) {
        throw new IllegalArgumentException();
    }

    if (!mapNodes.get(from).hasEdge(to)) {
        mapNodes.get(from).addNeighbor(to, roadName, roadType, length);
        numEdges++;
    }
}
```

Example D.8: GitHub sequence where substrings such as "IllegalArgumentException", "roadType" (and other variable names), "GeographicPoint" (and other data types), "containsKey" (and other function calls) are repeated throughout.

# E    SYNTHETIC MQAR EXPERIMENTAL DETAILS

In this paper, we propose MQAR as a useful tool to help explain gaps between three popular sequence modeling layers — Transformers or Sparse Transformers, Convolutions, and Recurrences. In this section, we detail the motivation behind the design of MQAR and include extended experiments on additional architectures. We provide a procedure for generating MQAR synthetic data, which can help in the development of new architectures.

## E.1    MQAR GENERATION PROCEDURE

Here we provide additional discussion on the properties of our MQAR synthetic data. The objective of the synthetic analysis is to help explain the differences in language modeling behavior between different classes of language modeling architectures, as observed on real-world data Section 3. Synthetic recall tasks were used in the development of Hyena Poli et al. (2023a) and H3 Fu et al. (2023a), and Olsson et al. (2022) to study Transformer in-context learning behavior.

1. **Attention**: Attention can perform recall trivially (Proposition 4.3). The bound is independent of the sequence length $N$.
2. **Convolutions**: The gated convolution models use input-independent filters. We show in Section 4 that the dimensionality for solving recall grows with the input sequence length. Real language modeling can require performing $\mathcal{O}(N)$ recalls in one forward pass. Our intuition is that it is difficult to efficiently compute **all token-interaction distances** with such a filter.
3. **Recurrences**: Recurrent models include a single hidden state as an information bottleneck. We show in Appendix H.6 that the gated recurrence requires $\Omega(N)$ bits to solve MQAR for $d \leq \sqrt{N}$ for model dimension $d$. Our intuition is that it is difficult to store **large numbers of key-value pairs** seen in the sequence in low-dimensional hidden states.

Motivated by this analysis, we propose Procedure 1 for generating synthetic MQAR data. This procedure allows toggling two simple properties of the synthetic dataset, specified below, to reveal the differences between the sequence modeling architecture families. We also discuss deviations from the prior work that uses synthetic recall data.

1. **Size of the Key-Value Map**: The number of unique key-value pairs appearing in each example in the dataset. In Procedure 1, we use the input parameter $D$ to toggle this value.
2. **Number of Gaps in the Data**: The number of unique token-interaction distances required to perform the recall task. Prior work assumes there is a single query token that requires recall per example, failing to test whether the architecture can support multiple token-interaction distances. In Procedure 1, we use the parameters $N$ (sequence length), $D$ (size of the key-value map per example), and $\alpha$ to toggle this property. We enforce that the token-interaction distances appearing in the synthetic examples follow a power-law distribution specified by $\alpha$ (based on Figure 3). Keeping $\alpha$ and $D$ constant while varying $N$ changes the number of unique token-interaction distances appearing in the example.

Finally, noting that the vocabulary size appears in the theoretical bounds for all three architecture families (Section 4), we increase the vocabulary size to be much larger than the model dimension as typically seen in language modeling (30k - 50k tokens). Prior work uses vocab sizes $\leq 40$ tokens.

---

**Algorithm 1** MQAR Synthetic Data Generation Procedure

---

**Input:**  Vocabulary $C$, Sequence length $N$, Power-law parameter $\alpha$, Number of Key-Value Pairs $D$
   **Output: Synthetic sequence**
1: Let the first half of $C$ be keys $K$ and the second half be values $V$.
2: Pair each key token $k \in K$ with a *random* value token $v \in V$.
3: Sub-select $D$ random key-value pairs to include in the sequence.
4: Place the $D$ key-value pairs at the start of the sequence (i.e. consuming the first $2D$ positions).
5: Place a second occurrence of each $d \in D$ at a distance from the first occurrence in the sequence. The distance for each $d \in D$ is selected at random from the positions $[2D..N]$, where the probability of choosing each position follows the power law distribution specified by $\alpha$.
6: Output the synthetic sequence.

---

## E.2    TRAINING DETAILS

We first describe the architectures evaluated on MQAR synthetic data and then the training hyperparameters. We evaluate the following four architecture categories in the experiments:

1. **Attention** Standard GPT-2 style multi-headed Transformer architecture with learned positional embeddings Brown et al. (2020). Attention, the core building block of Transformers, is defined in Section 2. The architecture for synthetics has 1 attention head.
2. **Gated convolutions** (Hyena Poli et al. (2023a), RWKV Peng et al. (2023)). Gating, convolutions, and the class of gated convolutions are defined in Section 2.
3. **Gated recurrences** (RetNet Sun et al. (2023)). RetNet proposes computing attention over chunks of the sequence and combining this with a recurrence over the sequence. We evaluate this architecture ("Chunked RNN") with chunk sizes of 32 and 8 in Figure 6.
   In RetNet, hidden states are updated as:

$$S_n = \gamma S_{n-1} + A_n^T V_n$$

   Outputs at each timestep are defined as

$$O_n = C_n S_n$$

   where $\gamma \in \mathbb{R}^1$, $A = xW_a$ for input $x \in \mathbb{N} \times$ and learned weight matrix $W_a \in \mathbb{R}^{d \times d}$, $C = xW_c$ for $W_c \in \mathbb{R}^{d \times d}$, $V = xW_v$ for $W_v \in \mathbb{R}^{d \times d}$.
   RetNet similar to a state-space-model (SSM) Gu et al. (2021), except for that the matrices $A$ and $C$ are *input-dependent* (i.e. they are functions of the input $x$). We note that RetNet is a special case of the recently proposed Mamba Anonymous (2023a) and GateLoop Anonymous (2023b) architectures, which replace the $\gamma$ term in RetNet with yet another input-dependent matrix.
   The RetNet model is formally defined in Appendix H.2.3.
4. **Sparse attention** Several works have proposed methods for efficiently computing attention Tay et al. (2022). We evaluate two classical methods: sliding window attention Beltagy et al. (2020); Zaheer et al. (2020) and blocked window attention Child et al. (2019b); Qiu et al. (2019).
   Consider a window size $w$. Sliding window attention permits each token to attend to the prior $w$ tokens in the sequence. Blocked window attention first splits the sequence into blocks of size $w$ tokens, and then computes standard causal attention within the block.

For all synthetic runs in the main paper and the appendix, we use the following training protocol:
- **Optimizer and schedule**: Weight decay 0.1, warmup duration $10\%$, linear warmup, AdamW optimizer. For each run, we sweep the learning rates in $\mathrm{np.logspace}(-4, -2, 4)$. We train for 64 epochs.
- **Training duration**: The global batch size is 8 for input sequence length or model dimension $\geq 512$, 16 for input sequence length or model dimension $\geq 256$, and 64 otherwise.
- **Width and depth**: For all synthetic experiments, we use exactly two-four layers (each with one sequence mixer and one MLP, interleaved with layer normalization). The model dimension, sequence length, and number of KV pairs are varied based on the relevant experiment (see Appendix F).
- **Position information**: No position embeddings are used for the pure convolution runs (input-dependent nor input-independent filters). Position embeddings are used for the runs with any attention variant.
- **Data**: We train and evaluate each model on $100,000$ and $3,000$ data points respectively. The data and data order for all runs are constant.

## F EXTENDED RESULTS ON MQAR ACROSS ARCHITECTURES

In this section, we provide further validation that MQAR is a useful diagnostic tool to explain the behaviors of a wide variety of architecture families. We show:
1. **Gated convolutions** In the main paper, we claim the gated convolution models with input-independent filters use larger dimensionality than attention as the number of token-interaction distances required for the task increases ($N$, holding $\alpha$ and $D$ constant in Procedure 1).
2. **Multi-head gated convolutions** Recent work Fu et al. (2023a); Massaroli et al. (2023); Fu et al. (2023b) uses *multi-head* convolutions. This extension groups the model dimension into heads (of dimension $d_h = \frac{d}{H}$ for $d$ model dimension and $H$ heads), computes the outer product between different input projections in each head (resulting in $\mathbb{R}^{N \times d_h \times d_h}$ tensors), passes the results through the long convolution, and multiplies with another projection of

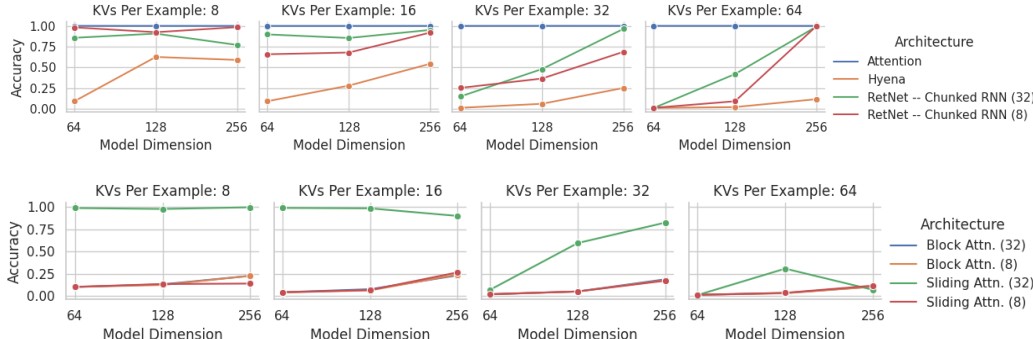

Figure 6: MQAR quality as we vary the model dimension for synthetic datasets that contain different numbers of key-value pairs per example. (**Top**) We focus on the evaluation of the input-dependent gated recurrent RetNet (Sun et al., 2023) architecture. The architecture combines chunked attention with recurrence and we evaluate at two different chunk sizes (8 and 32). For reference, we include gated convolution (Hyena) and attention. (**Bottom**) We evaluate two efficient attention variants – sliding window and blocked window attention – at two window sizes (8 and 32).

> the inputs to return to $\mathbb{R}^{N \times d_h}$, then concatenates. The full algorithm is provided in Fu et al. (2023b) (Appendix Algorithm 1).
>
> 3. **Gated recurrences** In this section, we show gated recurrent models use larger dimensionality than attention to solve the task as the number of unique key-value pairs to store ($D$ in Procedure 1) increases.
>
> 4. **Sparse attention** In this section, we validate that sliding window attention helps close the MQAR gap when the token-interaction distances are relatively short and degrades on longer distances. Meanwhile, blocked window attention struggles to close the gap (intuitively tokens earlier in a block are able to interact with few other tokens in the sequence). Concretely, in Appendix D, we observe that repeated bigrams in the Pile and RedPajama corpora tend to occur within neighboring tokens in the context.

We finally show the architectures that perform well on MQAR also perform well on downstream real-world AR on the Pile. We use the experimental protocols outlined in Appendix E for the synthetic experiments and Appendix C for the downstream experiments in this section.

### F.1 SYNTHETIC EXPERIMENTS

**Experiment 1: Increasing the number of token-interaction distances per example.** To construct the synthetic dataset following Procedure 1, we set $\alpha = 0.1$ and $|C| = 8,192$ for all experiments. We vary $N \in \{64, 128, 256, 512\}$ to increase the number of token-interaction distances that will appear in the examples and we vary the model dimension $\in \{64, 128, 256, 512\}$. The results of this experiment are shown in Fig. 2, validating that the gated convolutiosn use larger dimensionality than attention as the number of token-interaction distances required for the task increases.

**Experiment 2: Increasing the number of key-value pairs occurring per example.** To construct the synthetic dataset following Procedure 1, we set $\alpha = 0.1$ and $|C| = 8,192$ for all experiments. We fix $N = 256$ and vary $D \in \{\frac{N}{32}, \frac{N}{16}, \frac{N}{8}, \frac{N}{4}\}$ to increase the number of key-value pairs occurring per example. For each number of key-value pairs per example, we additionally vary the model dimension $\in \{64, 128, 256\}$. The results are shown in Fig. 6. We validate that as the number of associative recall keys and values exceeds the window size, the model again uses increased dimensionality relative to full $O(N^2)$ attention to solve the task. While Fig. 6 highlights RetNet, we note that it is a special case of the recently released Mamba Anonymous (2023a) and GateLoop Anonymous (2023b) architectures. Corresponding to this experiment, we theoretically show in Appendix H.6 that the gated recurrence uses $\Omega(N)$ bits to solve MQAR for $d \leq \sqrt{N}$. Intuitively, it is difficult to store a large number of key-value pairs in low dimensional hidden states.

**Experiment 3: Increasing the range of token-interaction distancs per example.** To construct the synthetic dataset following Procedure 1, we set $\alpha = 0.1$ and $|C| = 8,192$ for all experiments. We fix $N = 256$ and vary $D \in \{\frac{N}{32}, \frac{N}{16}, \frac{N}{8}, \frac{N}{4}\}$ to increase the number of key-value pairs occurring

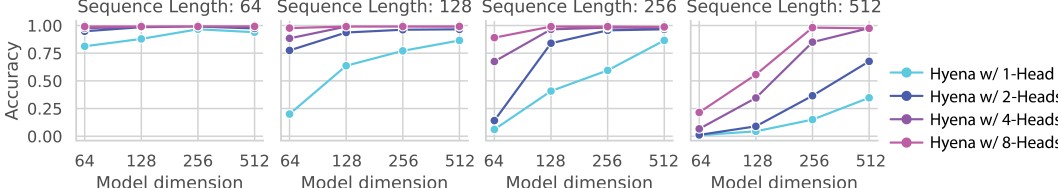

Figure 7: MQAR quality as we vary the model dimension. We consider number of convolution heads $\in \{1, 2, 4, 8\}$ and observe more heads improves scaling (at the cost of additional FLOPs).

| Model | Param (M) | TFLOPs | Overall | Slices | | % gap due to |
|---|---|---|---|---|---|---|
| | | | | AR Hits | Other Tokens | AR Hits |
| Attention | 73 | 1.52 | **12.99 (2.56)** | **2.41 (0.88)** | 14.76 (2.69) | — |
| Long Conv | 76 | 1.20 | 20.28 (3.01) | 40.25 (3.70) | 19.25 (2.96) | 44.4% |
| H3 | 72 | 1.33 | 15.78 (2.76) | 13.86 (2.63) | 15.94 (2.77) | 63.2% |
| Hyena | 72 | 1.34 | 15.13 (2.72) | 9.00 (2.20) | 15.74 (2.76) | 60.8% |
| RWKV | 72 | 1.89 | 16.10 (2.78) | 14.11 (2.65) | 16.26 (2.79) | 57.9% |
| Attention | 125 | 2.46 | **11.01** (2.40) | **2.16 (0.77)** | 12.45 (2.52) | — |
| Long Conv | 128 | 1.74 | 16.98 (2.83) | 25.62 (3.24) | 16.46 (2.80) | 40.1% |
| H3 | 168 | 2.55 | 12.06 (2.49) | 6.75 (1.91) | 12.60 (2.53) | 88.4% |
| RWKV | 169 | 2.08 | 11.64 (2.45) | 5.70 (1.74) | 12.29 (2.51) | 100.0% |
| RetNet (128) | 152 | 2.60 | 11.15 (2.41) | 3.01 (1.10) | 12.45 (2.55) | 100% |
| Hyena | 160 | 2.41 | 11.60 (2.45) | 5.00 (1.61) | 12.28 (2.51) | 100% |
| + Slide Attn. (64) | 159 | 2.28 | 11.57 (2.45) | 5.23 (1.65) | 12.30 (2.51) | 100% |
| + Slide Attn. (256) | 159 | 2.29 | **10.65** (2.37) | 3.42 (1.23) | 11.52 (2.45) | 100% |
| Attention | 360 | 6.23 | **9.44** (2.25) | **1.98 (0.69)** | 10.62 (2.36) | — |
| Long Conv | 360 | 4.08 | 13.13 (2.57) | 13.27 (2.59) | 13.12 (2.57) | 40.5% |
| H3 | 357 | 4.85 | 10.38 (2.34) | 4.81 (1.57) | 11.00 (2.40) | 65.8% |
| Hyena | 358 | 5.03 | 10.07 (2.31) | 3.83 (1.34) | 10.75 (2.38) | 98.2% |
| RWKV | 351 | 4.31 | 9.79 (2.28) | 3.82 (1.34) | 10.51 (2.35) | 100.0% |

Table 4: **Language modeling validation perplexity on the Pile.** After pretraining on 10B tokens of Pile data, we report log perplexity with negative log-likelihood in parentheses. We report overall scores, and for the AR vs. non-AR token slices defined in Section 3. FLOPs are computed for inputs of 2048 tokens based on the equations in Appendix C.

per example and the range of token-interaction distances per example. For each number of key-value pairs per example, we additionally vary the model dimension $\in \{64, 128, 256\}$.

The results are shown in Fig. 6, validating that sliding window attention closes the gap to attention when the number of KV pairs is within the window size. I.e., 16 KVs means that there are 16 keys and 16 values, for 32 total tokens. Sliding attention with window size 32 performs well up until 16 KVs and degrades beyond this point relative to attention models of the same dimensionality. Sliding window attention with window size 8 does not perform well on any setting, as expected. Blocked attention also not perform well – intuitively tokens at the beginning of a block lack sufficient prior context to attend to.

**Experiment 4: Increasing the number of convolution heads.** In Figure 7, we vary the number of heads in the gated convolution layers of Hyena following Algorithm 1 in Fu et al. (2023b). We observe that the models with more heads achieve higher recall quality at a fixed model dimension. This provides empirical evidence for theoretical arguments in Massaroli et al. (2023) that suggest multi-head convolutions increase the effective dimension and enable solving associative recall with reduced model dimension vs. single-head convolutions. However, additional heads increase the FLOP count and we continue to observe large gaps relative to attention-level recall quality.

## F.2 Demonstrating MQAR as a Tool for Architecture Development

In Table 4, Table 5 we show that the trends of RetNet, sparse attention, and multi-headed gated long convolutions on MQAR also carry to the downstream Pile results. We believe the MQAR synthetic may be useful in future architectural development.

| Model | Param (M) | TFLOPs | Overall | Slices | | % of gap due to |
|-------|-----------|--------|---------|--------|--------------|----------------|
| | | | | AR Hits | Other Tokens | AR Hits |
| Attention | 125 | 2.46 | **11.01** (2.40) | **2.16 (0.77)** | 12.45 (2.52) | — |
| Attention | 360 | 6.23 | **9.44** (2.25) | **1.98 (0.69)** | 10.62 (2.36) | — |
| Hyena ($w = 14$) | 158 | 2.41 | 11.60 (2.45) | 5.00 (1.61) | 12.28 (2.51) | 100% |
| Hyena ($w = 2$) | 158 | 2.41 | 11.47 (2.44) | 4.63 (1.53) | 12.30 (2.51) | 100% |
| + 4 Heads | 150 | 2.47 | 11.05 (2.40) | 3.47 (1.24) | 12.07 (2.49) | 100% |
| Hyena ($w = 14$) | 358 | 5.03 | 10.07 (2.31) | 3.83 (1.34) | 10.75 (2.38) | 98.2% |
| Hyena ($w = 2$) | 358 | 5.03 | 9.86 (2.29) | 3.46 (1.24) | 10.69 (2.37) | 100% |
| + 4 Heads | 358 | 5.18 | 9.48 (2.25) | 2.78 (1.02) | 10.42 (2.34) | 100% |

Table 5: **Language modeling validation perplexity on the Pile.** After pretraining on 10B tokens of Pile data, we report log perplexity with negative log-likelihood in parentheses. We report overall scores, and for the AR vs. non-AR token slices defined in Section 3. FLOPs are computed for inputs of 2048 tokens based on the equations in Appendix C. Here we evaluate Hyena at two different hyperparameter settings, sign frequencies $w$ ($w = 14$ is the default in Poli et al. (2023a)), and in the multi-head setting (described in Algorithm 1 (Fu et al., 2023b)).

## G   MQAR PERPLEXITY GAP AND MODEL SIZE

In this section, we investigate how the associative recall gap changes as we increase the model size to a billion parameters and beyond. Overall, the results suggest that the larger models improve in memorizing bigrams from the training data, but do not rapidly improve in in-context learning (perplexity on rare bigrams).

Below, we pretrain at the 1.4Bn parameter scale and observe that the 70M parameter attention model is a full perplexity point better on AR than this 1.4Bn Hyena model (Table 1). We discuss these results further in (Section 3).

| Model | Param (B) | TFLOPs | Overall | Slices | | % of gap due to |
|-------|-----------|--------|---------|--------|--------------|----------------|
| | | | | AR Hits | Other Tokens | AR Hits |
| Attention | 1.4 | 1.52 | 8.19 (2.10) | 1.91 (0.65) | 9.86 (2.29) | — |
| Hyena | 1.4 | 1.20 | 9.65 (2.27) | 3.43 (1.23) | 11.01 (2.40) | 40.3% |

Table 6: **Large-scale language model validation perplexity on the Pile.** After pretraining on 50B tokens of Pile data, we report log perplexity with negative log-likelihood in parentheses. We report overall scores, and for the AR vs. non-AR token slices defined in Section 3. FLOPs are computed for inputs of 2048 tokens based on the equations in Appendix C.

**Open-source 7B parameter models**   We next evaluate the RWKV (gated convolution) Peng et al. (2023) and Llama 2 (attention) pretrained models at the 7B parameter scale Touvron et al. (2023).[9] These are both popular models that took a significant amount of effort to train, towards maximizing quality. We find that there is a gap between RWKV and attention at the 7B scale and that it increases as the model needs to conduct more recalls per sequence ($P$ below).
We summarize the experimental protocol below.

1. **Justifying the experimental protocol.** Since frequent bigrams may just be memorized by the model and not require in-context recall, our work measures AR quality on infrequent bigrams in validation sequences (Figure 1, Appendix C.1). We do not have access to the custom training data mixtures used in training RWKV or Llama 2 to measure bigram frequencies, so we use a synthetic test to fairly measure the AR capabilities.
2. **Evaluation dataset.** We construct the synthetic dataset following Algorithm 1, where each sequence contains $P$ token pairs, or "bigrams". Each bigram, containing a "key" token and a "value" token, appears twice in the sequence. On the second occurrence of a "key" token, the model should look back to the prior occurrence to output the corresponding

---

[9]We specifically evaluate the RWKV-Raven model downloaded from `https://huggingface.co/docs/transformers/model_doc/rwkv` and the Llama 2 model downloaded from `https://github.com/facebookresearch/llama`.

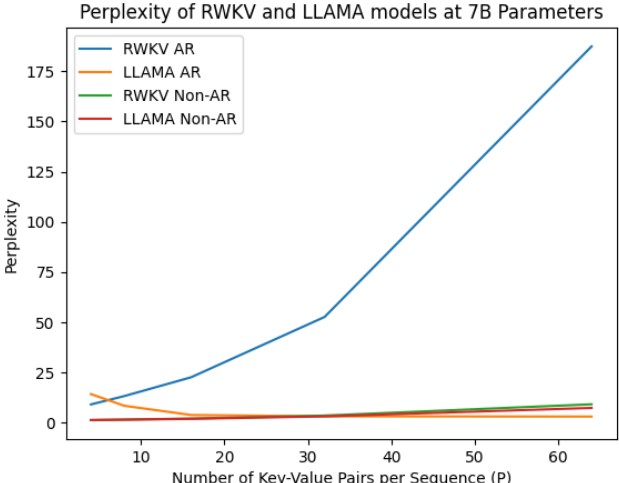

Figure 8: Perplexity of RWKV 7B and Llama2 7B parameter models on synthetic MQAR data as a function of the number of recalls required per input / number of key-value pairs per input ($P$). Inputs are constructed using each model's full vocabulary.

> "value" token. We measure the AR perplexity of the model based on its ability to predict the correct "value" token as the next word for these repeated keys. The sequences are constructed using the models' vocabulary tokens.
>
> 3. **Evaluation details.** We evaluate (inference only, no training) on sequence lengths of 1024 tokens. We measure the AR perplexity when the sequences contain $P \in \{16, 32, 64, 128, 256\}$ key-value pairs, using 1000 samples per $P$ value. The tokens that do not contain a key or value are simply filled with a fixed token id (so this token is repeated frequently in the sequence). We plot perplexity for AR and non-AR tokens (fixed token) vs. $P$. We find RWKV quality degrades with $P$ on the AR slice (blue line), while all other lines remain flat. MQAR remains problematic at scale.

Overall our results suggest that simply scaling the model size may not close the MQAR quality gap between the gated convolutions and attention.

**Measuring the AR Gap at Scale** In the main paper, we measure the AR perplexity on downstream data based on bigrams that are seen $< 1,250\times$ during pretraining. However, we hypothesize that larger models may memorize a larger number of bigram pairs seen in the training dataset, but do not rapidly gain the capability to perform associative recall as well as attention in-context. In-context learning is defined as learning from examples *provided in context* (Xie et al., 2021).

Concretely, the gap between the Hyena / RWKV and attention models at the 350m scale is 1.85 / 1.84 perplexity points when we focus on bigrams seen $< 1,250\times$ during pretraining (Table 1). If we instead focus on bigrams seen $1\times$ during pretraining, the gap to attention quality is 12.0 / 13.2 perplexity points respectively. The gated convolutions appear to struggle in the regime of rare bigrams that require the model to use the context.

## H  DETAILS ON THEORETICAL ANALYSIS

This section provides proofs and extensions to the theoretical results in the main paper.

### H.1  PRELIMINARIES AND NOTATION

#### H.1.1  NOTATION

We denote the all 1 row vector of size $k$, given by $[1 \quad 1 \quad \ldots \quad 1 \quad 1]$, and the all 0 row vector of size $k$, given by $[0 \quad 0 \quad \ldots \quad 0 \quad 0]$, as $\mathbf{1}^k$ and $\mathbf{0}^k$, respectively. We also construe the standard basis vector $\mathbf{e}_i$ as a column vector in these notes, and adhere to the following matrix indexing convention: $\mathbf{M}[i,j]$ is the entry in the $i$th row and the $j$th column, $\mathbf{M}[i,:] \in \mathbb{F}^{1 \times n}$ denotes the $i$th row, and $\mathbf{M}[:,j] \in \mathbb{F}^{m \times 1}$ denotes the $j$th column of $\mathbf{M} \in \mathbb{F}^{m \times n}$. We then use $\mathbf{1}^{m \times n}, \mathbf{0}^{m \times n} \in \mathbb{F}^{m \times 1}$ to denote the matrix of all 1s and 0s, respectively.

Next, we denote the *Hadamard product* of vectors $\mathbf{u}, \mathbf{v} \in \mathbb{F}^n$ as $\mathbf{u} \odot \mathbf{v}$; the operation can be extended to matrices by applying the Hadamard product column-wise across the matrices. This is commonly referred to as *(element-wise) gating*. For vectors $\mathbf{u}, \mathbf{v} \in \mathbb{F}^n$, we also denote their *linear (or acyclic) convolution* as $\mathbf{u} * \mathbf{v}$ and *cyclic convolution* as $\mathbf{u} \circledast \mathbf{v}$.

**Polynomial Notation.** Because convolution is intimately tied to operations on polynomials, it is convenient to use them to discuss the inputs and outputs of gated convolution models. Let us define maps $\mathrm{poly}, \mathrm{poly}^* : \mathbb{F}^n \to \mathbb{F}[X]/(X^n)$ such that

$$\mathrm{poly}(\boldsymbol{u}) = \sum_{i=0}^{n-1} \boldsymbol{u}[i] X^i, \text{ and } \mathrm{poly}^*(\boldsymbol{u}) = \sum_{i=0}^{n-1} \boldsymbol{u}[i] X^{n-1-i}.$$

This allows us to map between vectors and polynomial. Accordingly, we also define $\mathrm{coeff}$ : $\mathbb{F}[X]/(X^{n+1}) \to \mathbb{F}^n$ as the map converting polynomials back to vectors: $\mathrm{coeff}(\boldsymbol{u}(X)) = \boldsymbol{u}$ with $\boldsymbol{u}[i]$ defined as the coefficient in $\boldsymbol{u}(X)$ at degree $i$.

These operations allow us to interpret the convolution of vectors in terms of polynomial multiplication (Heideman and Burrus, 1988). More specifically, we have

$$\boldsymbol{u} * \boldsymbol{v} = \mathrm{coeff}\left(\boldsymbol{u}(X) \cdot \boldsymbol{v}(X) \mod X^n\right), \text{ and}$$
$$\boldsymbol{u} \circledast \boldsymbol{v} = \mathrm{coeff}\left(\boldsymbol{u}(X) \cdot \boldsymbol{v}(X) \mod X^n - 1\right).$$

We can similarly interpret the Hadamard product of vectors $\boldsymbol{u} \odot \boldsymbol{v}$ as the Hadamard product of polynomials $\boldsymbol{u}(X) \odot \boldsymbol{v}(X)$:

$$\boldsymbol{u} \odot \boldsymbol{v} = \mathrm{coeff}\left(\boldsymbol{u}(X) \odot \boldsymbol{v}(X)\right) = \mathrm{coeff}\left(\sum_{i=0}^{n-1}(\boldsymbol{u}[i] \cdot \boldsymbol{v}[i]) \cdot X^i\right).$$

**Arithmetic Circuit Notation.** We briefly introduce the notation of arithmetic circuits (Volkovich, 2016), the focus of Appendix H.5. An *arithmetic circuit* $\mathcal{C}$ with variables $X \triangleq \{x_1, x_2, \ldots, x_n\}$ over a field $\mathbb{F}$ is interpreted as a directed acyclic graph, where the input nodes are labelled by either the variables from $X$ or constants from $\mathbb{F}$ and the internal nodes are labelled by $+$ or $\times$ with the output being the polynomial computed at the output node.

We shall also refer to the *size* of the circuit as the number of nodes, the *depth* of the circuit as the length of the longest path between an input node and the output node, and the *width* of the circuit as the number of parallel operations in the circuit, or 'wires' which will be intersected by a horizontal 'cut' through the circuit. Moreover, the *degree* of a circuit is defined as the degree of the polynomial computed by the circuit. We summarize this with the following definition:

**Definition H.1.** An arithmetic circuit $\mathcal{C}$ is an $(n, s, \Delta, w)$-*circuit* if $\mathcal{C}$ is an $n$-variate arithmetic circuit of size $s$ and of depth at most $\Delta$, and width $w$.

**Model Notation.** Now we introduce the notation we will be using for defining layers. In what follows, we denote $\boldsymbol{u} \in \mathbb{R}^{N \times d}$ as the model input; $N$ as the sequence length; $L$ as the number of stacked layers, indexed by $\ell$; and $d$ as the input (embedding) dimension.

### H.1.2 SUMMARY OF THE RESULTS

The outline of our results are as follows: In Appendix H.2 we introduce *gated convolution models* and define H3, Hyena, RWKV, RetNet, and BASECONV. In Appendix H.3 we introduce a set of primitive operations that BASECONV can implement. We use these as tools in the subsequent proofs. Then, in Appendix H.5, we show that a general arithmetic circuit of size $s$ and degree at most $\Delta$ can be simulated by BASECONV. We use the above results to show all models built from gating and convolutions can be simulated by BASECONV. This helps us analyze the *class* of gated convolutions, beyond a specific proposal (e.g. Hyena).

Next, we study the representational power that gated convolutions and attention use to solve MQAR. In Appendix H.7, we derive a BASECONV model inspired from dyadic intervals to show the dimensionality for gated convolutions (with input-independent filters) solve MQAR. Next, we analyze the dimensionality for a BASECONV model with data-dependent kernels to solve MQAR in Appendix H.8.

### H.2 GATED ATTENTION-FREE MODELS

We now present formal definitions of gated convolution and recurrence models with respect to the 5-tuple of parameters $(N, L, d, N', d')$.

**Definition H.2.** An $(N, L, d, N', d') -$ Gated Convolution Model is a stacked sequence to sequence model with $L$ layers such that:

1. Input and output are $N \times d$ matrices,
2. Each layer's operations consist of element-wise gating, convolution, linear projection, and
3. All the individual gated convolution layers take in $N' \times d'$ matrices and output $N' \times d'$ matrices. We refer to the tuple $(N', d')$ as the *inner dimension* of the model.

We define the Hyena, RWKV, RetNet, and BASECONV layers to make step 2 more concrete. We also assume that the input $\boldsymbol{u} \in \mathbb{R}^{N \times d}$ is embedded into $\boldsymbol{u}' \in \mathbb{R}^{N' \times d'}$ such that

$$\boldsymbol{u}'[n, t] = \begin{cases} \boldsymbol{u}[n, t] & \text{if } n < N, \ t < d \\ 0 & \text{otherwise.} \end{cases}$$

The output from the last layer $\boldsymbol{z} \in \mathbb{R}^{N' \times d'}$ is transformed into output $\boldsymbol{y} \in R^{N \times d}$ by extracting the top left $N \times d$ entries in $\boldsymbol{z}$.

Next, we define the class of weight matrices that we will use in the linear projections in the models:

**Definition H.3.** The linear projection $\texttt{Linear}_{m,m}$ has its matrix representation as the weight matrix[10] $\mathbf{W} \in \mathbb{R}^{m \times m}$ taken to be a **K-matrix** for $\mathbf{W} \in (\mathcal{BB}^*)_{\text{poly-}\log m}^{\text{poly-}\log m}$ (Definition H.17). Consequently, each matrix $\mathbf{W}$ has $\tilde{\mathcal{O}}(m)$ parameters and runtime for matrix vector multiplication, and allows us to represent general linear transformations with low-depth linear arithmetic circuits (Dao et al., 2020). We will also need linear maps $\texttt{Linear}_{m,n}$ for $m < n$, where each take the corresponding square matrices from $\texttt{Linear}_{m,m}$ and note that such matrices has $\tilde{\mathcal{O}}(n)$ parameters and runtime for matrix vector multiplication.

**Remark H.4.** We note that the weight matrix $\mathbf{W} \in \mathbb{R}^{d \times d}$ above is taken to be a dense matrix in the experiments. However, for our theoretical results, we restrict the linear projection in our models as per Definition H.3. In the rest of the paper, unless mentioned otherwise all linear projections will follow Definition H.3.

Next, we define three popular gated convolution models that we study in our work: Hyena Poli et al. (2023a), RWKV Peng et al. (2023), and RetNet Sun et al. (2023).

### H.2.1 THE HYENA LAYER

We will now outline the Hyena layer (Poli et al., 2023b). Hyena takes a sequence $\boldsymbol{u} \in \mathbb{R}^{N \times d}$ as input and produces $L + 1$ projections $p^1, \ldots, p^L, v$ by passing $\boldsymbol{y}$ though a linear layer and applying a short convolution afterwards. The algorithm then recursively performs a point-wise multiplication of the projection with the convolution of the filter $h^l$ with the previous output. We summarize this process in Algorithm 3.

---

**Algorithm 2** $\texttt{Projection}(\boldsymbol{u}, \boldsymbol{h})$

---

**Input:** Input sequence $\boldsymbol{u} \in \mathbb{R}^{N \times d}$, a short convolution filter $\boldsymbol{h} \in \mathbb{R}^N$.
1: In parallel for $0 \leq n < N : \hat{\boldsymbol{z}}[n, :] \leftarrow \texttt{Linear}_{d,(L+1)d}(\boldsymbol{u}[n, :])$ so that $\hat{\boldsymbol{z}} \in \mathbb{R}^{N \times (L+1)d}$
2: In parallel for $0 \leq t < (L + 1)d : \boldsymbol{z}[:, t] \leftarrow \boldsymbol{h} * \hat{\boldsymbol{z}}[:, t]$
3: Reshape and split $\boldsymbol{z} \in \mathbb{R}^{N \times (L+1)d}$ into $\boldsymbol{p}^1, \ldots, \boldsymbol{p}^L, \boldsymbol{v}$, where $\boldsymbol{p}^\ell, \boldsymbol{v} \in \mathbb{R}^{N \times d}$ for $\ell \in [L]$.
4: **return** $\boldsymbol{p}^1, \ldots, \boldsymbol{p}^L, \boldsymbol{v}$.

---

**Algorithm 3** $\texttt{Hyena}(\boldsymbol{u}, \boldsymbol{h}, \boldsymbol{h}_s)$

---

**Input:** Input sequence $\boldsymbol{u} \in \mathbb{R}^{N \times d}$, set of convolution filters $\boldsymbol{h}^1, \ldots, \boldsymbol{h}^L \in \mathbb{R}^{N \times d}$, short convolution filter $\boldsymbol{h}_s \in \mathbb{R}^N$.
1: $\boldsymbol{p}^1, \ldots, \boldsymbol{p}^L, \boldsymbol{v} \leftarrow \texttt{Projection}(\boldsymbol{u}, \boldsymbol{h}_s)$.
2: $\boldsymbol{z}^0 \leftarrow \boldsymbol{v}$
3: **for** $\ell = 1, \ldots, L$ **do**
4: In parallel for $0 \leq t < d : \boldsymbol{z}^\ell[:, t] \leftarrow \boldsymbol{p}^\ell[t, :] \odot (\boldsymbol{h}^\ell[:, t] * \boldsymbol{z}^{\ell-1}[:, t])$.
5: **return** $\boldsymbol{z}^L$

---

[10]I.e. $\texttt{Linear}_{m,m}(\boldsymbol{z}) = \boldsymbol{W} \cdot \boldsymbol{z}$.

**Remark H.5.** In Algorithm 3, $L$ is used to denute the nuber of recursive applications of the Hadamard product and convolutions, not the number of layers Note that asymptotically, the recursive step does not make a difference.

Henceforth, we will refer to a model consisting of $L$ Hyena layers is a gated convolution model with associated tuple $(N, L, d, N, (L + 1)d)$ as $(N, L, d, N, (L + 1)d) - \text{Hyena}$.

### H.2.2 THE RWKV LAYER

We will now describe the RWKV layer (Peng et al., 2023). RWKV is typically referred to as an RNN, but, like some other recurrent models (*e.g.* S4 (Gu et al., 2021)), it can also be viewed as a convolution. Here, we present the convolutional view of RWKV. We will see that it is closely related to the Hyena layer.

RWKV takes a sequence $\boldsymbol{u} \in \mathbb{R}^{N \times d}$ as input and applies a short convolution. Next, it produces three projections $q, k, v$ by passing $\boldsymbol{u}$ through a linear layer. Then, it performs a convolution sandwiched by element-wise multiplication. We summarize this process in Algorithm 5.

---

**Algorithm 4** `RWKVProjection` $(\boldsymbol{u}, \boldsymbol{h})$

---

**Input:** Input sequence $\boldsymbol{u} \in \mathbb{R}^{N \times d}$, a short convolution filter $\boldsymbol{h} \in \mathbb{R}^N$.
 1: In parallel for $0 \leq t < d : \hat{\boldsymbol{z}}[:, t] \leftarrow \boldsymbol{h} * \boldsymbol{u}[:, t]$
 2: In parallel for $0 \leq n < N : \boldsymbol{z}[n, :] \leftarrow \text{Linear}_{d,3d} (\hat{\boldsymbol{z}}[n, :])$ so that $\boldsymbol{z} \in \mathbb{R}^{N \times 3d}$
 3: Reshape and split $\boldsymbol{z} \in \mathbb{R}^{N \times 3d}$ into $\boldsymbol{q}, \boldsymbol{k}, \boldsymbol{v} \in \mathbb{R}^{N \times d}$.
 4: **return** $\boldsymbol{q}, \boldsymbol{k}, \boldsymbol{v}$.

---

In practice, the short convolution filter $\boldsymbol{h}_s$ is restricted to a length two filter with $\boldsymbol{h}_s[0] = \mu$ and $\boldsymbol{h}_s[1] = 1 - \mu$, where $\mu \in [0, 1]$ is learned parameter. In the RWKV paper, the short convolution is referred to as a "time shift".

---

**Algorithm 5** `RWKV` $(\boldsymbol{u}, \boldsymbol{h}, \boldsymbol{h}_s)$

---

**Input:** Input sequence $\boldsymbol{u} \in \mathbb{R}^{N \times d}$, set of convolution filters $\boldsymbol{h}^1, \ldots, \boldsymbol{h}^L \in \mathbb{R}^{N \times d}$, short convolution filter $\boldsymbol{h}_s \in \mathbb{R}^N$.
 1: $\boldsymbol{q}, \boldsymbol{k}, \boldsymbol{v} \leftarrow \text{Projection}(\boldsymbol{u}, \boldsymbol{h}_s)$.
 2: In parallel for $0 \leq t < d : \boldsymbol{z}^\ell[:, t] \leftarrow \sigma(\boldsymbol{q}[:, t]) \odot \left(\boldsymbol{h}^\ell[:, t] * (\text{softmax}(\boldsymbol{k}) \odot \boldsymbol{v})[:, t]\right)$.
 3: **return** $\boldsymbol{z}^L$

---

In practice, the long convolution filter $\boldsymbol{h}$ is also restricted to $\boldsymbol{h}[:, t] = e^{w(t-1)}$, where $w \in \mathbb{R}$ is a learnable parameter that controls how quickly the magnitudes of the filter decreases as $t$ grows.

To summarize, the differences between Hyena and RWKV are: (1) Hyena applies the short convolution *after* the linear projection whereas RWKV applies it *before*, (2) RWKV includes non-linear activations (sigmoid and softmax) while Hyena does not, (3) RWKV and Hyena use different parameterizations for the convolutional filters, and (4) Hyena recursively performs a point-wise multiplication of the projections with the convolution filter whereas RWKV performs this operation only once (though, in practice, Hyena uses a recurrence depth of one, making them equivalent in this regard).

### H.2.3 THE RETNET LAYER

In this section, we introduce the RetNet layer Sun et al. (2023). To this end, we take an input sequence $\boldsymbol{u} \in \mathbb{R}^{N \times d}$ and project it using learnable weight matrices. We then compute the states $\boldsymbol{z}^n$ recurrently as in line 5 and the output $\text{Out}[n, :]$ as in line 6 for each $n \in [N]$ (see Algorithm 6 below).

---

**Algorithm 6** RetNet $(\boldsymbol{u}, \boldsymbol{h}, \boldsymbol{h}_s)$

---

**Input:** Input sequence $\boldsymbol{u} \in \mathbb{R}^{N \times d}$, a scalar $\gamma \in \mathbb{R}$, learnable weight matrices $\mathbf{W}_A, \mathbf{W}_C, \mathbf{W}_V \in \mathbb{R}^{d \times d}$.
1: $\mathbf{A}, \mathbf{C}, \mathbf{V} \leftarrow \boldsymbol{u}\mathbf{W}_A, \boldsymbol{u}\mathbf{W}_C, \boldsymbol{u}\mathbf{W}_V$ so that $\mathbf{A}, \mathbf{C}, \mathbf{V} \in \mathbb{R}^{N \times d}$.
2: Initialize the output $\texttt{Out} \in \mathbb{R}^{N \times d}$
3: Initialize the state $\boldsymbol{z}^0 \leftarrow (\mathbf{A}[0,:])^\top \mathbf{V}[0,:]$.
4: **for** $1 \leq n < N$ **do**
5: $\quad \boldsymbol{z}^n \leftarrow \gamma \boldsymbol{z}^{n-1} + (\mathbf{A}[n,:])^\top \mathbf{V}[n,:]$ for $\boldsymbol{z}^n \in \mathbb{R}^{d \times d}$
6: $\quad \texttt{Out}[n,:] \leftarrow \mathbf{C}[n,:]\boldsymbol{z}^n$
7: **return** $\texttt{Out}$

---

### H.2.4 BASECONV

Finally, we introduce the BASECONV here as follows:

$$\mathbf{Y} = (\boldsymbol{u}\mathbf{W} + \boldsymbol{b}_1) \odot (\boldsymbol{u} * \boldsymbol{h} + \boldsymbol{b}_2), \tag{4}$$

with input $\boldsymbol{u} \in \mathbb{R}^{N' \times d'}$, weight matrix $\mathbf{W} \in \mathbb{R}^{d' \times d'}$ and bias matrices $\boldsymbol{b}_i \in \mathbb{R}^{N' \times d'}$ defining linear projections of the input sequence, and $\boldsymbol{h} \in \mathbb{R}^{N' \times d'}$ is the a set of the $d'$ mixed length filters. The corresponding pseudocode for BASECONV is as follows:

---

**Algorithm 7** BASECONV $(\boldsymbol{u}, \boldsymbol{W}, \mathbf{b}_1, \boldsymbol{h}, \mathbf{b}_2)$

---

**Input:** Input sequence $\boldsymbol{u} \in \mathbb{R}^{N' \times d'}$, linear mapping $\boldsymbol{W} \in \mathbb{R}^{d' \times d'}$ so that $\boldsymbol{W} \in (\mathcal{B}\mathcal{B}^*)^{\text{poly} \log d'}_{\text{poly} \log d'}$, convolution filter $\boldsymbol{h} \in \mathbb{R}^{N' \times d'}$, bias matrices $\mathbf{b}_1, \mathbf{b}_2 \in \mathbb{R}^{N' \times d'}$.
1: In parallel for $0 \leq n < N' : \boldsymbol{x}[n,:] = \texttt{Linear}_{d',d'}(\boldsymbol{u}[n,:])$
2: In parallel for $0 \leq t < d' : \boldsymbol{z}[:,t] = \boldsymbol{h}[:,t] * \boldsymbol{u}[:,t]$

3: In parallel for $0 \leq t < d' : \boldsymbol{y}[:,t] \leftarrow (\boldsymbol{x}[:,t] + \boldsymbol{b}_1[:,t]) \odot (\boldsymbol{z}[:,t] + \boldsymbol{b}_2[:,t])$. $\quad \triangleright$ See equation 4
4: **return** $\boldsymbol{y}$

---

Note that the convolution in Algorithm 7 is not limited to causal convolution and allows circular convolution as well. For simplicity, we use $*$ here and will disambiguate when required. We will start by specifying the parameter count and complexity of a single layer of BASECONV below.

**Proposition H.6.** *A single layer of* BASECONV *requires* $\tilde{O}(N'd')$ *parameters and runtime.*

*Proof.* We know the parameters and runtime for the linear part of BASECONV via Definition H.3 is $\tilde{\mathcal{O}}(N'd')$. Further, each convolution operation requires $\mathcal{O}(N')$ parameters and $\mathcal{O}(N' \log N')$ runtime, and we employ $d'$ such operations. Finally the Hadamard product step takes $O(nd)$ time. $\quad \square$

Similar to Hyena, we will refer to a model consisting of $L$ BASECONV layers as $(N, L, d, N', d') -$ BASECONV. In our experiments, we extend BASECONV by adding an MLP after Algorithm 7. For simplicity we will denote BASECONV $(\boldsymbol{u}, \boldsymbol{h}, \boldsymbol{b}_1, \boldsymbol{b}_2)$ as BASECONV $(\boldsymbol{u}, \boldsymbol{h})$ when $\boldsymbol{b}_1 = \boldsymbol{b}_2 = \boldsymbol{0}$.
We will now show that there exists a BASECONV model that can emulate each of the basic operations in Algorithm 7.

**Lemma H.7.** *The functions* $\texttt{Linear}_{d,d}(\boldsymbol{u})$ *(Definition H.3), with* $d, d'$ *defined as in Algorithm 7, convolution with filter* $\boldsymbol{h} \in \mathbb{R}^{N \times d}$, *and element-wise gating can be computed with Algorithm 7 via a* $(N, 1, d, N, d') -$ BASECONV.

*Proof.* For each operation from Definition H.2 and Algorithm 7:
1. For any input $\boldsymbol{u} \in \mathbb{R}^{N' \times d'}$, $\texttt{Linear}_{d,d'}(\boldsymbol{u})$ with matrix representation $\boldsymbol{W} \in \mathbb{R}^{N' \times d'}$ can be performed by a single BASECONV layer computing BASECONV $(\boldsymbol{y}, \boldsymbol{W}, \boldsymbol{h}, \boldsymbol{b}_1, \boldsymbol{b}_2)$ with $\boldsymbol{b}_1$ and $\boldsymbol{b}_2$ being the matrix of all 0s and all 1s, respectively while and the convolution with the zero filter. That is, we have

$$\mathbf{Y} = (\boldsymbol{u}\mathbf{W} + \boldsymbol{0}^{N' \times d'}) \odot (\boldsymbol{u} * \boldsymbol{0}^{N' \times d'} + \boldsymbol{1}^{N' \times d'}) = (\boldsymbol{u}\mathbf{W}) \odot \boldsymbol{1}^{N' \times d'} = \boldsymbol{u}\mathbf{W} = \texttt{Linear}_{d,d'}(\boldsymbol{u}).$$

2. For any input $\boldsymbol{u} \in \mathbb{R}^{N \times d}$, convolution with filter $\boldsymbol{h} \in \mathbb{R}^{N \times d}$ can be performed by a single BASECONV layer computing BASECONV $(\boldsymbol{y}, \boldsymbol{W}, \boldsymbol{h}, \boldsymbol{b}_1, \boldsymbol{b}_2)$ where $\boldsymbol{W}, \boldsymbol{b}_2$ are all zeroes, and $\boldsymbol{b}_1$ is the matrix of all 1s so that we get

$$\mathbf{Y} = (\boldsymbol{u}\mathbf{0}^{N' \times d'} + \mathbf{1}^{N' \times d'}) \odot (\boldsymbol{u} * \boldsymbol{h} + \mathbf{0}^{N' \times d'}) = \mathbf{1}^{N' \times d'} \odot (\boldsymbol{u} * \boldsymbol{h}) = \boldsymbol{u} * \boldsymbol{h}.$$

3. We may compute element-wise gating between matrices $\boldsymbol{u}, \boldsymbol{v} \in \mathbb{R}^{N \times d}$, where $\boldsymbol{v}$ is some fixed factor, with a single layer computing BASECONV $(\boldsymbol{y}, ,) \mathbf{0}^{N' \times d'}, \boldsymbol{e}_0, \boldsymbol{v}, \mathbf{0}^{N' \times d'}$ where $\boldsymbol{e}_1$ is the identity filter, respectively, by Definition H.2.

$$\mathbf{Y} = (\boldsymbol{u}\mathbf{0}^{N' \times d'} + \boldsymbol{v}) \odot (\boldsymbol{u} * \boldsymbol{e}_0 + \mathbf{0}^{N' \times d'}) = \boldsymbol{v} \odot \boldsymbol{u}.$$

$\square$

## H.3 PRIMITIVES

In this section, we will establish some additional basic primitives that we expect a gated convolution model to emulate: `shift`, `remember` and `add`. We specify them below:

1. **Shift** an sequential input of length $N$ up or down by $s$ entries:

```
shift_up(y,s), shift_down(y,s)
```
- **Input:** $\boldsymbol{y} \in \mathbb{R}^{N \times d}$, $s \geq 0$
- **Output:** $\boldsymbol{z} \in \mathbb{R}^{N \times d}$ where $\boldsymbol{z}^+ = \texttt{shift\_down}(\boldsymbol{y}, s)$ and $\boldsymbol{z}^- = \texttt{shift\_up}(\boldsymbol{y}, s)$

2. **Add** a sequence $\boldsymbol{x} \in \mathbb{R}^{n \times d}$ to a running sum $\boldsymbol{S} \in \mathbb{R}^{n \times d}$ for some $2n \leq N'$ with both $\boldsymbol{x}$ and $\boldsymbol{S}$ contained as subvectors in $\boldsymbol{y} \in \mathbb{R}^{N \times d}$.

```
add_n(y : x, S):
```
- **Input:** sequence $\boldsymbol{y}$ containing $\boldsymbol{x}, \boldsymbol{S} \in \mathbb{R}^{n \times d}$ for $2n \leq N$ such that $\boldsymbol{y}[0 : n-1] \equiv \boldsymbol{x}, \boldsymbol{y}[n : 2n-1] \equiv \boldsymbol{S}$ and $\boldsymbol{y}[2n : N-1] \equiv \mathbf{0}^{N-2n}$.
- **Output:** $\boldsymbol{z} \in \mathbb{R}^{N \times d}$ containing the sum $\boldsymbol{y} + \boldsymbol{S}$ such that $\boldsymbol{y}[0 : n-1] \equiv \mathbf{1}^n, \boldsymbol{y}[n : 2n-1] \equiv \boldsymbol{S} + \boldsymbol{x}$ and $\boldsymbol{z}[2n : N-1] \equiv \mathbf{0}^{N-2n}$.

3. **Remember** $v \in \mathbb{R}^{m \times d}$ as part of a sequence of input $y \in \mathbb{R}^{N \times d}$ while performing gated convolution *only* on $x \in \mathbb{R}^{n \times d}$ for some $m, n \leq N \times d$.

These primitives are building blocks of our proofs in the sequel. We will show that each of these primitives can be solved by some $(N, L, d, N', d') - \text{BASECONV}$ model with a small constant $L$.

**Proposition H.8** (The Shift Primitive). *For any $y \in \mathbb{R}^{N \times d}$, there exist $(N, 1, d, N, d) - \text{BASECONV}$ and $(N, 3, d, N, d) - \text{BASECONV}$ that computes* $\mathtt{shift\_down}(y, s)$ *and* $\mathtt{shift\_up}(y, s)$ *for any $s \leq N$.*

*Proof.* Define the following kernel dependent on $s \leq N$

$$h_s[n, :] \equiv \begin{cases} \mathbf{1}^d & \text{if } n = s + 1 \\ \mathbf{0}^d & \text{otherwise.} \end{cases}$$

We now deal with the down and up shifts separately:

1. We define $\mathbf{W} := \mathbf{0}^{N \times d}, b_1 := \mathbf{1}^{N \times d}, b_2 := \mathbf{0}^{N \times d}$. Then, for input $y \in \mathbb{R}^{N \times d}$, $\text{BASECONV}\left(y, \mathbf{0}^{N \times d}, h_s, \mathbf{1}^{N \times d}, \mathbf{0}^{N \times d}\right)$ for BASECONV in Algorithm 7 is given by equation 4 as

$$\mathbf{Y} \equiv y * h_s.$$

Now, to perform $\mathtt{shift\_down}(y, s)$, we note that

$$\mathbf{Y}[:, j] = y[:, j] * h_s[:, j] = \text{coeff}(y[:, j](X) \cdot h_s[:, j](X))$$

$$= \text{coeff}\left(\left(\sum_{i=0}^{N-1} y[i, j] \cdot X^i\right) \cdot X^s \mod X^N\right)$$

$$= \text{coeff}\left(\sum_{i=0}^{N-1} y[i, j] \cdot X^{i+s} \mod X^N\right)$$

$$= \text{coeff}\left(\sum_{i=s}^{N-1+s} y[i-s, j] \cdot X^i \mod X^N\right)$$

$$= \text{coeff}\left(\sum_{i=s}^{N-1} y[i-s, j] \cdot X^i\right),$$

which implies that we exactly get what is specified in the output.

2. We again define $\mathbf{W} := \mathbf{0}^{N \times d}, b_1 := \mathbf{1}^{N \times d}, b_2 := \mathbf{0}^{N \times d}$. Then, for input $y \in \mathbb{R}^{N \times d}$, $\text{BASECONV}\left(y, \mathbf{0}^{N \times d}, e_0, \mathbf{1}^{N \times d}, \mathbf{0}^{N \times d}\right)$ for BASECONV in Algorithm 7 is given in equation 4 as

$$\mathbf{Y}_0 \equiv y \circledast e_0.$$

Now, to perform $\texttt{shift\_up}(\boldsymbol{y}, s)$, as before, we first apply the circular convolution to reverse the input

$$\mathbf{Y}_0[:,j] = \boldsymbol{y}[:,j] \circledast \boldsymbol{e}_0 = \text{coeff}\left(\sum_{i=0}^{N-1} \boldsymbol{y}[N-1-i,j] \cdot X^i\right),$$

We then apply $\boldsymbol{Y}_1 \equiv \texttt{shift\_down}(\boldsymbol{Y}_0, s)$ to get

$$\boldsymbol{Y}_1[:,j] \equiv \text{coeff}\left(\sum_{i=s}^{N-1} \boldsymbol{Y}_0[N-1-(i-s),j] \cdot X^i\right),$$

$$\equiv \text{coeff}\left(\sum_{i=s}^{N-1} \boldsymbol{Y}_0[N-1-i+s,j] \cdot X^i\right).$$

Finally, we apply another circular convolution with the identity filter to replace $N-1-i$ with $i$ to get

$$\mathbf{Y}_2[:,j] = \boldsymbol{Y}_1[:,j] \circledast \boldsymbol{e}_0 = \text{coeff}\left(\sum_{i=0}^{N-1} \boldsymbol{y}[i+s,j] \cdot X^i\right),$$

Here, we note that we can compute both of these primitives in one and three layers, respectively (see Lemma H.11).

$\square$

Now, we present a BASECONV model with two layers that implements the $\texttt{add}_n(\boldsymbol{y} : \boldsymbol{x}, \boldsymbol{S})$, the purpose of which is to add some window of computation $\boldsymbol{x}$ to a running sum $\boldsymbol{S}$.

**Proposition H.9** (The Running Sum Primitive). *For any $\boldsymbol{x}, \boldsymbol{S} \in \mathbb{R}^{n \times d}$ contained in some $\boldsymbol{y} \in \mathbb{R}^{N \times d}$, there exists a $(N, 2, d, N, d) - $ BASECONV that computes $\texttt{add}_n(\boldsymbol{y} : \boldsymbol{x}, \boldsymbol{S})$ for BASECONV as in Algorithm 7.*

*Proof.* We will show this for $d' = 1$ and the general case follows as we will explain at the end. We now specify the two layers that we use

$$\boldsymbol{z}^1 \equiv \text{BASECONV}\left(\boldsymbol{y}, \boldsymbol{0}^{N \times 1}, \boldsymbol{h}^1, \boldsymbol{b}_1^1, \boldsymbol{0}^{N \times 1}\right) \equiv \boldsymbol{b}_1^1 \odot \left(\boldsymbol{h}^1 * \boldsymbol{y}\right)$$

$$\boldsymbol{z} \equiv \text{BASECONV}\left(\boldsymbol{z}^1, \boldsymbol{0}^{N \times 1}, \boldsymbol{h}^2, \boldsymbol{b}_1^2, \boldsymbol{b}_1^2\right) \equiv \boldsymbol{b}_1^2 \odot \left(\boldsymbol{h}^2 * \boldsymbol{y} + \boldsymbol{b}_2^2\right),$$

where we will specify the kernels as we go along. Let us start by defining the kernel and the bias for the first layer as

$$\boldsymbol{h}^1 \equiv \begin{pmatrix} \boldsymbol{e}_0 \\ \boldsymbol{e}_0 \\ \boldsymbol{0}^n \\ \dots \\ \boldsymbol{0}^n \end{pmatrix}, \qquad \boldsymbol{b}_1 \equiv \begin{pmatrix} \boldsymbol{0}^n \\ \boldsymbol{1}^n \\ \boldsymbol{0}^n \\ \dots \\ \boldsymbol{0}^n \end{pmatrix}.$$

Let us first compute $\boldsymbol{h}^1 * \boldsymbol{y}$ as follows:

$$\boldsymbol{h}^1(X) \cdot \boldsymbol{y}(X) = (X^n + 1) \cdot \left(\boldsymbol{S}(X) \cdot X^n + \boldsymbol{x}(X)\right)$$

$$= \boldsymbol{S}(X) \cdot X^{2n} + (\boldsymbol{S} + \boldsymbol{x})(X) \cdot X^n + \boldsymbol{x}(X).$$

We then have

$$\boldsymbol{z}_1 \equiv \boldsymbol{b}_1^1 \odot \left(\boldsymbol{h}^1 * \boldsymbol{y}\right) \equiv \begin{pmatrix} \boldsymbol{0}^n \\ \boldsymbol{1}^n \\ \boldsymbol{0}^n \\ \dots \\ \boldsymbol{0}^n \end{pmatrix} \odot \begin{pmatrix} \boldsymbol{x} \\ \boldsymbol{S}+\boldsymbol{x} \\ \boldsymbol{S} \\ \dots \\ \boldsymbol{0}^n \end{pmatrix} \equiv \begin{pmatrix} \boldsymbol{0}^n \\ \boldsymbol{S}+\boldsymbol{x} \\ \boldsymbol{0}^n \\ \dots \\ \boldsymbol{0}^n \end{pmatrix}$$

**Resetting for Next Phase.**  We now use the next layer to reset for the next phase. Here, we need the first vector to be $\mathbf{1}^n$ in order to start adding the next vector. We thus use the kernel and the biases $\boldsymbol{h}^2, \boldsymbol{b}_1^2, \boldsymbol{b}_2^2$ defined as

$$
\boldsymbol{h}^2 \equiv \begin{pmatrix} \boldsymbol{e}_0 \\ \mathbf{0}^n \\ \mathbf{0}^n \\ \ldots \\ \mathbf{0}^n \end{pmatrix}, \qquad \boldsymbol{b}_1^2 \equiv \begin{pmatrix} \mathbf{1}^n \\ \mathbf{1}^n \\ \mathbf{0}^n \\ \ldots \\ \mathbf{0}^n \end{pmatrix}, \qquad \boldsymbol{b}_2^2 \equiv \begin{pmatrix} \mathbf{1}^n \\ \mathbf{0}^n \\ \mathbf{0}^n \\ \ldots \\ \mathbf{0}^n \end{pmatrix}.
$$

Explicitly, for the second layer, we compute the result of the convolution in terms of polynomials as follows:

$$
\boldsymbol{h}^2(X) \cdot \boldsymbol{z}^1(X) = 1 \cdot (\boldsymbol{S} + \boldsymbol{x})(X) \cdot X^n = (\boldsymbol{S} + \boldsymbol{x})(X) \cdot X^n.
$$

Thus, the output for the second layer is given by

$$
\boldsymbol{z} \equiv \boldsymbol{b}_1^2 \odot \left( \boldsymbol{h}^2 * \boldsymbol{z}^1 + \boldsymbol{b}_2^2 \right) \equiv \begin{pmatrix} \mathbf{1}^n \\ \mathbf{1}^n \\ \mathbf{0}^n \\ \ldots \\ \mathbf{0}^n \end{pmatrix} \odot \left( \begin{pmatrix} \mathbf{0}^n \\ \boldsymbol{S}+\boldsymbol{x} \\ \mathbf{0}^n \\ \ldots \\ \mathbf{0}^n \end{pmatrix} + \begin{pmatrix} \mathbf{1}^n \\ \mathbf{0}^n \\ \mathbf{0}^n \\ \ldots \\ \mathbf{0}^n \end{pmatrix} \right) \equiv \begin{pmatrix} \mathbf{1}^n \\ \mathbf{1}^n \\ \mathbf{0}^n \\ \ldots \\ \mathbf{0}^n \end{pmatrix} \odot \begin{pmatrix} \mathbf{1}^n \\ \boldsymbol{S}+\boldsymbol{x} \\ \mathbf{0}^n \\ \ldots \\ \mathbf{0}^n \end{pmatrix} \equiv \begin{pmatrix} \mathbf{1}^n \\ \boldsymbol{S}+\boldsymbol{x} \\ \mathbf{0}^n \\ \ldots \\ \mathbf{0}^n \end{pmatrix}.
$$

Therefore, we have used two BASECONV layers to add $\boldsymbol{x}$ to the running sum $\boldsymbol{S}$ and reset for the next phase. Here, we note that the only operations we perform and are convolutions and Hadamard product and they generalize in the obvious way to $d > 1$.  $\square$

Next, we show that a five layer BASECONV model can perform gated convolution on windows of the input (without changing the rest of the input).

**Proposition H.10** (The Remembering Primitive). *For any $\boldsymbol{x} \in \mathbb{R}^{n \times d}, \boldsymbol{v} \in \mathbb{R}^{m \times d}$ contained in some $\boldsymbol{y} \in \mathbb{R}^{N \times d}$ for some $n + m + s + t \leq N$ so that for $\boldsymbol{h} \in \mathbb{R}^{n \times d}$ and $\boldsymbol{p} \in \mathbb{R}^{(n+s) \times d}$ with $\boldsymbol{x} * \boldsymbol{h} \in \mathbb{R}^{(n+s) \times d}$ and $\boldsymbol{v} * \boldsymbol{h} \in \mathbb{R}^{(m+t) \times d}$, there exists a $(N, 5, d, N, d)-$BASECONV that computes* `remember`$(\boldsymbol{y} : \boldsymbol{x}, \boldsymbol{v}, \boldsymbol{h}, \boldsymbol{p})$ *for BASECONV as in Algorithm 7.*

*Proof.* We will again show this for $d' = 1$ and the general case should follow. We now specify the first two layers that we use

$$
\boldsymbol{z}^1 \equiv \text{BASECONV} \left( \boldsymbol{y}, \mathbf{0}^{N \times 1}, \boldsymbol{h}^1, \boldsymbol{b}_1^1, \mathbf{0}^{N \times d} \right) \equiv \boldsymbol{b}_1^1 \odot \left( \boldsymbol{h}^1 * \boldsymbol{y} \right)
$$
$$
\boldsymbol{z}^2 \equiv \text{BASECONV} \left( \boldsymbol{z}^1, \mathbf{0}^{N \times 1}, \boldsymbol{h}^2, \boldsymbol{b}_1^2, \mathbf{0}^{N \times d} \right) \equiv \boldsymbol{b}_1^2 \odot \left( \boldsymbol{h}^2 * \boldsymbol{y} \right),
$$

The kernel $\boldsymbol{h}^1$ and the bias $\boldsymbol{b}_1^1$ for the first layer are then given by

$$
\boldsymbol{h}^1 \equiv \begin{pmatrix} \boldsymbol{h} \\ \mathbf{0}^m \\ \boldsymbol{e}_{s+t} \\ \mathbf{0}^n \\ \mathbf{0}^n \\ \ldots \\ \mathbf{0}^n \end{pmatrix}, \qquad \boldsymbol{b}_1^1 \equiv \begin{pmatrix} \boldsymbol{p} \\ \mathbf{0}^{m+t} \\ \mathbf{0}^n \\ \mathbf{0}^s \\ \mathbf{1}^m \\ \ldots \\ \mathbf{0}^n \end{pmatrix}.
$$

where recall that $\boldsymbol{x} * \boldsymbol{h} \in \mathbb{R}^{(n+s)\times d}$ and $\boldsymbol{v} * \boldsymbol{h} \in \mathbb{R}^{(m+t)\times d}$.

We now want to first specify the result of applying the first kernel:

$$
\begin{aligned}
\left(\boldsymbol{h}^1 * \boldsymbol{y}\right) &= \operatorname{coeff}\left(\left(\boldsymbol{h}(X) + X^{n+m+s+t}\right) \cdot \left(\boldsymbol{v}(X) \cdot X^{n+s} + \boldsymbol{x}(X)\right)\right) \\
&= \operatorname{coeff}\left(\boldsymbol{h} * \boldsymbol{v}(X) \cdot X^{n+s} + \boldsymbol{h} * \boldsymbol{x}(X) + \boldsymbol{v}(X) \cdot X^{2n+2s+m+t} + \boldsymbol{x}(X) \cdot X^{n+m+s+t}\right)
\end{aligned}
$$

We then have

$$
\boldsymbol{z}_1 \equiv \boldsymbol{b}_1^1 \odot \left(\boldsymbol{h}^1 * \boldsymbol{y}\right) \equiv
\begin{pmatrix}
\boldsymbol{p} \\
\mathbf{0}^{m+t} \\
\mathbf{0}^n \\
\mathbf{0}^s \\
\mathbf{1}^m \\
\dots \\
\mathbf{0}^n
\end{pmatrix}
\odot
\begin{pmatrix}
\boldsymbol{h} * \boldsymbol{x} \\
\boldsymbol{h} * \boldsymbol{v} \\
\boldsymbol{x} \\
\mathbf{0}^s \\
\boldsymbol{v} \\
\dots \\
\mathbf{0}^n
\end{pmatrix}
\equiv
\begin{pmatrix}
\boldsymbol{p} \odot (\boldsymbol{h} * \boldsymbol{x}) \\
\mathbf{0}^{m+t} \\
\mathbf{0}^n \\
\mathbf{0}^s \\
\boldsymbol{v} \\
\dots \\
\mathbf{0}^n
\end{pmatrix}.
$$

We now describe the second kernel $\boldsymbol{h}^2$ and the bias matrix $\boldsymbol{b}_1^2$ as follows:

$$
\boldsymbol{h}^2 \equiv
\begin{pmatrix}
\boldsymbol{e}_0 \\
\mathbf{0}^{m+t} \\
\boldsymbol{e}^0 \\
\mathbf{0}^{n+s} \\
\mathbf{0}^m \\
\dots \\
\mathbf{0}
\end{pmatrix},
\qquad
\boldsymbol{b}_1^2 \equiv
\begin{pmatrix}
\mathbf{0}^{n+s} \\
\mathbf{0}^{m+t} \\
\mathbf{0}^n \\
\mathbf{1}^{n+s} \\
\mathbf{1}^m \\
\dots \\
\mathbf{0}
\end{pmatrix}
$$

This yields the following convolution computation:

$$
\begin{aligned}
\boldsymbol{h}^2 \odot \boldsymbol{z}^1 &\equiv \operatorname{coeff}\left(\left(X^{m+n+s+t} + 1\right) \cdot \left(\boldsymbol{v}(X) \cdot X^{2n+2s+m+t} + (\boldsymbol{p} \odot (\boldsymbol{h} * \boldsymbol{x}))(X)\right)\right) \\
&\equiv \operatorname{coeff}(\boldsymbol{v}(X) \cdot X^{3n+3s+2m+2t} + \boldsymbol{v}(X) \cdot X^{2n+2s+m+t} \\
&\quad + (\boldsymbol{p} \odot (\boldsymbol{h} * \boldsymbol{x}))(X) \cdot X^{m+n+s+t} + (\boldsymbol{p} \odot (\boldsymbol{h} * \boldsymbol{x}))(X))
\end{aligned}
$$

Thus we have

$$
\boldsymbol{z}^2 \equiv \boldsymbol{b}_2^1 \odot \left(\boldsymbol{h}^2 * \boldsymbol{z}^1\right) \equiv
\begin{pmatrix}
\mathbf{0}^{n+s} \\
\mathbf{0}^{m+t} \\
\mathbf{0}^n \\
\mathbf{1}^{n+s} \\
\mathbf{1}^m \\
\dots \\
\mathbf{0}
\end{pmatrix}
\odot
\begin{pmatrix}
\boldsymbol{p} \odot (\boldsymbol{h} * \boldsymbol{x}) \\
\mathbf{0}^{m+t} \\
\mathbf{0}^n \\
\boldsymbol{p} \odot (\boldsymbol{h} * \boldsymbol{x}) \\
\boldsymbol{v} \\
\dots \\
\mathbf{0}
\end{pmatrix}
\equiv
\begin{pmatrix}
\mathbf{0}^{n+s} \\
\mathbf{0}^{m+t} \\
\mathbf{0}^n \\
\boldsymbol{p} \odot (\boldsymbol{h} * \boldsymbol{x}) \\
\boldsymbol{v} \\
\dots \\
\mathbf{0}
\end{pmatrix}
$$

We now shift this up by $2n+s+m+t$ entries using the primitive operation defined in Proposition H.8 that costs three additional layers so that we end up with

$$
\boldsymbol{z} \equiv \begin{pmatrix} \boldsymbol{p} \odot (\boldsymbol{h} * \boldsymbol{u}) \\ \boldsymbol{v} \\ \ldots \\ \boldsymbol{0} \end{pmatrix}
$$

Again, we note that the only operations we perform and are convolutions and Hadamard product and they generalize in the obvious way to $d > 1$. $\qquad\square$

Finally, we show that these primitives may be composed by 'stacking' models with matching inner dimension $(N', d')$.

**Lemma H.11.** *For $f, g : \mathbb{R}^{N \times d} \to \mathbb{R}^{N \times d}$ that have $(N, L_1, d, N', d')$ and $(N, L_2, d, N', d')$ BASECONV models then their composition $f \circ g$ has an $(N, L_1 + L_2, d, N', d')$ BASECONV model which can be computed by performing their models in succession, or 'stacking'.*

*Proof.* This result follows from noting that for any $f(\boldsymbol{u})$ which requires $L_1$ layers to compute and that we can compute $f \circ g(\boldsymbol{u}) = g(f(\boldsymbol{u}))$ using the BASECONV model with $L_2$ layers, yielding $L_1 + L_2$ layers in total. $\qquad\square$

### H.3.1 BASECONV-HYENA EQUIVALENCE

We show that the equivalence between BASECONV and Hyena by showing that each layer can simulate the other's computation using a constant number of layers.

**Proposition H.12.** *For any input $\boldsymbol{u} \in \mathbb{R}^{N \times d}$ and $(N, L, d, N', d)-$Hyena such that $\boldsymbol{z}_{\text{Hyena}} \equiv \text{Hyena}(\boldsymbol{u})$ with a set of filters $\boldsymbol{h}^\ell$ and linear projections $\boldsymbol{p}^\ell$ as per Definition H.3 for $\ell \in [L]$, there exists a $(N, 5L, d, N' + N, d)$-BASECONVmodel such that $\boldsymbol{z}_{\text{Hyena}} \equiv \text{BASECONV}(\boldsymbol{u})$.*

*Similarly, for any input $\boldsymbol{u}_{\text{BASECONV}} \in \mathbb{R}^{N \times d}$ and $(N, L, d, N', d)-$Coyote such that $\boldsymbol{z}_{\text{BASECONV}} \equiv \text{BASECONV}(\boldsymbol{u})$ with a set of filters $\boldsymbol{h}^\ell$ for $\ell \in [L]$, there exists a series of Hyena layers such that we have*

$$
\underbrace{\text{Hyena}\left(\text{Hyena}\left(\ldots \text{Hyena}(\boldsymbol{u}_{\text{BASECONV},}\boldsymbol{h})\right)\right)}_{L \text{ layers}} \equiv \boldsymbol{z}_{\text{BASECONV}}.
$$

*Proof.* For the input $\boldsymbol{u}_{\text{Hyena}} \in \mathbb{R}^{N \times d}$, the output of the $\ell$th layer $\boldsymbol{z}_{\text{Hyena}}^\ell \in \mathbb{R}^{N' \times d'}$ for Hyena is given by (see Algorithm 3)

$$
\boldsymbol{z}_{\text{Hyena}}^\ell \equiv \boldsymbol{p}_{\text{Hyena}}^\ell \odot (\boldsymbol{h}^l * \boldsymbol{z}^{\ell-1}),
$$

where $\boldsymbol{p}_{\text{Hyena}}^\ell \equiv \text{Linear}(\boldsymbol{u}_{\text{Hyena}}) \in \mathbb{R}^{N' \times d}$. Now, using the original input $\boldsymbol{u}_{\text{Hyena}} \in \mathbb{R}^{N \times d}$ to Hyena, we define the following input for BASECONVusing one layer:

$$
\boldsymbol{u}_{\text{BASECONV}} \equiv \begin{pmatrix} \boldsymbol{u}_{\text{Hyena}} \\ \boldsymbol{0}^{(N'-N) \times d} \\ \boldsymbol{u}_{\text{Hyena}} \end{pmatrix}
$$

Then, we simply use the $\text{remember}_{N,N,N'-N,N'-N}(\boldsymbol{u}_{\text{BASECONV}} : \boldsymbol{u}_{\text{Hyena}}, \boldsymbol{u}_{\text{Hyena}}, \boldsymbol{h}_{\text{Hyena}}^\ell, \boldsymbol{p}_{\text{Hyena}}^\ell)$ primitive for BASECONV. Consequently, this allows us to "remember" the input $\boldsymbol{u}_{\text{Hyena}}$ in the output of the previous BASECONVlayer $\boldsymbol{z}_{\text{BASECONV}}^{\ell-1}$. We then use this to retrieve $\boldsymbol{p}_{\text{Hyena}}^\ell \equiv \text{linear}(\boldsymbol{u}_{\text{Hyena}})$ with the projection used for BASECONV given by

$$
p_{\text{BASECONV}}^\ell \equiv \text{Linear}(\boldsymbol{z}_{\text{BASECONV}}^{\ell-1}) \equiv \begin{pmatrix} \boldsymbol{1}^{N \times d} \\ \boldsymbol{p}_{\text{Hyena}}^\ell. \end{pmatrix}
$$

Overall, the output of the $\ell$th layer for BASECONV is given by

$$\boldsymbol{z}_{\text{BASECONV}}^{\ell} \equiv \begin{pmatrix} \boldsymbol{p}_{\text{Hyena}}^{\ell} \odot \left(\boldsymbol{h}_{\text{Hyena}}^{\ell} * \boldsymbol{u}_{\text{Hyena}}\right) \\ \mathbf{0}^{(M-N) \times d} \\ \boldsymbol{u}_{\text{Hyena}} \end{pmatrix} \equiv \begin{pmatrix} \boldsymbol{z}_{\text{Hyena}}^{\ell} \\ \mathbf{0}^{(M-N) \times d} \\ \boldsymbol{u}_{\text{Hyena}} \end{pmatrix}$$

Hence, we can reproduce the output of the $\ell$th layer of Hyena using five layers of BASECONV after augmenting the input and using the remembering primitive (Proposition H.10) with internal dimension $N' + N$.

Now, for the input $\boldsymbol{u}_{\text{BASECONV}} \in \mathbb{R}^{N \times d}$, the output of the $\ell$th layer for BASECONV is given by

$$z_{\text{BASECONV}}^{\ell} \equiv \texttt{Linear}(z_{\text{BASECONV}}^{\ell-1}) \odot \text{conv}(h^l, z_{\text{BASECONV}}^{\ell-1}).$$

Here, we show inductively that simply using $\ell$-many Hyena models recursively simulates $z_{\text{BASECONV}}^{\ell}$. For $\ell = 1$, we have

$$\text{Hyena}(\boldsymbol{u}_{\text{BASECONV},\boldsymbol{h}}) \equiv \texttt{Linear}(\boldsymbol{u}_{\text{BASECONV}}) \odot (\boldsymbol{h}^1 * \boldsymbol{u}_{\text{BASECONV}}) \equiv \boldsymbol{z}_{\text{BASECONV}}^{1}.$$

We now assume that $(\ell - 1)$-many recursive Hyena models produce $z_{\text{BASECONV}}^{(\ell-1)}$. For the $\ell$th layer, we then have

$$\begin{aligned} & \text{Hyena}\left(\text{Hyena}\left(\dots \text{Hyena}(\boldsymbol{u}_{\text{BASECONV},\boldsymbol{h}})\right)\right) \\ \equiv \quad & \text{Hyena}\left(z_{\text{BASECONV}}^{(l-1)}\right) \\ \equiv \quad & \text{linear}\left(z_{\text{BASECONV}}^{(l-1)}\right) \odot \text{conv}\left(h^l, z_{\text{BASECONV}}^{(l-1)}\right) \\ \equiv \quad & z_{\text{BASECONV}}^{\ell}. \end{aligned}$$

$\square$

## H.4 LINEAR ARITHMETIC CIRCUITS

In this section we show the relation between linear arithmetic circuits and BASECONV. We recall a few definitions from (Dao et al., 2020).

**Definition H.13** (Linear Arithmetic Circuit (Bürgisser et al., 1996)). An arithmetic circuit is called a *linear arithmetic circuit* if it only uses addition, subtraction and scalar multiplication. Further, every multiplication has a fixed constant from $\mathbb{F}$ as at least one of its two inputs. In other words, all gates in the circuit are linear functions of their inputs (i.e. of the form $ax + by$ for fixed constants $a, b \in \mathbb{F}$).

**Definition H.14** (Butterfly Matrices (Dao et al., 2020)). A *butterfly factor* of size $k \geq 2$ (denoted as $\mathbf{B}_k$) is a matrix of the form $\mathbf{B}_k = \begin{bmatrix} \mathbf{D}_1 & \mathbf{D}_2 \\ \mathbf{D}_3 & \mathbf{D}_4 \end{bmatrix}$ where each $\mathbf{D}_i$ is a $\frac{k}{2} \times \frac{k}{2}$ diagonal matrix. We restrict $k$ to be a power of $2$.

A *butterfly factor matrix* of size $n$ with block size $k$ (denoted as $\mathbf{B}_k^{(n)}$) is a block diagonal matrix of $\frac{n}{k}$ (possibly different) butterfly factors of size $k$:

$$\mathbf{B}_k^{(n)} = \text{diag}\left([\mathbf{B}_k]_1, [\mathbf{B}_k]_2, \dots, [\mathbf{B}_k]_{\frac{n}{k}}\right)$$

Finally, a *butterfly matrix* of size $n$ (denoted as $\mathbf{B}^{(n)}$) is a matrix that can be expressed as a product of butterfly factor matrices: $\mathbf{B}^{(n)} = \mathbf{B}_n^{(n)} \mathbf{B}_{\frac{n}{2}}^{(n)} \dots \mathbf{B}_2^{(n)}$. Equivalently, we may define $\mathbf{B}^{(n)}$ recursively as a matrix that can be expressed in the following form:

$$\mathbf{B}^{(n)} = \mathbf{B}_n^{(n)} \begin{bmatrix} \left[\mathbf{B}^{\left(\frac{n}{2}\right)}\right]_1 & 0 \\ 0 & \left[\mathbf{B}^{\left(\frac{n}{2}\right)}\right]_2 \end{bmatrix}$$

(Note that $\left[\mathbf{B}^{\left(\frac{n}{2}\right)}\right]_1$ and $\left[\mathbf{B}^{\left(\frac{n}{2}\right)}\right]_2$ may be different.)

From Definition H.14, we observe that size $n$ butterfly factor is comprised of three vectors $\boldsymbol{d}, \boldsymbol{d}^+, \boldsymbol{d}^- \in \mathbb{R}^n$ such that

$$\boldsymbol{d} = \left(\mathrm{diag}^{-1}\left(\mathbf{D}_1\right), \mathrm{diag}^{-1}\left(\mathbf{D}_4\right)\right),$$

$$\boldsymbol{d}^+ = \left(\mathbf{0}^{\frac{n}{2}}, \mathrm{diag}^{-1}\left(\mathbf{D}_2\right)\right), \text{ and}$$

$$\boldsymbol{d}^- = \left(\mathrm{diag}^{-1}\left(\mathbf{D}_3\right), \mathbf{0}^{\frac{n}{2}}\right),$$

where $\mathrm{diag}^{-1}(\mathbf{D}) : \mathbb{R}^{n \times n} \mapsto \mathbb{R}^n$ is the mapping from diagonal matrices to the vector of its diagonal entries. Let us define $\mathbf{D_1}, \mathbf{D_2}, \mathbf{D_3} \in \mathbb{R}^{n \times n}$ as $\mathrm{diag}\left(\boldsymbol{d}\right), \mathrm{diag}\left(\boldsymbol{d}^+\right)$, and $\mathrm{diag}\left(\boldsymbol{d}^-\right)$ respectively. Then we note that

$$\mathbf{D_1} \equiv \begin{bmatrix} \mathbf{D}_1 & \mathbf{0} \\ \mathbf{0} & \mathbf{D}_4 \end{bmatrix} \qquad \mathbf{D_2}\boldsymbol{S}^{\frac{n}{2}} \equiv \begin{bmatrix} \mathbf{0} & \mathbf{D}_2 \\ \mathbf{0} & \mathbf{0} \end{bmatrix} \qquad \boldsymbol{S}^{\frac{n}{2}}\mathbf{D_3} \equiv \begin{bmatrix} \mathbf{0} & \mathbf{0} \\ \mathbf{D}_3 & \mathbf{0} \end{bmatrix} \qquad (5)$$

where $\mathbf{S}^k \in \mathbb{F}^{n \times n}$ is a shift matrix for $i \in [n/2]$. This gives us the following proposition:

**Proposition H.15.** *For any powers of 2, $n = k \geq 2$, any butterfly factor matrix $\mathbf{B}_k^{(n)}$ is equivalent to*

$$\mathbf{B}_k^{(n)} = \mathbf{S}^{\frac{k}{2}}\mathbf{D_3} + \mathbf{D_2}\mathbf{S}^{\frac{n}{2}} + \mathbf{D_1}$$

*where $\mathbf{D_3}, \mathbf{D_2}, \mathbf{D_1}, \mathbf{S}^{\frac{n}{2}}$ are defined as in equation 5.*

We use Proposition H.15 to show that butterfly matrices can easily be computed by BASECONV .

**Lemma H.16.** *For any $n, d \geq 2$, $k \geq 1$, and arbitrary vector $\boldsymbol{x} \in \mathbb{R}^{nd}$:*

    *(1) there exists a $(N, L, d, N', d') - $BASECONV that can represent $\mathbf{B}_k^{(nd)} \cdot \boldsymbol{x}$ with $N = n$, $N' = \mathcal{O}(N)$, $L = \mathcal{O}(1)$, and $d' = \mathcal{O}(d)$, and*

    *(2) there exists a $(N, L, d, N', d') - $BASECONV that can represent $\mathbf{B}^{(nd)} \cdot \boldsymbol{x}$ with $N = n$, $N' = \mathcal{O}(N)$, $L = \mathcal{O}(\log nd)$, and $d' = \mathcal{O}(d)$.*

*Proof.*     (1) Given $\boldsymbol{x} \in \mathbb{R}^{nd}$, construct $\boldsymbol{u} \in \mathbb{R}^{n \times d}$ where $\boldsymbol{x}$ is the row-major form of $\boldsymbol{u}$. We show that BASECONV can compute $\mathbf{B}_{nd} \cdot \boldsymbol{x}$ column by column.

    Let $\mathbf{A} = \mathbf{S}^{\frac{k}{2}}\mathbf{D}_3'$, $\mathbf{C} = \mathbf{D}_2'\mathbf{S}^{\frac{n}{2}}$, and $\mathbf{D} = \mathbf{D}_1$ for $\mathbf{D}_i, \mathbf{S}^{\frac{k}{2}} \in \mathbb{R}^{nd \times nd}$ for $1 \leq i \leq 3$ as defined in Proposition H.15. We take $\boldsymbol{d}_1 = \mathbf{1}^{nd}\mathbf{D}$, $\boldsymbol{d}_2 = \mathbf{1}^{nd}\mathbf{C}_2$, $\boldsymbol{d}_3 = \mathbf{1}^{nd}\mathbf{A}$, which extracts the diagonal entries of $\mathbf{D}_i$. With this we construct $\mathbf{D}_i' \in \mathbb{R}^{n \times d}$ where $\boldsymbol{d}_i$ is the row major form of $\bar{\boldsymbol{D}}_i'$. This implies that

$$\mathbf{D}_i\boldsymbol{x} \equiv \mathbf{D}_i' \odot \boldsymbol{u}.$$

    Then we can decompose $\mathbf{B}_{nd} \cdot \boldsymbol{x}$ into

$$\mathbf{B}_{nd}\boldsymbol{x} \equiv \mathbf{D_1} \odot \boldsymbol{u} + \mathbf{D_2} \odot \boldsymbol{u} + \mathbf{D_3} \odot \boldsymbol{u}.$$

    By Lemma H.7, each Hadamard product $\mathbf{A} \odot \boldsymbol{u}, \mathbf{B} \odot \boldsymbol{u}, \mathbf{C} \odot \boldsymbol{u}$ can be trivially be performed with a single layer BASECONV model. Let each of these model outputs be denoted $\boldsymbol{y_1}, \boldsymbol{y_2}, \boldsymbol{y_3}$, respectively. Finally all that remains is to compute the $\boldsymbol{y_1} + \boldsymbol{y_2} + \boldsymbol{y_3}$. We achieve this using layers of add primitives[11]:

$$\mathtt{add}_n(\boldsymbol{y}^1 : \boldsymbol{y}_1, \mathbf{0})$$
$$\mathtt{add}_n(\boldsymbol{y}^2 : \boldsymbol{y}_2, \boldsymbol{y}_1)$$
$$\mathtt{add}_n(\boldsymbol{y}^3 : \boldsymbol{y}_3, \boldsymbol{y}_1 + \boldsymbol{y}_2),$$

    where using by Proposition H.9 and Lemma H.11, this requires six more layers, and we get

$$\boldsymbol{y}^3 \equiv \boldsymbol{y_1} + \boldsymbol{y_2} + \boldsymbol{y_3} \equiv \mathbf{B}_{nd}\boldsymbol{x}.$$

    Then we can construct the $(N, L, d, N', d') - $BASECONV as desired with $L = O(1)$ layers.

    (2) From Definition H.14, $\mathbf{B}^{(nd)} = \mathbf{B}_{nd}^{(nd)}\mathbf{B}_{\frac{nd}{2}}^{(nd)} \dots \mathbf{B}_2^{(nd)}$. From (1), BASECONV can compute any butterfly matrix by simulating the $\log(nd)$ butterfly factor matrices which comprise $\mathbf{B}^{(nd)}$. With Lemma H.11, this creates a BASECONV with $5 \cdot \log(nd) = \mathcal{O}(\log(nd))$ layers. Lemma H.11

                                                               $\square$

---

[11] Recall that $\mathtt{add}_n(\boldsymbol{y} : \boldsymbol{x}, \boldsymbol{S})$ adds the subvector $\boldsymbol{x}$ to $\boldsymbol{S}$ for the input $\boldsymbol{y}$.

Butterfly matrices comprise the kaleidoscope hierarchy, which we define below:

**Definition H.17** (The Kaleidoscope Hierarchy (Dao et al., 2020))**.**

- Define $\mathcal{B}$ as the set of all matrices that can be expressed in the form $\mathbf{B}^{(n)}$ (for some $n$).
- Define $(\mathcal{B}\mathcal{B}^*)$ as the set of matrices $\mathbf{M}$ of the form $\mathbf{M} = \mathbf{M}_1\mathbf{M}_2^*$ for some $\mathbf{M_1}, \mathbf{M}_2 \in \mathcal{B}$.
- Define $(\mathcal{B}\mathcal{B}^*)^w$ as the set of matrices $\mathbf{M}$ that can be expressed as $\mathbf{M} = \mathbf{M}_w \ldots \mathbf{M}_2\mathbf{M}_1$, with each $\mathbf{M}_i \in (\mathcal{B}\mathcal{B}^*)\,(1 \le i \le w)$. (The notation $w$ represents width.)
- Define $(\mathcal{B}\mathcal{B}^*)_e^w$ as the set of $n \times n$ matrices $\mathbf{M}$ that can be expressed as $\mathbf{M} = \mathbf{S}\mathbf{E}\mathbf{S}^\top$ for some $en \times en$ matrix $\mathbf{E} \in (\mathcal{B}\mathcal{B}^*)^w$, where $\mathbf{S} \in \mathbb{F}^{n \times en} = [\mathbf{I}_n \quad 0 \quad \ldots \quad 0]]$ (i.e. $\mathbf{M}$ is the upper-left corner of $\mathbf{E}$). (The notation $e$ represents expansion relative to $n$.)

We similarly show how BASECONV can simulate any kaleidoscope matrix.

**Lemma H.18.** *Given $n, d \ge 2$, $e > 0$ for any $nd \times nd$ matrix $\mathbf{M} \in (\mathcal{B}\mathcal{B}^*)_e^w$, and $\boldsymbol{x} \in \mathbb{R}^{nd}$ there exists a $(N, L, d, N', d') - \text{BASECONV}$ that can represent $\mathbf{M} \cdot \boldsymbol{x}$ with $N = n, L = \mathcal{O}(w\log(end))$, $N' = en$, and $d' = d$.*

*Proof.* By Definition H.17, $\mathbf{M}$ can be decomposed with respect to size $end \times end$ matrix

$$\mathbf{E} = \mathbf{E}_1 \cdot \mathbf{E}_2 \cdots \mathbf{E}_w.$$

Further, any $\mathbf{E}_i \in (\mathcal{B}\mathcal{B}^*)$ can be expressed as a product of $2\log end$ butterfly factor matrices. Then by Lemma H.16 and Lemma H.11 we can compute $\mathbf{E}_i\boldsymbol{x}'$ in by stacking $2\log end\,(n, d, L, en, d) - \text{BASECONV}$ models each with $L = \mathcal{O}(1)$. Because $\mathbf{E}$ has width $w$, Lemma H.11 implies that composing with each $\mathbf{E}_i$ for $1 \le i \le w$ constructs a final model with $\mathcal{O}(w\log(end))$ layers. $\quad\square$

Finally, the kaleidoscope hierarchy is related to linear arithmetic circuits via the following result. We note that in (Dao et al., 2020) it is assumed that $w = s$, yet inspection of the proof yields the following stronger result:

**Theorem H.19** ( (Dao et al., 2020))**.** *Let $\mathbf{M}$ be an $n \times n$ matrix such that multiplication of $\mathbf{M}$ times an arbitrary vector $\boldsymbol{u}$ can be be represented as $(n, s, \Delta, w)$-linear arithmetic circuit $\mathcal{C}$. Then, $\mathbf{M} \in (\mathcal{B}\mathcal{B}^*)_{\mathcal{O}(w/n)}^{\mathcal{O}(\Delta)}$.*

We combine Theorem H.19 and Lemma H.18 to show that BASECONV can compute any linear arithmetic circuit with polylogarithmic factors in $\Delta$.

**Corollary H.20.** *For any $(nd, s, \Delta, w)$-linear arithmetic circuit $\mathcal{C}$ that can be represented by a matrix $\mathbf{M} \in \mathbb{R}^{nd \times nd}$ multiplied by a vector $\boldsymbol{x} \in \mathbb{R}^{nd}$, there exists an equivalent $(n, \Delta', d, w, d) - \text{BASECONV}$ with $\Delta' = \mathcal{O}(\Delta\log(w))$ such that $\mathbf{M}\boldsymbol{x} = \text{BASECONV}\,(\boldsymbol{u}, \boldsymbol{h})$ where $\boldsymbol{x}$ is the row major form of $\boldsymbol{u} \in \mathbb{R}^{n \times d}$.*

## H.5 GENERAL ARITHMETIC CIRCUITS

We are now ready to prove the result that yields the equivalency between arithmetic circuits and BASECONV.

**Theorem H.21.** *For any $(nd, s, \Delta, w)$-arithmetic circuit $\mathcal{C}$, there exists an equivalent $(N, \Delta', d, N', d') - \text{BASECONV}$ with $N = n, \Delta' = \mathcal{O}(\Delta\log w)$, $N' = \mathcal{O}(w), d' = d$ that simulates $\mathcal{C}$.*

For the reader's convenience, we begin with a proof sketch and then provide the details afterwards.

*Proof Sketch of Theorem H.21.* Let us layer $\mathcal{C}$ so that each layer $\mathcal{C}_\ell$ for $\ell \in [L_\mathcal{C}]$ either only has linear gates or multiplication gates, where the number of such layers $L_\mathcal{C} = \mathcal{O}(\Delta)$. The composition of all $\mathcal{C}_\ell$ layers results in $\mathcal{C}$. We use $\boldsymbol{z}^{\ell+1} \in \mathbb{R}^w$ to denote the output of the $\ell$-th layer $\mathcal{C}_\ell$ which feeds as the input to the $(\ell + 1)$-th layer $\mathcal{C}_{\ell+1}$. Here, we note that if we can simulate each $\mathcal{C}_\ell$ with BASECONV, then we can simulate the entire layered circuit $\mathcal{C}$ due to Lemma H.11.

Now, if the layer $\mathcal{C}_\ell^{\text{lin}}$ is a linear layer (with only addition gates), then it can be represented by a matrix $\boldsymbol{M} \in \mathbb{R}^{w \times w}$ multiplied by $\boldsymbol{z}^\ell \in \mathbb{R}^w$ (We can append with 0s if necessary so that the input from the previous gates can be written as $w$-length vector). Thus, we can apply Corollary H.20 to simulate $\mathcal{C}_\ell^{\text{lin}}$ with an equivalent $(n, \log w, d, \mathcal{O}(\log w), d) - \text{BASECONV}$ model.

Next, if $\mathcal{C}_\ell^{\text{mult}}$ instead consists of only the multiplication gates. Then, we note here that the output $\boldsymbol{z}^\ell$ may not exactly equal the input to $\mathcal{C}_\ell^{\text{mult}}$. Nevertheless, we can apply a $\mathcal{O}(w)$ sparse linear map $\mathbf{R} \in \mathbb{R}^{w \times w}$ so that $\mathbf{R}\boldsymbol{z}^\ell$ yields vectors $\boldsymbol{v}^1, \boldsymbol{v}^2$, and $\boldsymbol{v}^3$, where $\boldsymbol{v}_1$ constitutes the "first" input to all the multiplication gates and $\boldsymbol{v}_2$ constitutes all the "second" inputs while $\boldsymbol{v}^3$ consists of all entries needed as inputs in the subsequent layers. That is, for the $i$th gate in $\mathcal{C}_\ell^{\text{mult}}$, we compute $\boldsymbol{v}_i^1 \cdot \boldsymbol{v}_i^2$. This implies that for all the gates in $\mathcal{C}_\ell^{\text{mult}}$, we can simply compute $\boldsymbol{v}_1 \odot \boldsymbol{v}_2$. To this end, we use the `remember` primitive with constant number of layers from Proposition H.10 to define

a $(n, \mathcal{O}(\log w), d, w, d) - \text{BASECONV}$ model that remembers $\boldsymbol{v}^3$ while performing the Hadamard product of $\boldsymbol{v}^1$ with $\boldsymbol{v}^2$.

Overall, we can then collect all the resulting BASECONV layers and compose them as in Lemma H.11 to simulate $\mathcal{C}$. Overall, the number of layers used is given by $\mathcal{O}(\Delta \log w)$ while the internal dimension remains fixed at $w$. $\qquad\square$

Using the outline in the proof sketch, we now delve into a detailed analysis of the arithmetic circuit $\mathcal{C}$ to an equivalent BASECONVmodel.

*Proof of Theorem H.21.* Let $\boldsymbol{u} \in \mathbb{R}^{nd}$ be the input to the arithmetic circuit $\mathcal{C}$ with depth $\Delta$ and width $w$. We begin by rearranging the circuit into layers of addition and multiplication gates. That is, each layer $\mathcal{C}^\ell$ has either all addition gates or all multiplication gates. This allows us to readily apply the results from Appendix H.4. Note here that we can assert that the number of such layers $L_\mathcal{C} = \mathcal{O}(\Delta)$. Moreover, we construe the input to each such circuit as a vector of length $w$ by appending with extra 0s if necessary so that the composition of the layers results in a circuit equivalent to $\mathcal{C}$. See Fig. 9 for an example of such decomposition

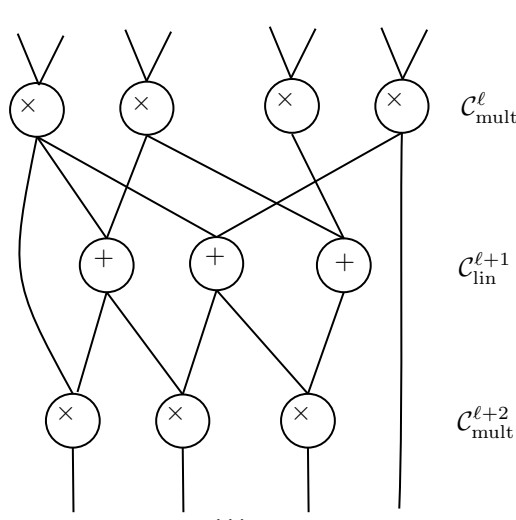

Figure 9: An example decomposition of an arithmetic circuit as layers of only addition or multiplication gates.

Let $\boldsymbol{z}^\ell \in \mathbb{R}^w$ denote the input to each layer $\mathcal{C}_\ell$ for $\ell \in [L_\mathcal{C}]$ with $\boldsymbol{z}^1 \equiv (\boldsymbol{u}, \boldsymbol{0}^{w-nd})$. It is important to note here that we may *not* have $\boldsymbol{z}^{\ell+1} \equiv \mathcal{C}_\ell(\boldsymbol{z}^\ell)$ since the inputs for gates at the $(\ell+1)$-th layer may come from any of the previous layers. We now handle the case of layers with addition gates, multiplication gates, and the intermediate stage separately.

**Addition Gates.** Let $\mathcal{C}_\ell^{\text{lin}}$ denote an arbitrary linear layer which only contains the addition gates that takes in $\boldsymbol{z}^\ell$ as input. We know that there exists a matrix $\mathbf{M} \in \mathbb{R}^{w \times w'}$ such that we have

$$\mathcal{C}_\ell^{\text{lin}}(\boldsymbol{z}^\ell) \equiv \mathbf{M}\boldsymbol{z}^\ell.$$

Here, note here that we may need entries from the vector $\boldsymbol{z}^\ell$ in subsequent layers. Let $s$ be the total number of such entries, then for at each index for such entries $i \in [w]$, we have the corresponding index $i_s \in [s]$. We then append the $i$th standard row vector into the matrix $\mathbf{M}$ to get the matrix $\mathbf{M}' \in \mathbb{R}^{w \times (w'+s)}$ so that we have

$$\left(\mathbf{M}'\boldsymbol{z}^\ell\right)[j] = \begin{cases} \left(\mathbf{M}\boldsymbol{z}^\ell\right)[j] & \text{if } j < w' \\ \boldsymbol{z}^\ell[i] & \text{if } j = w' + i_s \end{cases}$$

Here, note that we must have $w' + s \leq w$ as $w$ is the width of the circuit. If needed, we then append the matrix $\mathbf{M}'$ with all-zero rows $\boldsymbol{0}^w$ so that we get the matrix $\mathbf{M}'' \in \mathbb{R}^{w \times w}$. Note here that we have thus preserved all entries needed in subsequent layers and incorporated the output of the $\ell$th layer with the output $\mathbf{M}''\boldsymbol{z}^\ell \equiv \boldsymbol{z}^{\ell+1}$ serving as the input of the $(\ell+1)$-th layer. Applying Corollary H.20 then yields an equivalent $(n, \mathcal{O}(\log w), d, w, d) - \text{BASECONV}$ model.

**Multiplication Gates.** Next, we deal with the layer $\mathcal{C}_{\text{mult}}^{\ell}$ of multiplication gates by collecting the two inputs to each multiplication gates as $\boldsymbol{v}_1^{\ell}, \boldsymbol{v}_2^{\ell}$. Note that for the input $\boldsymbol{z}^{\ell} \in \mathbb{R}^w$ from the previous layer, we will again have entries that need to be used in the subsequent layers that we will denote as $\boldsymbol{v}_3^{\ell}$. We thus need to permute the entries of $\boldsymbol{z}^{\ell}$ to get the vector $[\boldsymbol{v}_1^{\ell} :: \boldsymbol{v}_2^{\ell} :: \boldsymbol{v}_3^{\ell}]$ so that we can remember $\boldsymbol{v}_3^{\ell}$ while taking the Hadamard product $\boldsymbol{v}_1^{\ell} \odot \boldsymbol{v}_2^{\ell}$. To this end, we can achieve this permutation of entries using a $\mathcal{O}(w)$-sparse linear matrix[12] $\mathbf{R}^{\ell} \in \mathbb{R}^{w \times w}$ with which is equivalently represented by an $(w, w, 1, w)$-linear arithmetic circuit that simply moves the appropriate wires from the previous layer. This can again be achieved by a equivalent $(n, \mathcal{O}(\log w), d, w, d) - \text{BASECONV}$ model. That is, $\mathbf{R}^{\ell} \boldsymbol{z}^{\ell}$ has the following form:

$$\mathbf{R}^{\ell} \boldsymbol{z}^{\ell} \equiv \begin{pmatrix} \boldsymbol{v}_1^{\ell} \\ \boldsymbol{v}_2^{\ell} \\ \boldsymbol{v}_3^{\ell} \end{pmatrix}.$$

Next, we can now define a $(n, 1, d, n, d) - \text{BASECONV}$ which extracts $\boldsymbol{v}_1^{\ell}$ as the projection with the input to the remember primitive given by $\boldsymbol{y}^{\ell} \equiv (\boldsymbol{v}_2^{\ell}, \boldsymbol{v}_3^{\ell}, \boldsymbol{0})$. We then specify the `remember` primitive as $\texttt{remember}\left(\boldsymbol{y}^{\ell} : \boldsymbol{v}_2^{\ell}, \boldsymbol{v}_3^{\ell}, \boldsymbol{e}_0, \boldsymbol{v}_1^{\ell}\right)$ which computes the following

$$\boldsymbol{z}^{\ell+1} \equiv \begin{pmatrix} \boldsymbol{v}_1^{\ell} \odot (\boldsymbol{e}_0 * \boldsymbol{v}_2^{\ell}) \\ \boldsymbol{v}_3^{\ell} \\ \boldsymbol{0} \end{pmatrix}$$

$$\equiv \begin{pmatrix} \boldsymbol{v}_1^{\ell} \odot \boldsymbol{v}_2^{\ell} \\ \boldsymbol{v}_3^{\ell} \\ \boldsymbol{0} \end{pmatrix}$$

$$\equiv \mathcal{C}_{\text{mult}}^{\ell}(\boldsymbol{z}^{\ell}) \in \mathbb{R}^w.$$

Using Proposition H.10, we know that this requires a $(n, \mathcal{O}(\log w), d, w, d) - \text{BASECONV}$ which remembers the entries that will serve as inputs in subsequent layers while performing the Hadamard product $\boldsymbol{v}^1 \odot \boldsymbol{v}^2$.

Overall, we can now stack all the resulting BASECONV layers and compose them as in Lemma H.11 to simulate $\mathcal{C}$. Overall, the number of layers blows up by $\mathcal{O}(\Delta \log w)$ while the internal dimension remains fixed at $w$. $\qquad \square$

## H.6 THE RETNET REDUCTION AND SPACE COMPLEXITY FOR AR

We are now in a position to show that there exists an equivalent BASECONV model that can simulate a Retnet layer from Algorithm 6. Recall that for an input sequence $\boldsymbol{u} \in \mathbb{R}^{N \times d}$, we project the input using weight matrices $\mathbf{W}_A, \mathbf{W}_C, \mathbf{W}_V \in \mathbb{R}^{d \times d}$ to $\mathbf{A}, \mathbf{C}, \mathbf{V} \in \mathbb{R}^{N \times d}$ and compute the following recurrence (see line 5 in Algorithm 6):

$$\boldsymbol{z}^n \equiv \gamma \boldsymbol{z}^{n-1} + (\mathbf{A}[n, :])^{\top} \mathbf{V}[n, :], \tag{6}$$

where $\boldsymbol{z}^n \in \mathbb{R}^{d \times d}$. We unroll this recurrence to express each state $\boldsymbol{z}^n$ in terms of $\boldsymbol{u}$ with coefficients given by $\gamma$ and the weight matrices $\mathbf{W}_A, \mathbf{W}_C, \mathbf{W}_V$ as follows:

$$\boldsymbol{z}^0 \equiv ((\boldsymbol{u}\mathbf{W}_A)[0, :])^{\top} (\boldsymbol{u}\mathbf{W}_V)[0, :]$$
$$\equiv (\boldsymbol{u}[0, :]\mathbf{W}_A)^{\top} (\boldsymbol{u}[0, :]) \mathbf{W}_V$$
$$\equiv \mathbf{W}_A^{\top} (\boldsymbol{u}[0, :])^{\top} (\boldsymbol{u}[0, :]) \mathbf{W}_V$$
$$\equiv \mathbf{W}_A^{\top} (\boldsymbol{u}^{\top}[:, 0]\boldsymbol{u}[0, :]) \mathbf{W}_V.$$

---

[12]Each row of $\mathbf{R}^{\ell}$ has exactly one non-zero entry.

Similarly, we have

$$
\begin{aligned}
\boldsymbol{z}^1 &\equiv \gamma \boldsymbol{z}^0 + \mathbf{W}_A^\top \left(\boldsymbol{u}^\top[:,1]\boldsymbol{u}[1,:]\right) \mathbf{W}_V \\
&\equiv \gamma \left(\mathbf{W}_A^\top \left(\boldsymbol{u}^\top[:,1]\boldsymbol{u}[0,:]\right) \mathbf{W}_V\right) + \mathbf{W}_A^\top \left(\boldsymbol{u}^\top[:,1]\boldsymbol{u}[1,:]\right) \mathbf{W}_V \\
&\equiv \mathbf{W}_A^\top \left(\gamma \left(\boldsymbol{u}^\top[:,0]\boldsymbol{u}[0,:]\right) + \boldsymbol{u}^\top[:,1]\boldsymbol{u}[1,:]\right) \mathbf{W}_V
\end{aligned}
$$

We can then generalize this to $n \in [N]$ to get

$$
\boldsymbol{z}^n \equiv \mathbf{W}_A^\top \left(\sum_{i=0}^n \gamma^{n-i}\boldsymbol{u}^\top[:,i]\boldsymbol{u}[i,:]\right) \mathbf{W}_V. \tag{7}
$$

The above helps us infer that $\boldsymbol{z}^n$ can be expressed as a polynomial in $\{\boldsymbol{u}[0,0], \boldsymbol{u}[0,1], \ldots, \boldsymbol{u}[N-1,d-1]\}$. For each polynomial, there exists an arithmetic circuit that computes the polynomial in a natural way, whence we can apply Theorem H.21 to get an equivalent BASECONV model.

**Corollary H.22.** *For a* RetNet *model with $\mathcal{O}(d^2)$ parameters and $N$ layers, there exists an equivalent* BASECONV*model that uses $\mathcal{O}(Nd)$ parameters and $\mathcal{O}(N \log d)$ layers.*

*Proof.* We will start by describing the arithmetic circuit that computes the states $\boldsymbol{z}^n \in \mathbb{R}^{d \times d}$ here. Indeed, each state requires exactly two alternative layers with only multiplication gates and only addition gates, respectively. First, we note that for each $n \in N$, we have

$$
\boldsymbol{z}^n \equiv \gamma \boldsymbol{z}^{n-1} + (\mathbf{A}[n,:])^\top \mathbf{V}[n,:].
$$

Since we can compute $\boldsymbol{z}^0$ in $\mathcal{O}(1)$ layers and computing $\gamma \cdot \boldsymbol{z}^{n-1}$ needs exactly one layer with each entry from $\boldsymbol{z}^{n-1}$ serving as an input to a multiplication gate along with $\gamma$, the depth of the circuit is $\mathcal{O}(N)$ with width $d^2$. We can thus apply Theorem H.21 to get an $(\mathcal{O}(N), \mathcal{O}(N \log d), d^2, \mathcal{O}(d^2), \mathcal{O}(d^2))-$BASECONV model that can compute $\boldsymbol{z}^n$ for each $n \in N$. $\qquad\square$

**The Space Complexity of AR** In the context of the Retnet Model and associative recall, it is worth exploring the space complexity of the associative recall (AR) problem:

> The AR problem takes key-value pairs $\{\boldsymbol{k}_i, \boldsymbol{v}_i\}_{i=0}^{n-1}$ along with a query $\boldsymbol{q}$ appended at the end as input and the goal is to output $\boldsymbol{v}_i$ if $\boldsymbol{q} = \boldsymbol{k}_i$ for some $i \in [0, N-1]$.

Indeed, we will first provide a lower-bound for *any* model that purports to solve AR. To this end, we will require a randomized communication complexity lower bound result for the *index problem*:

> The index problem has two agents, Alice and Bob, where Alice has a string $\boldsymbol{x} \in \{0,1\}^n$ and Bob has an index $i \in [n]$, and the goal for the players is to output the $i$-th entry $\boldsymbol{x}_i$. Moreover, we also require the communication to be *one-way*: only Alice is allowed to send a single message to Bob and Bob needs to output the answer.

We will make use of the following lower-bound result.

**Theorem H.23** ((Jayram et al., 2008)). *The one-way randomized communication complexity[13] of the index problem for sending an $n$-length bit string is $\Omega(n)$.*

We now use Theorem H.23 to provide a lower bound on the number of bits required by the Retnet model to solve AR.

**Corollary H.24.** *The RetNet model requires $\Omega(N)$-bits to solve AR for $d \le \sqrt{N}$.*

*Proof.* Consider an instance $(\boldsymbol{x}, i)$ of the index problem with $\boldsymbol{x} \in \{0,1\}^N$. We now describe the corresponding instance of the AR problem:

$$
\{i, \boldsymbol{x}_i\}_{i=0}^{N-1}, i. \tag{8}
$$

Next, consider the following one-way protocol for solving the index problem using the RetNet model. Alice with their access of $\boldsymbol{x} \in \{0,1\}^N$ generate an input for AR (without the query) as in equation 8. Alice then runs the RetNet model on $\{i, \boldsymbol{x}_i\}_{i=0}^{N-1}$ and sends the memory content of running the RetNet model to Bob. This should include the output $\boldsymbol{z}^{N-1}$ as we can reasonably assume that both have access weight matrices $\mathbf{W}_A, \mathbf{W}_C, \mathbf{W}_V$ and the scalar $\gamma$. Since we assume

---

[13]The randomized communication complexity of function $f$ is defined as $\min_\pi \|\pi\|$, where $\pi$ ranges over all randomized protocols that can solve $f$ with probability of success at least $2/3$.

that this model solves AR, the output $\mathtt{Out}[N,:] = \boldsymbol{x}_i$ should contain the associated value of $i$. Here, Bob can compute $\mathtt{Out}[N,:]$ by using the memory content sent by Alice along with the term $\gamma \boldsymbol{z}^{N-1}$:

$$\boldsymbol{x}_i = \mathtt{Out}[N,:] = \mathbf{C}[N,:]\boldsymbol{z}^N = \mathbf{C}[N,:]\left(\gamma \boldsymbol{z}^{N-1} + (\mathbf{A}[N,:])^\top \mathbf{V}[N,:]\right).$$

That is, the total number of bits that are communicated in this protocol is $O(d^2)$. For $d \leq \sqrt{N}$, have shown that a one-way communication protocol exists for solving the index problem exists that uses $o(N)$ communication complexity. This contradicts Theorem H.23 and hence, we conclude that the ReNet model solving AR also needs $\Omega(N)$ bits. $\qquad\square$

### H.7 THE MULTIPLE-QUERY ASSOCIATIVE RECALL PROBLEM

#### H.7.1 INTRODUCTION

In this section, we consider a general version of the associative recall problem (Ba et al., 2016).

**Setup.** Next, we will redefine the *multiple-query associative recall* problem (MQAR) from Definition 3.1 to a slightly more general problem:

Suppose we are given an input sequence $\boldsymbol{u}[0 \cdots N - 1] \triangleq \left\{ (\boldsymbol{k}_0, \boldsymbol{v}_0, \boldsymbol{q}_0), \ldots, \left( \boldsymbol{k}_{\frac{N}{3}-1}, \boldsymbol{v}_{\frac{N}{3}-1}, \boldsymbol{q}_{\frac{N}{3}-1} \right) \right\}$ with each $\boldsymbol{k}_i, \boldsymbol{v}_i, \boldsymbol{q}_i \in C$ is a token drawn from a vocabulary of size $c = |C|$. Our goal is then to check, for each $1 \leq i \leq \frac{N}{3} - 1$, whether there exists $0 \leq j < i$ such that $\boldsymbol{q}_i \equiv \boldsymbol{k}_j$, and if so, output $\boldsymbol{v}_j$.

Here, we note that it suffices to have $d \approx \log(c)$ so that $\boldsymbol{k}_i, \boldsymbol{v}_i, \boldsymbol{q}_i$ is embedded in $\{0,1\}^d$. However, we will specify the specific embedding being used for the results below. Here, we construe the tokens $\boldsymbol{k}_i, \boldsymbol{q}_i$ and $\boldsymbol{v}_i$ to be the *keys*, the *queries*, and the *associated values*. Indeed, it might be helpful to think of the input $\boldsymbol{u}$ as a streaming sequence of key-value pairs for which we sequentially employ standard associative recall for every key that shows up in the sequence so far.

To see that the above generalizes Definition 3.1, considers a sequence of length $\frac{N}{3}$: $\boldsymbol{u}[0 \cdots N - 1] := \{\boldsymbol{x}_0, \ldots, \boldsymbol{x}_{N-1}\}$, where each $\boldsymbol{x}_i \in C$. The goal of Definition 3.1 is then to check, for each $1 \leq i < N - 1$, whether there exists $0 \leq j < i$ such that $\boldsymbol{x}_i \equiv \boldsymbol{x}_j$, and if so, output $\boldsymbol{x}_{j+1}$, and continue otherwise. We can reduce this problem to the above general formulation by taking the following sequence of tuples as the input $\{(\boldsymbol{x}_i, \boldsymbol{x}_{i+1}, \boldsymbol{x}_i)\}$.

**Remark H.25.** As noted above, this version is more general than Definition 3.1. Thus, the results proven in the sequel, which are proven for the above general MQAR can be ported (with constant blowup in parameters) to get the results corresponding results for Definition 3.1.

#### H.7.2 MQAR SOLUTION VIA ATTENTION

Before describing how BASECONV solves the multiple-query associative recall problem, we discuss how Attention solves it trivially using pairwise inner-products. To this end, we will specify how the input is presented to attention.

**Remark H.26.** We note that the input for the multiple-query associative recall problem $\boldsymbol{u} \in \{0,1\}^{N \times d}$ has designated indices for the keys, queries, and values in the sequence. We gather these indices below:

$$\begin{aligned} \mathcal{K} &= \{i \in \{0, \ldots, N - 1\} \,|\, i \equiv 0 \mod 3\}, \\ \mathcal{V} &= \{i \in \{0, \ldots, N - 1\} \,|\, i \equiv 1 \mod 3\}, \\ \mathcal{Q} &= \{i \in \{0, \ldots, N - 1\} \,|\, i \equiv 2 \mod 3\}, . \end{aligned} \tag{9}$$

The input $\boldsymbol{u} \in \mathbb{R}^{N \times d}$ to Attention for $d = 3c$ is then given by

$$\boldsymbol{u}[i,:] \equiv \begin{cases} [\boldsymbol{k}_i : \mathbf{0}^c : \mathbf{0}^c] & \text{if } i \in \mathcal{K} \\ [\mathbf{0}^c : \boldsymbol{v}_i : \mathbf{0}^c] & \text{if } i \in \mathcal{V} \\ [\mathbf{0}^c : \mathbf{0}^c : \boldsymbol{q}_i] & \text{if } i \in \mathcal{Q} \end{cases}$$

Here, each $\boldsymbol{k}_i, \boldsymbol{v}_i, \boldsymbol{q}_i$ is embedded as a one-hot encoding in $\{0,1\}^c$.
Without softmax, the output for an attention layer $\mathbf{O} \in \mathbb{R}^{N \times d}$ is given by

$$\mathbf{O} \equiv \left(\mathbf{Q}\mathbf{K}^\top\right)\mathbf{V}, \tag{10}$$

where $\mathbf{Q}, \mathbf{K}, \mathbf{V} \in \mathbb{R}^{N \times d}$ are defined as $\boldsymbol{u}\mathbf{W}_Q, \boldsymbol{u}\mathbf{W}_K, \boldsymbol{u}\mathbf{W}_V$ for $\boldsymbol{u} \in \mathbb{R}^{N \times d}$. Instead of position embeddings, we use ALiBi, a popular technique that biases the attention scores $\mathbf{Q}\mathbf{K}^\top$ with a lower-triangular Toeplitz matrix $\mathbf{B} \in \mathbb{R}^{N \times N}$ (Press et al., 2021). The values in this matrix are controlled by a fixed hyperparameter so they do not count towards the number of parameters in the model.

---

**Algorithm 8** `ALiBi-without-softmax` $(\boldsymbol{u}[0\cdots N-1], \mathbf{O}_{\mathrm{prev}}[0\cdots N-1], \mathbf{B}))$

---

**Input:** Input sequence $\boldsymbol{u}[0\cdots N-1] \triangleq \{(\boldsymbol{k}_i, \boldsymbol{v}_i, \boldsymbol{q}_i)\}_{i=0}^{\frac{N}{3}-1}$ with each $\boldsymbol{k}_i, \boldsymbol{v}_i, \boldsymbol{q}_i \in \{0,1\}^{3c}$, previous layer's output $\mathbf{O}_{\mathrm{prev}}[0\cdots N-1] \in \mathbb{R}^{N\times 3c}$, and linear bias $\mathbf{B} \in \mathbb{R}^{N\times N}$.
 1: Add $\boldsymbol{u}_{\mathrm{curr}} \leftarrow \boldsymbol{u} + \mathbf{O}_{\mathrm{prev}}$ as an input to this layer.
 2: $\mathbf{K}, \mathbf{Q}, \mathbf{V} \leftarrow \boldsymbol{u}_{\mathrm{curr}}\mathbf{W}_Q, \boldsymbol{u}_{\mathrm{curr}}\mathbf{W}_K, \boldsymbol{u}_{\mathrm{curr}}\mathbf{W}_V.$
 3: $\mathbf{O} \leftarrow \left(\mathbf{Q}\mathbf{K}^\top + \mathbf{B}\right)\mathbf{V}$
 4: **return O** as the output of this layer.

---

**Proposition H.27.** *Given an input* $\boldsymbol{u} \in \{0,1\}^{N\times d}$ *(encoded as in Remark H.26) where* $d = 3c$, *Attention with linear biases (even without using soft-max) solves* MQAR *for* $\boldsymbol{u}$ *using* $\mathcal{O}(c^2)$ *parameters,* $\mathcal{O}(Nc^2 + N^2c)$ *time complexity and* $\mathcal{O}(1)$ *layers.*

*Proof.* We use two layers of attention. We will start by specifying the projection matrices for the first layer $\mathbf{W}_Q^1, \mathbf{W}_K^1, \mathbf{W}_V^1 \in \mathbb{R}^{d\times d}$ as:

$$\mathbf{W}_K^1 \equiv \mathbf{W}_Q^1 \equiv \mathbf{0}, \quad \mathbf{W}_V^1 \equiv \begin{pmatrix} \mathbf{0} & \mathbf{0} & \mathbf{0} \\ \mathbf{0} & \mathbf{I_{c\times c}} & \mathbf{0} \\ \mathbf{0} & \mathbf{0} & \mathbf{0} \end{pmatrix}$$

Above, $\mathbf{W_V^1}$ is meant to isolate the $\boldsymbol{v}_i$ embeddings. For the first layer, we then have $\mathbf{Q}^1 \equiv \mathbf{K}^1 \equiv \mathbf{0}$, and

$$\mathbf{V}^1[i,:] := \boldsymbol{u}^1[i,:]\mathbf{W}_V^1 \equiv \begin{cases} [\mathbf{0}^c : \boldsymbol{v}_i : \mathbf{0}^c] & \text{if } i \in \mathcal{V}, \\ \mathbf{0}^d & \text{otherwise} \end{cases},$$

where $\mathcal{K}, \mathcal{Q}, \mathcal{V}$ are defined as in equation 9. The output for the first layer is given by the following:

$$\begin{aligned} \mathbf{O}^1[i,:] &= \left(\left(\mathbf{Q}\mathbf{K}^\top + \mathbf{B}^1\right)\mathbf{V}\right)[i,:] \\ &= \mathbf{B}^1\mathbf{V}[i,:] \\ &= \begin{cases} [\mathbf{0}^c : \boldsymbol{v}_i : \mathbf{0}^c] & \text{if } i \in \mathcal{K} \\ \mathbf{0}^d & \text{otherwise} \end{cases}, \end{aligned}$$

The $\mathbf{Q}\mathbf{K}^\mathbf{T}$ is ignored $(\equiv \mathbf{0})$ and we isolate the shifted sequence by setting the bias matrix $\mathbf{B}$ appropriately. In particular, the last equality follows from the fact that $\mathbf{B}^1$ is an up-shift matrix that shifts each row of $\mathbf{V}$ by 1. We apply the residual connection at the end of the standard Transformer block to insert the $\boldsymbol{k}_i$ adjacent to the $\boldsymbol{v}_i$. For the second layer, the input $\boldsymbol{u}^2 \in \mathbb{R}^{N\times d}$ is given by

$$\boldsymbol{u}^2[i,:] \equiv \begin{cases} [\boldsymbol{k}_i : \boldsymbol{v}_i : \mathbf{0}^c] & \text{if } i \in \mathcal{K} \\ [\mathbf{0}^c : \boldsymbol{v}_i : \mathbf{0}^c] & \text{if } i \in \mathcal{V} \\ [\mathbf{0}^c : \mathbf{0}^c : \boldsymbol{q}_i] & \text{if } i \in \mathcal{Q} \end{cases},$$

Further, we take the following projection matrices:

$$\mathbf{W}_K^2 \equiv \begin{pmatrix} \mathbf{I}_{c\times c} & \mathbf{0} & \mathbf{0} \\ \mathbf{0} & \mathbf{0} & \mathbf{0} \\ \mathbf{0} & \mathbf{0} & \mathbf{0} \end{pmatrix}, \mathbf{W}_Q^2 \equiv \begin{pmatrix} \mathbf{0} & \mathbf{0} & \mathbf{0} \\ \mathbf{0} & \mathbf{0} & \mathbf{0} \\ \mathbf{I}_{c\times c} & \mathbf{0} & \mathbf{0} \end{pmatrix}, \mathbf{W}_V^2 \equiv \begin{pmatrix} \mathbf{0} & \mathbf{0} & \mathbf{0} \\ \mathbf{I}_{c\times c} & \mathbf{0} & \mathbf{0} \\ \mathbf{0} & \mathbf{0} & \mathbf{0} \end{pmatrix}$$

We then have the following matrices as input to attention after applying projection:

$$\begin{aligned} \mathbf{Q}^2[i,:] &:= \boldsymbol{u}^2[i,:]\mathbf{W}_K^2 \equiv \begin{cases} [\boldsymbol{q}_i : \mathbf{0}^{2c}] & \text{if } i \in \mathcal{Q}, \\ \mathbf{0}^d & \text{otherwise} \end{cases}, \\ \mathbf{K}^2[i,:] &:= \boldsymbol{u}^2[i,:]\mathbf{W}_Q^2 \equiv \begin{cases} [\boldsymbol{k}_i : \mathbf{0}^{2c}] & \text{if } i \in \mathcal{K}, \\ \mathbf{0}^d & \text{otherwise} \end{cases}, \quad (11) \\ \mathbf{V}^2[i,:] &:= \boldsymbol{u}^2[i,:]\mathbf{W}_V^2 \equiv \begin{cases} [\boldsymbol{v}_i : \mathbf{0}^{2c}] & \text{if } i \in \mathcal{K} \cup \mathcal{V}, \\ \mathbf{0}^d & \text{otherwise} \end{cases}. \end{aligned}$$

Here, note that the values have been shifted to the corresponding key position. Next, we compute the term in the parenthesis for the second layer as

$$
\begin{aligned}
\left(\mathbf{Q}^2\mathbf{K}^{2\top}\right)[i,j] = \left(\mathbf{Q}^2\mathbf{K}^{2\top}\right)[i,j] &= \langle \mathbf{Q}^2[i,:], \mathbf{K}^{2\top}[:,j]\rangle \\
&= \langle \mathbf{Q}^2[i,:], \mathbf{K}^2[j,:]\rangle \\
&= \begin{cases} \langle \boldsymbol{q}_i, \boldsymbol{k}_j\rangle & \text{if } i \in \mathcal{Q}, j \in \mathcal{K} \\ 0 & \text{otherwise} \end{cases} \\
&= \begin{cases} 1 & \text{if } i \in \mathcal{Q}, j \in \mathcal{K}, \boldsymbol{q}_i \equiv \boldsymbol{k}_j \equiv \boldsymbol{e}_k \text{ for some } k \\ 0 & \text{otherwise.} \end{cases}
\end{aligned}
$$

Finally, we compute the output as follows:

$$
\begin{aligned}
\mathbf{O}^2[i,:] &= \left(\left(\mathbf{Q}^2\mathbf{K}^{2\top} + \mathbf{B}^2\right)\mathbf{V}^2\right)[i,:] \\
&= \left(\left(\mathbf{Q}^2\mathbf{K}^{2\top} + \mathbf{0}\right)\mathbf{V}^2\right)[i,:] \\
&= \left(\mathbf{Q}^2\mathbf{K}^{2\top}\right)[i,:] \cdot \mathbf{V}^2 \\
&= \sum_{j=0}^{N-1} \left(\mathbf{Q}^2\mathbf{K}^{2\top}\right)[i,j] \cdot \mathbf{V}^2[j,:] \\
&= \sum_{j\in\mathcal{K}} \left(\mathbf{Q}^2\mathbf{K}^{2\top}\right)[i,j] \cdot [\boldsymbol{v}_j : \mathbf{0}^{2c}] \\
&= \begin{cases} [\boldsymbol{v}_j : \mathbf{0}^{2c}] & \text{if } j \in \mathcal{K}, i \in \mathcal{Q}, \boldsymbol{q}_i \equiv \boldsymbol{k}_j \\ \mathbf{0}^d & \text{otherwise} \end{cases},
\end{aligned}
$$

where we use the fact that for each index $j \in \mathcal{K}$, the matrix $\mathbf{V}^2$ contains the associated value from equation 11. Thus, for each query $\boldsymbol{q}_i$, we solve the associated value problem yielding a match for the $j$th key.

In total, we only need $\mathcal{O}(c^2)$-many parameters to perform these multiplications and the linear bias in the first layer is a hyperparameter that is static and unlearned; the time complexity comes from the multiplication of $\mathbf{Q}\mathbf{K}^\top$ in $\mathcal{O}(N^2 c)$, and projections in $\mathcal{O}(Nc^2)$. Finally, we only need $\mathcal{O}(1)$ layers for this solution. □

In the sequel, we develop a parallel algorithm to solve the multiple-query associative recall problem with $\mathcal{O}(Nd \cdot \log^2 N)$ work complexity and $\mathcal{O}(d \cdot \log^2 N)$ time. We then convert the algorithm into a BASECONV model via the route of arithmetic circuits, which then solves the multiple-query associative recall problem with $\tilde{\mathcal{O}}(1)$ layers and $\tilde{\mathcal{O}}(Nd)$ parameters.

### H.7.3 INITIAL ATTEMPT: A SEQUENTIAL ALGORITHM

We will first discuss the algorithm that simply uses an associative array to solve the multiple-query associative recall problem. Specifically, we want to use a data structure that allows for logarithmic insertion and membership query. Here, we do not specify a choice but data structures including self-balancing binary search trees which allow for $\mathcal{O}(\log N \cdot d)$ `insert` and `find` operations for $d$-bit entries should be sufficient.

---

**Algorithm 9** `Sequential-MQ-AR`$(\boldsymbol{u})[0\cdots N-1]$

---

**Input:** Input sequence $\boldsymbol{u}[0\cdots N-1] \triangleq \{(\boldsymbol{k}_i, \boldsymbol{v}_i, \boldsymbol{q}_i)\}_{i=0}^{\frac{N}{3}-1}$ with each $\boldsymbol{k}_i, \boldsymbol{v}_i, \boldsymbol{q}_i \in \{0,1\}^d$.
 1: Initialize an associative array with `insert` and `find` and an output array `out` $\leftarrow []$.
 2: **for** $i \in \{0, \dots, \frac{N}{3}-1\}$ **do**
 3:      $(\boldsymbol{k}_j, \boldsymbol{v}_j) \leftarrow$ `find`$(\boldsymbol{q}_i)$          ▷ Query for $\boldsymbol{q}_i$ in the data structure.
 4:      **if** $\boldsymbol{k}_j$ is not `null` **then**
 5:          Add $\boldsymbol{v}_j$ to `out`.
 6:      `insert`$(\boldsymbol{k}_i, \boldsymbol{v}_i)$          ▷ Add the key-value pair to the data structure.
 7: **return** `out`.

---

**Proposition H.28.** *Algorithm 9 solves the multiple-query associative recall problem* (MQAR) *in* $\mathcal{O}(dN \log N)$ *time for an input sequence* $\boldsymbol{u} \in \{0,1\}^{N\times d}$.

*Proof.* For any $i \in \{0, \dots, \frac{N}{3} - 1\}$, we know that both insertion and lookup operations take $\mathcal{O}(\log(i) \cdot d)$ time. Overall, the runtime of the algorithm is

$$\sum_{i=0}^{\frac{N}{3}-1} \mathcal{O}(\log(i) \cdot d) = \mathcal{O}(\log(N!) \cdot d) = \mathcal{O}(N \log N \cdot d).$$

$\square$

### H.7.4 Algorithm via Parallel Binary Search

Our plan is to convert the algorithm for solving the multiple-query associative recall problem in the RAM model into an arithmetic circuit, which by Theorem H.21 will lead to a BASECONV model that solves the multiple-query associative recall problem. With respect to Algorithm 9, it may be the case that the arithmetic circuit has a large number of layers $\Omega(N)$. Unfortunately, this would imply that the resulting BASECONV model may have near quadratic complexity in $N$. Instead, we now initiate our effort into designing a BASECONV model with both small enough number of parameters and number of layers. Here, we will first subdivide the problem using dyadic intervals into $\mathcal{O}(N)$ subproblems and reduce each such subproblem into a *multiple search problem* (Akl and Meijer, 1990). To this end, we briefly introduce the multiple search problem below.

> Given two array of numbers $A \triangleq a_0 \leq \dots \leq a_{n-1}$ and $B \triangleq (b_0 \leq \dots \leq b_{m-1})$ with $n \leq m$, for each $a_j \in A$, the goal is to find the smallest element in $B$ that is larger than or equal to $a_j$.

The multiple search problem is solved by a *parallel binary search* (pbs) algorithm in (Akl and Meijer, 1990) with work complexity $\mathcal{O}(n \cdot \log m)$ and time $\mathcal{O}(\log n \log m)$. Specifically, for sorted arrays $A[0 \cdots n-1]$ and $B[0 \cdots m-1]$, pbs constructs the array $C[0 \cdots n-1]$ defined as

$$C[i] \triangleq \begin{cases} \min_{0 \leq j < m} \{j \mid A[i] \leq B[j]\} & \text{if } A[i] \leq B[m-1] \\ m & \text{otherwise.} \end{cases} \tag{12}$$

The algorithm itself runs in exclusive-read exclusive-write (EREW) PRAM model—no two processors are allowed to read from or write into the same memory location at the same time.

We now augment the algorithm copied from (Akl and Meijer, 1990) for our purposes below.

---

**Algorithm 10** pbs-key-values $(\boldsymbol{q}[s \cdots t], \boldsymbol{k}[x \cdots y], n, m)$

---

**Input:** sorted arrays $\boldsymbol{q}[s \cdots t] := \{\boldsymbol{q}_i\}_{i=s}^{t}$, $\boldsymbol{k}[x \cdots y] := \{(j, \boldsymbol{k}_j)\}_{j=x}^{y}$.

1: Initialize $n$ processors denoted $P_0, P_1, \dots, P_{n-1}$      ▷ {Sequential steps are assumed to be executed by $P_s$.}
2: Initialize the output array $C := [m]_{i=s}^{t}$.
3: **if** $s \leq t$ **then**
4:      mid $\leftarrow \lfloor (s+t)/2 \rfloor$
5:      **if** $\boldsymbol{q}[\text{mid}] \leq \boldsymbol{k}[x][1]$ **then**
6:          **for** $i := s$ to mid in parallel **do**
7:              $C[i] \leftarrow j$      ▷ Step executed in parallel by $P_i$
8:          pbs-key-values $(\boldsymbol{q}[\text{mid}+1 \cdots t], \boldsymbol{k}[x \cdots y])$
9:      **else**
10:          **if** $\boldsymbol{q}[\text{mid}] > \boldsymbol{k}[y][1]$ **then**
11:              **for** $i := \text{mid}$ to $t$ in parallel **do**
12:                  $C[i] \leftarrow y + 1$      ▷ Step executed in parallel by $P_i$
13:              pbs-key-values $(\boldsymbol{q}[s \cdots \text{mid}-1], \boldsymbol{k}[x \cdots y])$
14:          **else**      ▷ C[mid] is determined using sequential binary search
15:              $z \leftarrow \min_{x \leq j \leq y} \{j \mid \boldsymbol{q}[\text{mid}] \leq \boldsymbol{k}[j][1]\}$
16:              $C[\text{mid}] \leftarrow z$
17:              **do** steps 18 and 19 in parallel
18:                  pbs-key-values $(\boldsymbol{q}[s \cdots \text{mid}-1], \boldsymbol{k}[x \cdots z-1])]$
19:                  pbs-key-values $(\boldsymbol{q}[\text{mid}+1 \cdots t], \boldsymbol{k}[z \cdots y])$
20: **return** $C$.

---

Let $\Sigma$ be the set $\{0, 1\}$ and denote the set of binary strings of size $n$ as $\Sigma^n$. We define $\text{prefix}(\boldsymbol{x})$ for $n$-bit strings as the set of all initial substrings of $\boldsymbol{x} \in \Sigma^n$ which includes the empty string and

$\boldsymbol{x}$ itself. Next, let $\text{dec} : \{0,1\}^n \to \mathbb{N}$ be the decimal representation of an $n$-bit string $\boldsymbol{x}$ with $\boldsymbol{x}[0]$ denoting the least significant bit. We also use $\text{sort}(A)$ as a procedure that sorts an array $A$. Finally, wlog, we assume that $N$ is a power of 2. We are now ready to present a parallel algorithm that solves the multiple-query associative recall problem below.

---

**Algorithm 11** `Parallel-MQAR` $(\boldsymbol{u}[0 \cdots N-1])$

---

**Input:** Input sequence $\boldsymbol{u}[0 \cdots N-1] \triangleq \{(\boldsymbol{k}_i, \boldsymbol{v}_i, \boldsymbol{q}_i)\}_{i=0}^{\frac{N}{3}-1}$ with each $\boldsymbol{k}_i, \boldsymbol{v}_i, \boldsymbol{q}_i \in \{0,1\}^d$.
1: Initialize $\frac{N}{3} \log\left(\frac{N}{3}\right)$ processors denoted $P_0, \ldots, P_{\frac{N}{3}\log\left(\frac{N}{3}\right)-1}$.
2: Initialize the index and output array `idx, val` $\leftarrow []$.
3: **for** $k := \{0, \ldots, \log\left(\frac{N}{3}\right) - 1\}$ **do**
4:     **for** $\boldsymbol{x} := \{\boldsymbol{x} \in \Sigma^{\log\left(\frac{N}{3}\right)-k}|\ \boldsymbol{x}[\log\left(\frac{N}{3}\right) - k - 1] = 0\}$ **do**
5:         {*All the steps below are executed in parallel by* $\{\{\{P_i^{\boldsymbol{x},k}\}_{\boldsymbol{x}}\}_{i\in[0,2k-1]}\}_k\}$
6:         $I_k^{\boldsymbol{x}} \leftarrow \{\boldsymbol{y} \in \Sigma^{\log\left(\frac{N}{3}\right)}|\ \boldsymbol{x} \in \text{prefix}(\boldsymbol{y})\}$.
7:         $\boldsymbol{k}_{\text{sorted}}^{k\boldsymbol{x}}, I_{\text{permuted}}^{k\boldsymbol{x}} \leftarrow \text{sort}\left(\{\boldsymbol{k}_{\text{dec}(i)}\}_{i\in I_k^{\boldsymbol{x}}}\right)$                 ▷
    $I_{\text{permuted}}^{k\boldsymbol{x}} := \{(j, \text{dec}(i))|\ \boldsymbol{k}_{\text{sorted}}^{k\boldsymbol{x}}[j] \equiv \boldsymbol{k}_{\text{dec}(i)}\}$
8:         $\boldsymbol{x}[\log\left(\frac{N}{3}\right) - k - 1] \leftarrow 1$
9:         $J_k^{\boldsymbol{x}} \leftarrow \{\boldsymbol{y} \in \Sigma^{\log\left(\frac{N}{3}\right)}|\ \boldsymbol{x} \in \text{prefix}(\boldsymbol{y})\}$.
10:        $\boldsymbol{q}_{\text{sorted}}^{k\boldsymbol{x}}, J_{\text{permuted}}^{k\boldsymbol{x}} \leftarrow \text{sort}\left(\{\boldsymbol{q}_{\text{dec}(j)}\}_{j\in J_k^{\boldsymbol{x}}}\right)$            ▷
    $J_{\text{permuted}}^{k\boldsymbol{x}} := \{(\text{dec}(j), k)|\ \boldsymbol{q}_{\text{sorted}}^{k\boldsymbol{x}}[k] \equiv \boldsymbol{q}_{\text{dec}(j)}\}$
11:        $C_k \leftarrow \text{pbs-key-values}\left(\boldsymbol{q}_{\text{sorted}}^{k\boldsymbol{x}}, \boldsymbol{k}_{\text{sorted}}^{k\boldsymbol{x}}, 2^k, 2^k\right)$
12:        **for** $j \in J_k^{\boldsymbol{x}}$ **do**
13:            **if** $C_k[\text{dec}(j)] \neq 2^k$ **then**
14:                $\boldsymbol{c}_j^{k\boldsymbol{x}} \leftarrow C_k[J_{\text{permuted}}^{k\boldsymbol{x}}(\text{dec}(j))]$
15:                **if** $\boldsymbol{c}_j^{k\boldsymbol{x}} \neq 2^k$ **then**                  ▷ cf. equation 12
16:                   Add $I_{\text{permuted}}^{k\boldsymbol{x}}(\boldsymbol{c}_j^{k\boldsymbol{x}})$ to `idx`$[\text{dec}(j)]$.
17: **for** $i \in \{1, \ldots, \frac{N}{3} - 1\}$ **do**
18:     {*Executed in parallel by* $P_i$.}
19:     **if** $\exists j \in$ `idx`$[i]$ **then**
20:         Add $\boldsymbol{v}_{j+1}$ to `val`
21: **return** `val`.

---

**Remark H.29.** In lines 7 and 10, we keep track of the sorted permutation of the indices of keys and queries, respectively. This helps us in the retrieval of the index of the matching key as in line 16.

### H.7.5 CORRECTNESS AND COMPLEXITY

**Proposition H.30.** *Algorithm 11 solves the multiple-query associative recall problem with work complexity* $\mathcal{O}(Nd \cdot \log^2 N)$ *and time* $\mathcal{O}(d \cdot \log^2 N)$.

*Proof.* The correctness of `pbs` implies the correctness of Algorithm 11 if we can show that, for each $1 \leq i < \frac{N}{3}$, we check for a match among the keys $\{\boldsymbol{k}_j\}_{j\in[i-1]}$. To this end, for each $1 \leq i < \frac{N}{3}$, let the set of all iterator indices associated with an index $i$ be defined as $K_i \triangleq \{(k, x)|i \in J_k^{\boldsymbol{x}}\}$ with $J_k^{\boldsymbol{x}}$ as noted in line 9. Then, we define the corresponding set for keys as $\mathcal{I}_i \triangleq \bigcup_{(k,\boldsymbol{x})\in K_i} I_k^{\boldsymbol{x}}$ with $I_k^{\boldsymbol{x}}$s defined as in line 6. That is, for all the calls to `pbs-key-values` that $i$ is part of (given by $K_i$) where the algorithm checks for a match among the keys in $\mathcal{I}_i$, it then suffices to show that $\mathcal{I}_i = [i-1]$.

Here, first note that if some index $j \in I_k^{\boldsymbol{x}} \subseteq \mathcal{I}_i$ for some $\boldsymbol{x} \in \Sigma^{\log\left(\frac{N}{3}\right)-k}$, then, by definition, $\boldsymbol{x} \in \text{prefix}(\text{bin}(j))$. Here, let $\boldsymbol{x}^1 := \boldsymbol{x}|_{\boldsymbol{x}[\log\left(\frac{N}{3}\right)-k]=1}$ where we set the $(\log\left(\frac{N}{3}\right) - k)$-th index of $\boldsymbol{x}$ to be 1. Consequently, as we have $i \in J_k^{\boldsymbol{x}}$ for the same $k$ and $\boldsymbol{x}$ as in $I_k^{\boldsymbol{x}}$ (*cf.* line 6), we must have $\boldsymbol{x}^1 \in \text{prefix}(\text{bin}(i))$. Thus we get $j < i$, whence we can claim that $\mathcal{I}_i \subseteq [i-1]$.
For the other direction, for any $i$, let $b$ denote the position of the most significant bit in $\text{bin}(i)$ which differs from $\text{bin}(j)$ for any $j \in [i-1]$. Then, there must exist a binary string that is in the prefix set of both $\text{bin}(i)$ and $\text{bin}(i)$. That is, there exists $\boldsymbol{x} \in \text{prefix}(\text{bin}(i)) \cap \text{prefix}(\text{bin}(j))$ with $\boldsymbol{x} \in \Sigma^b$.

Thus, we then must have $\mathrm{bin}(j) \in I^{\boldsymbol{x}}_{\log\left(\frac{N}{3}\right)-b}$ and $\mathrm{bin}(i) \in J^{\boldsymbol{x}}_{\log\left(\frac{N}{3}\right)-b}$ with $\boldsymbol{x}$ as the corresponding witness. Hence, we have $[i-1] \subseteq \mathcal{I}_i$.

Overall, we have shown that $\mathcal{I}_i = [i-1]$. Since this holds for all $1 \leq i < \frac{N}{3}$, we can conclude that Algorithm 11 solves the multiple-query associative recall problem.

Next, it is easy to see that we execute lines 6 to 16 $\sum_{k=0}^{\log\left(\frac{N}{3}\right)-1} \frac{\frac{N}{3}}{2^{k+1}}$-many times. We note that sorting $n$ values each of size $d$ can be done with work complexity $n \log n \cdot d$. We note that, at each instance, we are sorting sorting $2^k$ values. Meanwhile, remembering $2^k$ sorted permutation of indices can be done in linear time using arrays. Moreover, each call to $\texttt{pbs-key-values}$ has $n = m = 2^k$ which has work complexity $n \log m$. Finally, we know that the work complexity of lines 18 to 20 is $\mathcal{O}(N)$. Thus, the overall work complexity of Algorithm 11 is

$$d \cdot \sum_{k=0}^{\log\left(\frac{N}{3}\right)-1} \frac{\frac{N}{3}}{2^{k+1}} \mathcal{O}(2^k \cdot \log 2^k) = d \cdot \mathcal{O}(N) \cdot \sum_{k=0}^{\log N/3 - 1} \mathcal{O}(k) = \mathcal{O}(Nd \cdot \log^2 N). \quad (13)$$

We will now analyze the depth of Algorithm 11. We know that the depth of computation for Algorithm 10 is $\mathcal{O}(\log n \log m)$ for input sizes. Moreover, we have $\mathcal{O}(1)$ depth for the computation in 18 to 20 as each entry in $\texttt{idx}$ can have at most one entry. Since the nested for loops iterating over $k$s and the associated $\boldsymbol{x}$s runs in parallel, the depth of Algorithm 11 is dominated by the largest depth among all calls to $\texttt{pbs-key-values}$ and to $\texttt{sort}$. The largest such call to $\texttt{pbs-key-values}$ is of size $n = m = 2^{\log\left(\frac{N}{3}\right)-1} = N/6$ which yields a depth of $d \cdot \log^2 \frac{N}{3}$. Moreover, using sorting networks (Definition H.34), we know that the largest depth is for sorting $\frac{N}{6}$ values of size $d$ given by $d \cdot \Theta(\log N)$ (Lemma H.35). Thus, we can conclude that Algorithm 11 takes $\mathcal{O}(d \cdot \log^2 N)$, time where $N$ is the length of the input. $\qquad\square$

### H.7.6 Conversion to Arithmetic Circuit

We will convert Algorithm 11 to an arithmetic circuits modularly. In particular, after writing out an explicit circuit for Algorithm 10, we will uses this circuit as a black-box along with circuits for sorting networks.

**Circuit for $\texttt{pbs-key-values}$.** We will denote the corresponding arithmetic circuit for Algorithm 10 as $\texttt{pbs-key-values}$ as well with the input gates comprising of each entry from $\boldsymbol{q}[s \cdots t]$ and $\boldsymbol{k}[x \cdots y]$ and the $i$-th output gate yielding the value $C[i]$ as in Algorithm 10.

Here, we first convert the comparisons for the $\texttt{if}$ statements in Algorithm 10. To this end, we briefly introduce comparators.

**Definition H.31** (Comparators). A *comparator* is a device with inputs $\boldsymbol{x}$ and $\boldsymbol{y}$ and outputs $\boldsymbol{x}'$ and $\boldsymbol{y}'$, that performs the following function:

$$\boldsymbol{x}' = \min(\boldsymbol{x}, \boldsymbol{y}),$$
$$\boldsymbol{y}' = \max(\boldsymbol{x}, \boldsymbol{y}).$$

Using comparators, we can use bit-wise XOR and AND to define the result of the comparisons in lines 5 and 10 as the following fixed variables:

$$\begin{aligned}
\ell_x &:= \mathbb{1}\{\boldsymbol{q}[\mathrm{mid}] \leq \boldsymbol{k}_x\}, \\
g_y &:= \mathbb{1}\{\boldsymbol{q}[\mathrm{mid}] > \boldsymbol{k}_y\}, \\
z &:= \texttt{bin-search}(\{\boldsymbol{k}_i\}_{i=s}^t, \boldsymbol{q}[\mathrm{mid}]).
\end{aligned} \quad (14)$$

This then allows us to infer the index of the key array for each recursive call to $\texttt{pbs-key-values}$ in lines 8, 13, 18, and 19 from Algorithm 10. Specifically, let $z_s$ and $z_t - 1$ denote the *starting and ending indices* for the keys as inputs to the recursive calls in Algorithm 10) below:

$$\texttt{pbs-key-values}(\boldsymbol{q}[\mathrm{mid}+1 \cdots t], \boldsymbol{k}[z_t \cdots y]); \text{ (lines 8 and 19)} \quad (15)$$
$$\texttt{pbs-key-values}(\boldsymbol{q}[s \cdots \mathrm{mid}-1], \boldsymbol{k}[x \cdots z_s - 1]); \text{ (lines 13 and 18)} \quad (16)$$

Here, $z_t$ and $z_s$ can assume values dependent on the results of the comparisons in lines 5 and 10. Specifically, we have

$$
z_t = \ell_x \cdot x + (1 - \ell_x)(1 - g_y) \cdot z = \begin{cases} x & \text{if } \boldsymbol{q}[\text{mid}] \leq \boldsymbol{k}_x \text{ (line 8)} \\ z & \text{if } \boldsymbol{q}[\text{mid}] \in (\boldsymbol{k}_x, \boldsymbol{k}_y] \text{ (line 19)} \\ 0 & \text{otherwise} \end{cases},
$$

$$
z_s = g_y \cdot (y + 1) + (1 - \ell_x)(1 - g_y) \cdot z = \begin{cases} y + 1 & \text{if } \boldsymbol{q}[\text{mid}] > \boldsymbol{k}_y \text{ (line 13)} \\ z & \text{if } \boldsymbol{q}[\text{mid}] \in (\boldsymbol{k}_x, \boldsymbol{k}_y] \text{ (line 18)} \\ 0 & \text{otherwise} \end{cases}.
$$

Here, $z_s$ or $z_t$ getting a value 0 signifies that the branch is dead, and we do not execute the recursive call.

Finally, let the arrays $C_t[\text{mid} + 1 \cdots t]$ and $C_s[s \cdots \text{mid} - 1]$ denote the outputs to the recursive calls in equation 15 and equation 16, respectively. We can then succinctly express the outputs for each index of the output array $C$ as

$$
C[i] = \begin{cases} \ell_x \cdot x + z_s \cdot (C_1[i]) & i \in [s \cdots \text{mid} - 1] \\ g_y \cdot (y + 1) + z_t \cdot (C_2[i]) & i \in [\text{mid} + 1 \cdots t] \\ \ell_x \cdot x + (1 - \ell_x)(1 - g_y) \cdot z + g_y \cdot (y + 1) & i = \text{mid} \end{cases} \tag{17}
$$

We can thus state the circuit schematically in Fig. 10.

Now, before accounting for the complexity of the circuit for Algorithm 10, we must first assert the complexity of the comparators that we use in Fig. 10.

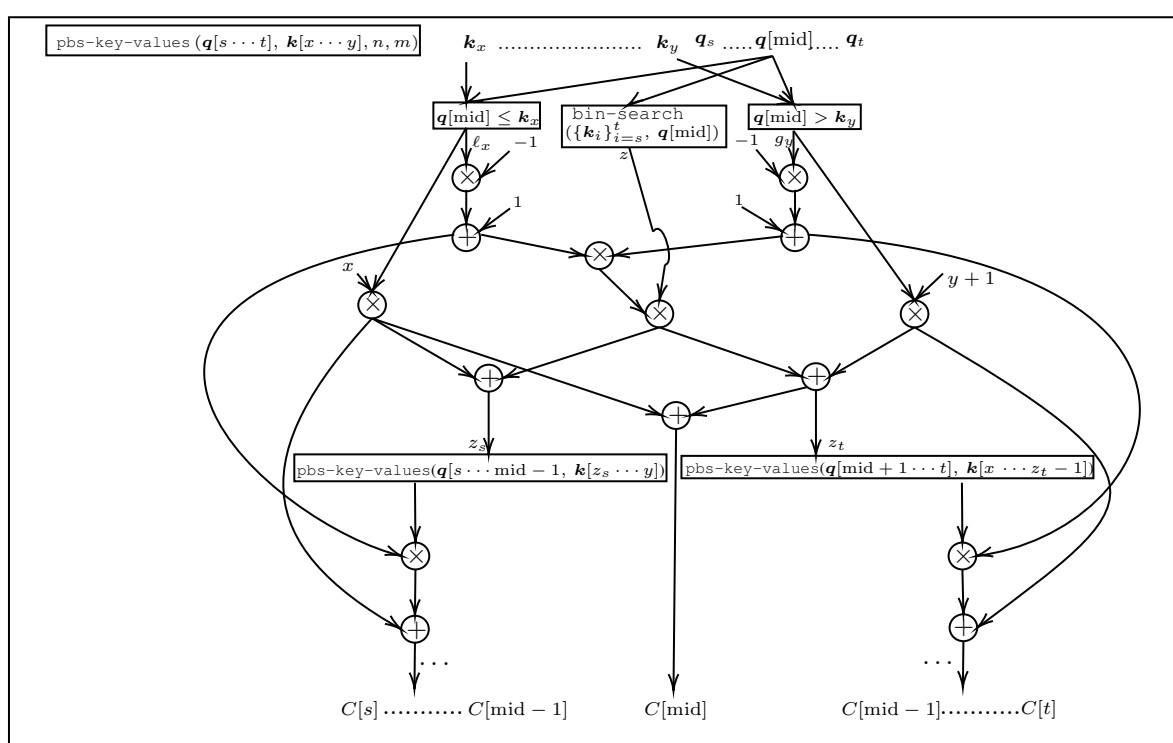

Figure 10: `pbs-key-values` $(\boldsymbol{q}[s \cdots t], \boldsymbol{k}[x \cdots y], n, m)$ as a circuit with recursive calls and subprocedures as "black boxes."

**Lemma H.32** (Cormen et al. (2022)). *For binary strings $\boldsymbol{x}, \boldsymbol{y} \in \Sigma^d$ of length $d$, there exists a comparison network of size $\mathcal{O}(d)$, width $\mathcal{O}(d)$, and depth $\mathcal{O}(\log d)$.*

We use this lemma to compute the complexity of the arhmetic circuit from Fig. 10.

**Proposition H.33.** *There exists an* $(\mathcal{O}((n + m)d), \mathcal{O}(nd \cdot (\log m + \log n)), \mathcal{O}(\log n(\log m + \log n \log d)), \mathcal{O}(nd))$-*arithmetic circuit*[14] *equivalent to* `pbs-key-values` *(Algorithm 10) with inputs $\boldsymbol{q}$ and $\boldsymbol{k}$ of lengths $n$ and $m$ with $d$-bit entries.*

*Proof.* The size of the circuit for `pbs-key-values` should equal the work-complexity of Algorithm 10 but we also need to account for the comparison gates in equation 14. Further, the circuit for binary search also has a size of $\mathcal{O}(d \cdot n)$ instead of $\mathcal{O}(d \cdot \log n)$ work as in Algorithm 10. Using Lemma H.32, along with the fact that the comparison gates and the binary search are used at most $\mathcal{O}(\log n)$ times, we deduce that we are adding $\mathcal{O}(nd \log n)$ size to the circuit in addition to the work complexity of the parallel algorithm. Thus, the overall size of the arithmetic circuit for `pbs-key-values` is $\mathcal{O}(dn \log m + nd \log n)$. Further, the depth of the circuit here is determined by the runtime of Algorithm 10 along with the depth of the comparison gates and binary search $\mathcal{O}(\log n \log d)$, and finally, at most $n$ processors are used for the parallel algorithm in Algorithm 10, which yields the width of the circuit. $\qquad\square$

**Circuit for `Parallel-MQAR`.** Now, we will call the circuit for `Parallel-MQAR` with the same name while the input gates contain the inputs of Algorithm 11. Indeed, we can directly "translate" Algorithm 11 to an arithmetic circuit as the values for $I_k^{\boldsymbol{x}}$ and $I_k^{\boldsymbol{x}}$ for each $\boldsymbol{x}$ and $k$ are predetermined from $N$. Thus, we start by placing the corresponding sorting networks which feeds into the `pbs-key-values` $(\boldsymbol{q}[s \cdots t],\ \boldsymbol{k}[x \cdots y], n, m)$ circuit for Algorithm 10 in Fig. 10 so that the output values from the calls to `pbs-key-values` result in the checks as in line 16 of Algorithm 11. That is, we get outputs $C_k[J_{\text{permuted}}^{k\boldsymbol{x}}(\text{dec}(i))]$ from each call to `pbs-key-values`. We can then use a comparison gate to check if this value equals $2^k$, and if not, we have found a match $C_k[J_{\text{permuted}}^{k\boldsymbol{x}}(\text{dec}(i))]$ for the query $\boldsymbol{q}_i$ which results in the output of the associated value $\boldsymbol{v}_{C_k[J_{\text{permuted}}^{k\boldsymbol{x}}(\text{dec}(i))]+1}$, exactly as in Algorithm 11. That is, we first define the following variable as the output of the comparison gate:

$$\boldsymbol{c}_{\text{dec}(i)}^{k} := \mathbb{1}\{C_k[J_{\text{permuted}}^{k\boldsymbol{x}}(\text{dec}(i))] \neq 2^k\}. \tag{18}$$

Here, as $C_k[J_{\text{permuted}}^{k\boldsymbol{x}}(\text{dec}(i))] \neq 2^k$ implies that $I_{\text{permuted}}^{k\boldsymbol{x}}\left(C_k[J_{\text{permuted}}^{k\boldsymbol{x}}(\text{dec}(i))]\right)$ equals the index of the matching key $\boldsymbol{k}_j$ corresponding to the query $\boldsymbol{q}_i$, the $i$th output is then simply given by $\boldsymbol{c}_{\text{dec}(i)}^{k} \cdot I_{\text{permuted}}^{k\boldsymbol{x}}\left(C_k[J_{\text{permuted}}^{k\boldsymbol{x}}(\text{dec}(i))]\right)$, where the 0 output implies that there does not exist a matching key.

Here, we also briefly introduce the the sorting networks that we use to sort the keys and queries:

**Definition H.34** (Informal). *Sorting networks are circuits with gates and wires where the gates of the circuit are comparators (Definition H.31) connecting two wires. Each such circuit can perform sorting on a fixed number of values.*

We can then show the circuit schematically as in Fig. 11.

We now dilineate the complexity of the circuit, starting with the complexity of the sorting networks.

**Lemma H.35** (Ajtai et al. (1983)). *Let $A$ be an array with $d$-bit entries of size $n$. Then, one can implement a sorting network to sort the array $A$ with size $\mathcal{O}(d \cdot n \log n)$ and depth $\mathcal{O}(\log d \log n)$.*

**Proposition H.36.** *There exists an* $(N \cdot d, \mathcal{O}(Nd \cdot \log^2 N), \mathcal{O}(\log d \log^2 N), \mathcal{O}(Nd \log N))$-*arithmetic circuit that solves the multiple-query associative recall problem.*

*Proof.* We note here that for each $k$, there are $\frac{\frac{N}{3}}{2^{k+1}}$ parallel problems of size $2^k$ for both the sorting networks and the `pbs − key − values` circuit. Using Lemma H.35, the cumulative size of these sorting networks is $\mathcal{O}(d \cdot N \log^2 N)$ (see equation 13) with overall depth $\mathcal{O}(\log d \log N)$.

Similarly, the next layer again runs $\sum_{k=0}^{\log \frac{N}{3} - 1} \frac{\frac{N}{3}}{2^{k+1}}$-many circuits for `pbs − key − values` each of which has size $\mathcal{O}(2^k d(\log 2^k + \log 2^k)) = O(d \cdot 2^k \log 2^k)$, depth $\mathcal{O}(\log^2 2^k \log d)$ and width $\mathcal{O}(2^k)$ (Proposition H.33). Again, the cumulative size of this layer is given by $\mathcal{O}(Nd \cdot \log^2 N)$ (see equation 13). Since we run each of these circuits in parallel, the depth of this layer is again $\mathcal{O}(\log d \log^2(N))$ while the width is $\mathcal{O}(N \cdot \log N)$.

Finally, we perform $\frac{N}{3} \log \frac{N}{3}$ comparisons at the end of $d$-bit strings in parallel which results in size $\mathcal{O}(N \log N \cdot d)$, depth $\mathcal{O}(\log d)$ and width $\mathcal{O}(N \log N \cdot d)$ (Lemma H.32). Therefore, the resulting

---

[14]Recall that a $(n, s, \Delta, w)$-arithmetic circuit is an $n$-variate circuit with size $s$, depth at most $\Delta$, and width $w$.

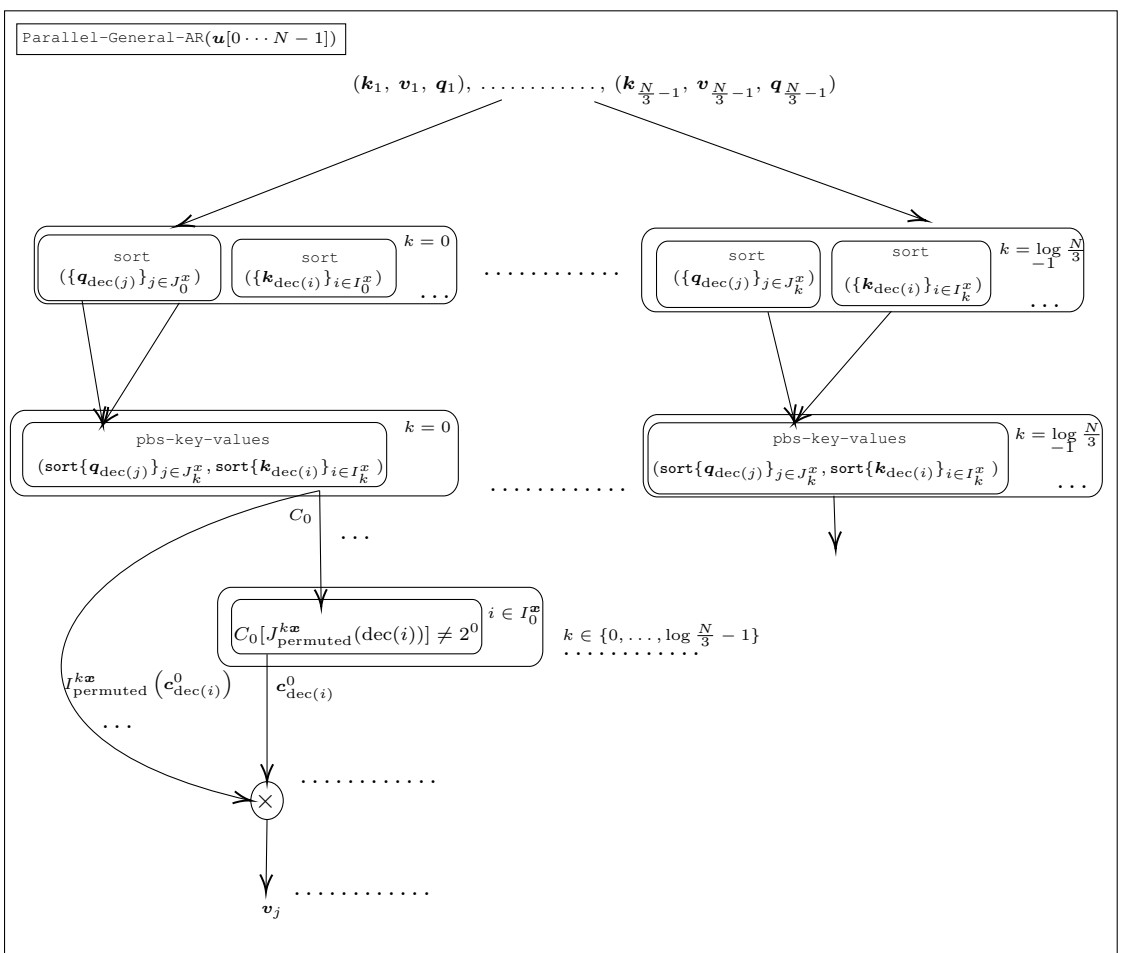

Figure 11: `Parallel-MQAR`($\boldsymbol{u}[0 \cdots N-1]$) as a circuit that includes sorting networks and the circuit for `pbs-key-values` as subroutines.

arithmetic circuit has size $\mathcal{O}(d \cdot N \log^2 N + Nd \cdot \log^2 N + N \log N \cdot d) = O(Nd \log^2 N)$, depth $\mathcal{O}(\log d \log^2 N)$ and width $\mathcal{O}(Nd \log N)$. □

### H.7.7 THE RESULTING BASECONV MODEL

As we have an arithemtic circuit for solving the multiple-query associative recall problem, we can now invoke Theorem H.21 to claim that there is a corresponding BASECONV model that solves the multiple-query associative recall problem with $\tilde{\mathcal{O}}(N \log c)$ parameters and $\tilde{\mathcal{O}}(1)$ layers.

**Theorem H.37.** *There exists a* $\left(Nd, \tilde{\mathcal{O}}(1), \tilde{\mathcal{O}}(1), \tilde{\mathcal{O}}(Nd), \tilde{\mathcal{O}}(1)\right)$ − BASECONV *solves the multiple-query associative recall problem.*

*Proof.* Directly applying Theorem H.21 yields a BASECONV model with the number of layers $\mathcal{O}(\log d \cdot \log^2 N \cdot \log Nd \log N) = \mathcal{O}(1)$ layers while the claim on the input and inner dimensions follow trivially. □

### H.8 DATA-DEPENDENT CONVOLUTIONS

### H.8.1 INTRODUCTION

In this section, we are again concerned with solving the multiple-query associative recall problem (Appendix H.7.1). However, in contrast to Appendix H.7.4, which yields a circuit that is unchanged and works for *all* inputs, we instead take the viewpoint of adapting the sequence mixing weights (Section 2) with respect to the particular sequence that the model receives as input. More specifically,

we take the distance between the tokens in the sequence as a measure for designing data-dependent convolutions.

**Setup.** To formally setup the problem, as in our discussion of designing a parallel algorithm, we consider the following problem description of the multiple-query associative recall problem.

Suppose we are given an input $\boldsymbol{u}[0 \cdots N - 1] \triangleq \left\{ (\boldsymbol{k}_0, \boldsymbol{v}_0, \boldsymbol{q}_0), \ldots, \left( \boldsymbol{k}_{\frac{N}{3}-1}, \boldsymbol{v}_{\frac{N}{3}-1}, \boldsymbol{q}_{\frac{N}{3}-1} \right) \right\}$ with each $\boldsymbol{k}_i, \boldsymbol{v}_i, \boldsymbol{q}_i \in C$. Here, each token is embedded using the standard one-hot encoding in $\{0, 1\}^c$ (i.e. we assume $d = c$).[15] Our goal is again to check, for each $1 \leq i \leq \frac{N}{3} - 1$, whether there exists $0 \leq j < i$ such that $\boldsymbol{q}_i \equiv \boldsymbol{k}_j$, and if so, output $\boldsymbol{v}_j$.
Here, we define the interaction distance between the $i$th query $\boldsymbol{q}_i$ and the matching key $\boldsymbol{k}_j$ as $i - j$. We then also assume that *number of distinct interaction distances* is bounded by $t$.

We can identify these distances using an autocorrelation ([Chatfield, 1995](#)), which has an elegant underlying formulation. We will briefly introduce the relevant mathematical machinery in the context of elucidating the data-dependent model that we seek to develop in the sequel.

**Autocorrelations** We introduce autocorrelation convolutions. Let $\tilde{\boldsymbol{u}}[t] := \boldsymbol{u}[-t]$, then the cross correlation of two vectors $\boldsymbol{u}$ and $\boldsymbol{v}$ is given by

$$\boldsymbol{u} \star \boldsymbol{v} \triangleq \tilde{\boldsymbol{u}} * \boldsymbol{v}.$$

The *autocorrelation* of a vector $\boldsymbol{u} \in \mathbb{R}^n$ is the cross correlation of $\boldsymbol{u}$ with itself. Moreover, in terms of polynomials, we have $\tilde{\boldsymbol{u}}(X) = X^{n-1} \cdot \boldsymbol{u}(1/X)$. Thus, in analogy with our interpretation of convolution in terms of polynomial multiplication, we characterize the autocorrelation of a vector $\boldsymbol{u} \in \mathbb{R}^n$, given by $\mathbf{w} \in \mathbb{R}^n$ as follows:

$$\mathbf{w} = \text{coeff}\left( \boldsymbol{u}(X) \cdot \tilde{\boldsymbol{u}}(X) \mod X^n - 1 \right).$$

### H.8.2 BASECONV WITH KERNELS GENERATED USING AUTO-CORRELATION

We are now ready to describe the model that solves the multiple-query associative recall problem using data-dependent kernels derived using auto-correlations.

**The Input-Dependent Kernels.** There are several potential strategies to identify the best token-interaction distances for an input using a convolution. Here we will focus on an autocorrelation convolution for exposition. Autocorrelation allows us to identify the top $t$ distinct shifts of the sequence that result in highly overlapping values (e.g. matches between query and keys in our associative recall setting). We can then construct convolution filters that perform each of these $t$ shifts. That is, we define a function Top such that $\text{Top}(u \star u, t)$ returns a list of the top $t$ shifts $\{s_\ell\}_{\ell \in [t]}$. We then use these top $t$ distances $\{s_\ell\}_{\ell \in [t]}$ to define the following two kernels:

$$\begin{aligned}
\mathbf{h}^k(X) &\equiv \sum_{\ell \in [t]} X^{s_\ell + (\ell-1) \cdot \frac{N}{3}}, \\
\mathbf{h}^v(X) &\equiv \sum_{\ell \in [t]} X^{s_\ell - 1 + (\ell-1) \cdot \frac{N}{3}}.
\end{aligned} \tag{19}$$

Here, we note that we only only have two kernels as we will assume $N' = t\frac{N}{3}$ and the shift will be done in "parallel." Obviously, one can instead define $t$ distinct shift kernels but then there is a cost of $\mathcal{O}(t)$ in the number of layers.

**Projections.** We define the following projections $\mathbf{K}, \mathbf{Q}, \mathbf{V} \in \{0, 1\}^{N \times d}$ of the input that we shall use below using equation [9](#).

$$\mathbf{K}[i, :] := \begin{cases} \boldsymbol{u}[i, :] & \text{if } i \in \mathcal{K}, \\ \mathbf{0}^d & \text{otherwise} \end{cases},$$

$$\mathbf{Q}[i, :] := \begin{cases} \boldsymbol{u}[i, :] & \text{if } i \in \mathcal{Q}, \\ \mathbf{0}^d & \text{otherwise} \end{cases}, \tag{20}$$

$$\mathbf{V}[i, :] := \begin{cases} \boldsymbol{u}[i, :] & \text{if } i \in \mathcal{V}, \\ \mathbf{0}^d & \text{otherwise} \end{cases},$$

---

[15]Our arguments do need $c = d$. However we do not need $d = N$ but we made this simplification for ease of presentation.

Finally, we present the BASECONV model that solves the multiple-query associative recall problem with input-dependent kernels using $\mathcal{O}(1)$ layer and $\mathcal{O}(t \cdot Nd)$-many parameters.

**Theorem H.38.** *There exists a BASECONV model with gated data-dependent convolutions that solves the multiple-query associative recall problem on inputs from $\{0,1\}^{3N \times c}$ with the total number of distinct interaction distances bounded by $t$ in $\mathcal{O}(1)$ layers and $\mathcal{O}(t \cdot Nc)$ total parameters.*

*Proof.* We note that we have the input dimension $d = c$. Now, for any input sequence $\boldsymbol{u} \in \{0,1\}^{N \times d}$, we get the input-dependent kernels as in equation 19 using autocorrelation of the input. We will now outline the following computations for the BASECONV layers:

$$\boldsymbol{y} = \texttt{Linear}_{\mathbf{Q}}(\boldsymbol{u}) \odot \left( \mathbf{h}^K * \texttt{Linear}_{\mathbf{K}}(\boldsymbol{u}) \right)$$
$$= \mathbf{Q} \odot \left( \mathbf{h}^K * \mathbf{K} \right) \tag{21}$$
$$\boldsymbol{z} = \texttt{Linear}_{\mathbf{E}}(\boldsymbol{y}) \odot \left( \mathbf{h}^V * \texttt{Linear}_{\mathbf{V}}(\boldsymbol{u}) \right)$$
$$= \mathbf{E} \odot \left( \mathbf{h}^V * \mathbf{V} \right), \tag{22}$$

where we have the linear projections $\texttt{Linear}_{\mathbf{Q}}(\boldsymbol{u}) = \mathbf{Q}$, $\texttt{Linear}_{\mathbf{K}}(\boldsymbol{u}) = \mathbf{K}$, $\texttt{Linear}_{\mathbf{V}}(\boldsymbol{u}) = \mathbf{V}$ and $\texttt{Linear}_{\mathbf{E}}(\boldsymbol{y}) = \mathbf{E}$ defined as

$$\mathbf{E}[i,:] := \texttt{Linear}_{\mathbf{E}}(\boldsymbol{y})[i,:] = \begin{cases} \mathbf{1}^d & \text{if } \exists\, j \in [d] \text{ such that } \boldsymbol{y}[i,j] = 1 \\ \mathbf{0}^d & \text{otherwise} \end{cases}. \tag{23}$$

Here, we will first present the argument for the special case when we have $t = 1$ as that will help us elucidate the general case. To this end, as the kernels from equation 19 for $t = 1$ are given by

$$\mathbf{h}^k(X) \equiv X^{s_1};$$
$$\mathbf{h}^v(X) \equiv X^{s_1 - 1}, \tag{24}$$

we observe that convolving with these kernels $\mathbf{h} * \boldsymbol{y}$ is equivalent to operating with the following primitives (Appendix H.3):

$$\texttt{shift\_down}(\boldsymbol{y}, s_1);$$
$$\texttt{shift\_down}(\boldsymbol{y}, s_1 - 1). \tag{25}$$

We note that we shift down instead of shifting up as the index of the top-left entry is $(0,0)$. We can then write down the computations performed in equation 21 and equation 22 as follows:

$$\boldsymbol{y} = \mathbf{Q} \odot \texttt{shift\_down}(\mathbf{K}, s_1) \tag{26}$$
$$\boldsymbol{z} = \mathbf{E} \odot \texttt{shift\_down}(\mathbf{V}, s_1 - 1), \tag{27}$$

We begin by examining $\boldsymbol{y}$ below:

$$\boldsymbol{y}[i,:] = \left( \mathbf{Q} \odot \texttt{shift\_down}(\mathbf{K}, s_1) \right)[i,:]$$
$$= \mathbf{Q}[i,:] \odot \texttt{shift\_down}(\mathbf{K}, s_1)[i,:] \tag{28}$$
$$= \left( \begin{cases} \boldsymbol{u}[i,:] & \text{if } i \in \mathcal{Q} \\ \mathbf{0}^d & \text{otherwise} \end{cases} \right) \odot \left( \begin{cases} \boldsymbol{u}[i - s_1,:] & \text{if } i - s_1 \in \mathcal{K}, \\ \mathbf{0}^d & \text{otherwise} \end{cases} \right) \tag{29}$$
$$= \begin{cases} \boldsymbol{u}[i,:] \odot \boldsymbol{u}[i - s_1,:] & \text{if } i \in \mathcal{Q} \text{ and } i - s_1 \in \mathcal{K}, \\ \mathbf{0}^d & \text{otherwise} \end{cases}$$

Here, we use the fact that the Hadamard product is row-independent in equation 28, and the definitions of the projections from equation 20 in equation 29. Examining the $j$th entry, we get

$$\boldsymbol{u}[i,j] \odot \boldsymbol{u}[i - s_1, j] = \begin{cases} 1 & \text{if } i \in \mathcal{Q}, i - s_1 \in \mathcal{K} \text{ and } \boldsymbol{q}_i \equiv \boldsymbol{k}_{i-s_1} \equiv \mathbf{e}_j \\ 0 & \text{otherwise}. \end{cases}$$

That is, we can express

$$\boldsymbol{y}[i,:] = \begin{cases} \mathbf{e}_j & \text{if } i \in \mathcal{Q}, i - s_1 \in \mathcal{K} \text{ and } \boldsymbol{q}_i \equiv \boldsymbol{k}_{i-s_1} \equiv \mathbf{e}_j \\ \mathbf{0}^d & \text{otherwise} \end{cases}. \tag{30}$$

Consequently, as per the definition in equation 23, we get

$$\mathbf{E}[i,:] = \begin{cases} \mathbf{1}^d & \text{if } i \in \mathcal{Q}, i - s_1 \in \mathcal{K} \text{ and } \boldsymbol{q}_i \equiv \boldsymbol{k}_{i-s_1} \\ \mathbf{0}^d & \text{otherwise} \end{cases} \tag{31}$$

We can now finally specify the output $z$ from equation 27 as follows:

$$
\begin{aligned}
z[i,:] &= (\mathbf{E} \odot \texttt{shift\_down}(\mathbf{V}, s_1 - 1))\,[i,:] \\
&= \mathbf{E}[i,:] \odot \texttt{shift\_down}(\mathbf{V}, s_1 - 1)[i,:] \tag{32} \\
&= \left( \begin{cases} \mathbf{1}^d & \text{if } i \in \mathcal{Q}, i - s_1 \in \mathcal{K} \text{ and } \boldsymbol{q}_i \equiv \boldsymbol{k}_{i-s_1} \\ \mathbf{0}^d & \text{otherwise} \end{cases} \right) \odot \left( \begin{cases} \boldsymbol{u}[i - s_1 + 1, :] & \text{if } i - s_1 + 1 \in \mathcal{V}, \\ \mathbf{0}^d & \text{otherwise} \end{cases} \right) \\
&\tag{33} \\
&= \begin{cases} \boldsymbol{u}[i - s_1 + 1, :] & \text{if } i \in \mathcal{Q}, i - s_1 \in \mathcal{K}, i - s_1 + 1 \in \mathcal{V} \text{ and } \boldsymbol{q}_i \equiv \boldsymbol{k}_{i-s_1} \\ \mathbf{0}^d & \text{otherwise} \end{cases} \\
&= \begin{cases} \boldsymbol{v}_{i-s_1} & \text{if } \boldsymbol{q}_i \equiv \boldsymbol{k}_{i-s_1} \\ \mathbf{0}^d & \text{otherwise} \end{cases} \tag{34}
\end{aligned}
$$

Again, we use the fact that the Hadamard product is row-independent in equation 32, and the definitions of the projections from equation 20 in equation 33. Overall, we have solved associative recall for all queries that have interaction distance exactly equal to $s_1$.

In order to generalize this to arbitrary $t \leq \frac{N}{3}$, we first increase the internal dimension so that the input to the kernels $\boldsymbol{u}' \in \mathbb{R}^{(N \cdot t) \times d}$ in equation 19 and the projections $\mathbf{K}', \mathbf{Q}', \mathbf{V}' \in \mathbb{R}^{(N \cdot t) \times d}$ are given by

$$
\boldsymbol{u}' \equiv \begin{pmatrix} \mathbf{0}^d \\ \vdots \\ \mathbf{0}^d \\ \boldsymbol{u} \end{pmatrix}, \mathbf{K}' \equiv \begin{pmatrix} \mathbf{0}^d \\ \vdots \\ \mathbf{0}^d \\ \mathbf{K} \end{pmatrix}, \mathbf{Q}' \equiv \begin{pmatrix} \mathbf{Q} \\ \vdots \\ \mathbf{Q} \\ \mathbf{Q} \end{pmatrix}, \mathbf{V}' \equiv \begin{pmatrix} \mathbf{0}^d \\ \vdots \\ \mathbf{0}^d \\ \mathbf{V} \end{pmatrix},
$$

We then observe that for $\mathbf{h}_\ell^k(X) := X^{s_\ell}$ and $\mathbf{h}_\ell^v(X) := X^{s_\ell - 1}$, we have

$$
\mathbf{h}^k(X) \equiv \sum_{\ell \in [t]} \mathbf{h}_\ell^k(X) \cdot X^{(\ell - 1) \cdot \frac{N}{3}},
$$

$$
\mathbf{h}^v(X) \equiv \sum_{\ell \in [t]} \mathbf{h}_\ell^k(X) \cdot X^{(\ell - 1) \cdot \frac{N}{3}}.
$$

In analogy with equation 25, we can then equivalently write

$$
\left(\mathbf{h}^K * \mathbf{K}\right) \equiv
\begin{pmatrix}
\mathbf{h}_t^K \\
\vdots \\
\mathbf{h}_2^K \\
\mathbf{h}_1^K
\end{pmatrix}
*
\begin{pmatrix}
\mathbf{0}^d \\
\vdots \\
\mathbf{0}^d \\
\mathbf{K}
\end{pmatrix}
$$

$$
\equiv
\begin{pmatrix}
\mathbf{h}_t^K * \mathbf{K} \\
\vdots \\
\mathbf{h}_2^K * \mathbf{K} \\
\mathbf{h}_1^K * \mathbf{K}
\end{pmatrix}
$$

$$
\equiv
\begin{pmatrix}
\texttt{shift\_down}(\mathbf{K}, s_t) \\
\vdots \\
\texttt{shift\_down}(\mathbf{K}, s_2) \\
\texttt{shift\_down}(\mathbf{K}, s_1)
\end{pmatrix} .
$$

Similarly, we also have

$$
\left(\mathbf{h}^V * \mathbf{V}'\right) \equiv
\begin{pmatrix}
\texttt{shift\_down}(\mathbf{V}, s_t - 1) \\
\vdots \\
\texttt{shift\_down}(\mathbf{V}, s_2 - 1) \\
\texttt{shift\_down}(\mathbf{V}, s_1 - 1)
\end{pmatrix} .
$$

That is, the argument for $t = 1$ now applies to each of the $t$ shifts as we now have (*cf.* equation 21)

$$
\begin{aligned}
\boldsymbol{y}' &\equiv \mathbf{Q}' \odot \left( \mathbf{h}^V * \mathbf{V} \right) \\
&\equiv \begin{pmatrix} \mathbf{Q} \\ \vdots \\ \mathbf{Q} \\ \mathbf{Q} \end{pmatrix} \odot \begin{pmatrix} \texttt{shift\_down}(\mathbf{V}, s_t - 1) \\ \vdots \\ \texttt{shift\_down}(\mathbf{V}, s_2 - 1) \\ \texttt{shift\_down}(\mathbf{V}, s_1 - 1) \end{pmatrix} \\
&\equiv \begin{pmatrix} \mathbf{Q} \odot \texttt{shift\_down}(\mathbf{V}, s_t - 1) \\ \vdots \\ \mathbf{Q} \odot \texttt{shift\_down}(\mathbf{V}, s_2 - 1) \\ \mathbf{Q} \odot \texttt{shift\_down}(\mathbf{V}, s_1 - 1) \end{pmatrix} \\
&\equiv \begin{pmatrix} \boldsymbol{y}_t \\ \vdots \\ \boldsymbol{y}_2 \\ \boldsymbol{y}_1 \end{pmatrix},
\end{aligned}
$$

where, for each $\ell \in [t]$, we have (*cf.* equation 30)

$$
\boldsymbol{y}_\ell[i, :] \equiv \begin{cases} \mathbf{e}_j & \text{if } i \in \mathcal{Q}, i - s_\ell \in \mathcal{K} \text{ and } \boldsymbol{q}_i \equiv \boldsymbol{k}_{i-s_\ell} \equiv \mathbf{e}_j, \\ \mathbf{0}^d & \text{otherwise} \end{cases}.
$$

We then analogously get $\mathbf{E}'$ as follows:

$$
\mathbf{E}' \equiv \texttt{Linear}_{\mathbf{E}}(\boldsymbol{y}') \equiv \begin{pmatrix} \texttt{Linear}_{\mathbf{E}}(\boldsymbol{y}_t) \\ \vdots \\ \texttt{Linear}_{\mathbf{E}}(\boldsymbol{y}_2) \\ \texttt{Linear}_{\mathbf{E}}(\boldsymbol{y}_1) \end{pmatrix} \equiv \begin{pmatrix} \mathbf{E}_t \\ \vdots \\ \mathbf{E}_2 \\ \mathbf{E}_1 \end{pmatrix},
$$

where, for each $\ell \in [t]$, we have (*cf.* equation 31)

$$
\mathbf{E}_\ell[i, :] = \begin{cases} \mathbf{1}^d & \text{if } i \in \mathcal{Q}, i - s_\ell \in \mathcal{K} \text{ and } \boldsymbol{q}_i \equiv \boldsymbol{k}_{i-s_\ell} \\ \mathbf{0}^d & \text{otherwise} \end{cases}
$$

The output in the general case is then given by

$$z' \equiv \mathbf{E}' \odot \left( \mathbf{h}^V * \mathbf{V}' \right)$$

$$\equiv \begin{pmatrix} \mathbf{E}_t \\ \vdots \\ \mathbf{E}_2 \\ \mathbf{E}_1 \end{pmatrix} \odot \begin{pmatrix} \texttt{shift\_down}(\mathbf{V}, s_t - 1) \\ \vdots \\ \texttt{shift\_down}(\mathbf{V}, s_2 - 1) \\ \texttt{shift\_down}(\mathbf{V}, s_1 - 1) \end{pmatrix}$$

$$\equiv \begin{pmatrix} \mathbf{E}_t \odot \texttt{shift\_down}(\mathbf{V}, s_t - 1) \\ \vdots \\ \mathbf{E}_2 \odot \texttt{shift\_down}(\mathbf{V}, s_2 - 1) \\ \mathbf{E}_1 \odot \texttt{shift\_down}(\mathbf{V}, s_1 - 1) \end{pmatrix}$$

$$\equiv \begin{pmatrix} z_t \\ \vdots \\ z_2 \\ z_1 \end{pmatrix},$$

where, for each $\ell \in [t]$, we have (*cf.* equation 34)

$$z_\ell[i, :] \equiv \begin{cases} v_{i-s_\ell} & \text{if } q_i \equiv k_{i-s_\ell} \\ \mathbf{0}^d & \text{otherwise} \end{cases}$$

Finally, we define the last output layer to compute $z_{out} \equiv \texttt{Linear}_{\text{sum}}(z') \equiv \sum_{\ell \in [t]} z_\ell$ so that we have

$$z_{out}[i, :] \equiv \begin{cases} v_{i-s_\ell} & \text{if } q_i \equiv k_{i-s_\ell} \text{ for some } \ell \in [t] \\ \mathbf{0}^d & \text{otherwise} \end{cases}$$

To recall, we retrieved the top $t$ interaction distances of the input $u$ using auto-correlation and defined the corresponding convolution kernels (equation 19). We then shifted djown the keys $\mathbf{K}$ using the first kernel and gated with the corresponding queries $\mathbf{Q}$ so that we got a match exactly when there exists a key that is at $s_\ell$ interaction distance from the corresponding query. After "smearing" this match to get $\mathbf{E}$, we used it as a mask to retrieve the value in the next layer. Overall, since we have $t$ atomic kernels that perform $t$ shifts with each of these kernels using $\mathcal{O}(Nd)$ parameters, we can conclude that the output solves the associative recall problem for all queries with exactly $\ell$ interaction distance from the corresponding keys for all $\ell \in [t]$ using $\mathcal{O}(1)$ layers and $\mathcal{O}(t \cdot Nc)$ parameters as we have $d = c$. $\qquad \square$

Table 7: Attention Training Settings

|  | 73M | 125M | 360M |
|---|---|---|---|
| Optimizer | | Adam | |
| Optimizer momentum | | $\beta_1, \beta_2 = 0.9, 0.95$ | |
| Optimizer eps | | $1e - 8$ | |
| Precision | | BFloat16 | |
| Learning rate decay | | Cosine | |
| Learning rate (min, base) | | 8e-5, 8e-4 | |
| Global batch size | | 256 | |
| Training iterations | | 20000 | |
| Warmup Duration (Linear) | | 0.01 | |
| Weight decay | | 0.1 | |
| Num Layers | 6 | 12 | 24 |
| Hidden Size | 704 | 768 | 1024 |
| FFN Width | | 4 | |
| Position Embeddings | | Rotary | |
| Weight Tying | | True | |
| Number of Heads ($H$) | 8 | 12 | 16 |

Table 8: Attention FLOPs Computation

|  | Equation |
|---|---|
| Input Layer | $B \times V \times N \times D$ |
| Sequence Mixer $\mathbf{Q}, \mathbf{K}, \mathbf{V}$ Projections | $B \times N \times D \times D \times 3$ |
| Sequence Mixer Attention | $B \times H \times H \times D + H \times N \times N + B \times N \times N \times D$ |
| Sequence Mixer Output Projection | $B \times N \times D \times D$ |
| Channel Mixer (FFN Width 4) | $B \times D \times D \times 8 \times \frac{2}{3} \times N$ |
| Language Modeling Head | $B \times V \times N \times D$ |

Table 9: BASECONV Training Settings

|  | 168M | 354M |
|---|---|---|
| Optimizer | Adam | |
| Optimizer momentum | $\beta_1, \beta_2 = 0.9, 0.95$ | |
| Precision | BFloat16 | |
| Learning rate decay | Cosine | |
| Learning rate (min, base) | 8e-5, 8e-4 | |
| Global batch size | 256 | |
| Training iterations | 20000 | |
| Warmup Duration (Linear) | 0.01 | |
| Weight decay | 0.1 | |
| Num Layers | 30 | 48 |
| Hidden Size | 852 | 1080 |
| FFN Width | 2 | |
| Position Embeddings | - | |
| Weight Tying | True | |
| Short Conv. Filter Size | 3 | |
| Exp. Mod. Decay (Fast, Slow) | 0.3, 1.2 | |
| Filter Sine Freq. (w) | 14 | |
| Filter Order | 64 | |
| Filter Inner MLP | 2 | |
| Filter Weight Decay | 0 | |

Table 10: BASECONV FLOPs Computation

|  | Equation |
|---|---|
| Input Layer | $B \times V \times N \times D$ |
| Sequence Mixer Long Convolution | $10 \times N \times \log(N) \times 0.5(D) \times B$ |
| Sequence Mixer Short Convolution | $B \times N \times 0.5(D)$ |
| Sequence Mixer Implicit MLP (Order 64) | $0.5(D) \times 64$ |
| Sequence Mixer Linear Projection | $B \times N \times D \times D$ |
| Channel Mixer (FFN Width 2) | $B \times D \times D \times 2 \times 2 \times N$ |
| Language Modeling Head | $B \times V \times N \times D$ |

Table 11: Hyena (Poli et al., 2023a) Training Settings

|  | 72M | 158M | 358M |
|---|---|---|---|
| Optimizer | | Adam | |
| Optimizer momentum | | $\beta_1, \beta_2 = 0.9, 0.98$ | |
| Precision | | BFloat16 | |
| Learning rate decay | | Cosine | |
| Learning rate (min, base) | | (1e-5, 8e-4) | |
| Global batch size | | 256 | |
| Training iterations | | 20000 | |
| Warmup Duration (Linear) | | 0.01 | |
| Weight decay | | 0.1 | |
| Num Layers | 8 | 18 | 24 |
| Hidden Size | 768 | 864 | 1024 |
| FFN Width | | 2 | |
| Position Embeddings | | None | |
| Weight Tying | | True | |
| Short Conv. Filter Size | | 3 | |
| Exp. Mod. Decay (Fast, Slow) | | 0.3, 1.2 | |
| Filter Sine Freq. (w) | | 14 | |
| Filter Order | | 64 | |
| Filter Inner MLP | | 2 | |
| Filter Weight Decay | | 0 | |

Table 12: Hyena FLOPs Computation

|  | Equation |
|---|---|
| Input Layer | $B \times V \times N \times D$ |
| Sequence Mixer Input Projection | $B \times N \times D \times D \times 3 + B \times N \times 9 \times D$ |
| Sequence Mixer Long Convolution | $10 \times N \times \log(N) \times D \times B$ |
| Sequence Mixer Short Convolution | $3 \times B \times N \times D$ |
| Sequence Mixer Implicit MLP (Order 64) | $D \times 64$ |
| Sequence Mixer Output Projection | $B \times N \times D \times D$ |
| Channel Mixer (FFN Width 2) | $B \times D \times D \times 2 \times 2 \times N$ |
| Language Modeling Head | $B \times V \times N \times D$ |

Table 13: H3 (Fu et al., 2023c) Training Settings

|  | 72M | 168M | 357M |
|---|---|---|---|
| Optimizer | | Adam | |
| Optimizer momentum | | $\beta_1, \beta_2 = 0.9, 0.95$ | |
| Precision | | BFloat16 | |
| Learning rate decay | | Cosine | |
| Learning rate (min, base) | | (8e-5, 8e-4) | |
| Global batch size | | 256 | |
| Training iterations | | 20000 | |
| Warmup Duration (Linear) | | 0.01 | |
| Weight decay | | 0.1 | |
| Weight Tying | | True | |
| Num Layers | 8 | 19 | 33 |
| Hidden Size | 720 | 864 | 1024 |
| FFN Width | | 2 | |
| Position Embeddings | | None | |
| State Space Model State Size | | 64 | |

Table 14: RWKV Peng et al. (2023) Training Settings

|  | 72M | 169M | 351M |
|---|---|---|---|
| Optimizer | | Adam | |
| Optimizer momentum | | $\beta_1, \beta_2 = 0.9, 0.999$ | |
| Optimizer eps | | $1e-8$ | |
| Precision | | BFloat16 | |
| Learning rate decay | | Cosine | |
| Learning rate (min, base) | | 1e-5, 8e-4 | |
| Global batch size | | 256 | |
| Training iterations | | 20000 | |
| Warmup Duration (Linear) | | 0.01 | |
| Weight decay | | 0.1 | |
| Weight Tying | | False | |
| Num Layers | 6 | 12 | 20 |
| Hidden Size | 624 | 768 | 984 |
| Position Embeddings | | None | |
| Initialization | | From Reference Impl. | |

Table 15: RetNet (Sun et al., 2023) Training Settings

|  | 152M |
| --- | --- |
| Optimizer | Adam |
| Optimizer momentum | $\beta_1, \beta_2 = 0.9, 0.98$ |
| Optimizer eps | 1.0e-8 |
| Precision | BFloat16 |
| Learning rate decay | Linear |
| Learning rate (min, base) | (1e-5, 8e-4) |
| Global batch size | 256 |
| Training iterations | 20000 |
| Warmup Duration (Linear) | 0.01 |
| Weight decay | 0.05 |
| Normalization | Layernorm |
| Weight Tying | True |
| Num Layers | 12 |
| Hidden Size | 768 |
| Value Hidden Size | 1280 |
| Window Size | 128 |
| Position Embeddings | xPos (relative position embeddings) |
| Initialization | DeepNet |

Table 16: Simple Long Convolution Fu et al. (2023c) Training Settings

|  | 76M | 128M | 360M |
| --- | --- | --- | --- |
| Optimizer |  | Adam |  |
| Optimizer momentum |  | $\beta_1, \beta_2 = 0.9, 0.95$ |  |
| Precision |  | BFloat16 |  |
| Learning rate decay |  | Linear |  |
| Learning rate (min, base) |  | 8e-5, 8e-4 |  |
| Global batch size |  | 256 |  |
| Training iterations |  | 20000 |  |
| Warmup Duration (Linear) |  | 0.01 |  |
| Weight decay |  | 0.1 |  |
| Num Layers | 6 | 12 | 24 |
| Hidden Size | 704 | 864 | 1024 |
| FFN Width |  | 4 |  |
| Position Embeddings |  | - |  |
| Weight Tying |  | True |  |
| Channels |  | 1 |  |
| Lam |  | 0.001 |  |
| Kernel Dropout |  | 0.1 |  |
| Kernel LR |  | $5e-5$ |  |
| Activation |  | GeLU |  |
| Exponential Modulation |  | True |  |

