# OpenReview forum: "Zoology: Measuring and Improving  Recall in Efficient Language Models"
_ICLR.cc/2024/Conference — ICLR 2024 poster_

### Official Review · Reviewer_5EW4 · 2023-10-31

**Soundness:** 4 excellent
**Presentation:** 3 good
**Contribution:** 4 excellent
**Rating:** 8
**Confidence:** 4

**Summary:**

The authors systematically study the impact of neural architecture on language modeling performance. The authors identify a persisting quality gap between convolution and attention networks. Specifically, they identify a single failure mode, i.e., multi-query associative recall (MQAR), and demonstrate that convolution networks fall short in this. To verify the impact of this gap, the authors conduct systematic studies and provide both empirical and theoretical evidence. Moreover, the authors present strategies for migrating this gap.

**Strengths:**

The studied problem is important and may have a big impact. The pinpointed failure model (i.e., associative recall) is novel and reasonable. Both empirical and theoretical studies are conducted to support the argument. To further demonstrate the impact of the analyses, the authors examine two alternative strategies, which support the intuition of the author.

**Weaknesses:**

The proposed attention hybrid method seems to perform well in the experiment. However, it is not clear how it would perform on a larger scale.

**Questions:**

As to the training stability and the sensitivity to the training hyper-parameters, I'm wondering how the proposed look-up method and the hybrid method compare with the attention network.

---

> ### Author Response · Authors · 2023-11-18
> **Response to Reviewer 5EW4**
>
> Thank you for your thoughtful review! We appreciated that you found the work novel and insightful, and the problem impactful as gated convolution sequence models are receiving increasing adoption in the ML community. Here we address your questions.
>
>
> **Presentation**
>
> We take note the presentation score of 3 and have taken several steps to improve the overall delivery of our work. We hope these changes are helpful! The changes include:
>
> 1. Section 3 and Appendix C.1: We update the sections to describe how associative recall quality is measured in the real language modeling data. We also clarify the definitions and add simple examples of prior vs. our proposed recall problem.
> 2. Theory: We thoroughly checked for consistency with respect to notation and further clarified the assumptions underlying our theoretical results.
> 3. Section 5.1: We provide clarification of the input-dependent architectures we evaluate.
> 4. Appendix A: We include a new detailed related works section that discusses the history of associative recall in machine learning and an extensive list of previously proposed architectures to better contextualize our work.
> 5. Appendix B: We provide a simple PyTorch implementation of the BaseConv architecture to complement the theoretical statements.
> 6. Appendix E: We provide a new section and algorithm box to document exactly how the MQAR synthetic data is constructed. We also provide more details on our setup for the synthetic experiments.
>
> Please let us know if you have any additional suggestions on how we can improve the presentation and thank you for your help!
>
> **Scaling architectures.**
>
> Thank you for your question on scaling performance! In response, we have conducted three new experiments to better understand the scaling trends. We kindly direct you to the discussion of these experiments that is included in the main response to all reviewers as well as Appendix G of the revised submission.
>
> In these experiments, we show that simply scaling the gated convolution architectures does not solve the MQAR problem. We scale up to 7B parameter models in our new experiments. We hope these experiments build confidence that hybrid solutions continue to help at scale since attention layers can solve MQAR, complementing the gated-convolution backbone which struggles to solve MQAR efficiently.
>
> **Sensitivity to hyperparameters.**
>
> You raised the important question of how the training stability and sensitivity of the selective attention methods compare to the baseline attention network. We do not apply any hyperparameter tuning (e.g., LR, Optimizer, Batch Sizes, etc.) when producing these results. We simply use the backbone gated convolution models with hyperparameters provided by the prior work and replace a few layers with the hybrid layer. Overall, training is stable and the hybrid models work even without tuning!
>
> Further, to back up these results we are including our training code in the revised submission. We will also release the pretrained checkpoints upon publication. Please let us know if you have further questions about this!

---

### Official Review · Reviewer_xdZW · 2023-11-02

**Soundness:** 2 fair
**Presentation:** 1 poor
**Contribution:** 2 fair
**Rating:** 3
**Confidence:** 3

**Summary:**

This paper demonstrates that convolutional LMs are worse at bigram associative recall (AR) than transformers on real data by demonstrating that the test AR perplexity is sensitive to the number of occurences of the given bigrams in the training set. This possibly suggests that convolutional LMs are worse at in-context learning than previously thought in other works that test on synthetic tasks. This work also proposes a new synthetic task called a multiquery associative recall (MQAR), and uses new theoretical insights to devise a simple convolution-only alternative competitive to attention on AR.

**Strengths:**

Using the bigram frequency and the test perplexity of real data was insightful.

**Weaknesses:**

There's a lot of typos and confusing writing. It's hard to properly understand all the theoretical claims of the work. For starters, Proposition 4.3 should say `u \in \{ 0, 1 \}^{N \times \log c}`. The description of definition 3.1 could be improved; a simple example would be quite helpful. In appendix, `N` sometimes means the entire sequence length or the number of triplets in MQAR; this confusion is exacerbated by the fact that the meaning changes in the same theorem/proof.

Page 6 states "[e]ach BaseConv layer uses O(Nd) parameters and ... O(Nd) operations". I believe it uses `O(Nd +d^2)` parameters and `O(d N log N + Nd^2)` FLOPs? This makes me believe that the rest of the theoretical results may have to be carefully revisited by the authors.

Page 26. Proof of C.19. It's unclear how `Q[i, :]` can be set to a zero vector when `i \notin Q` (also boldface 0 suffices to express a vector; no need for a superscript d; also, d is defined as `log(C)` in the same page but `C` is a set. `d = log(c)` small c since `c := |C|`.) and similarly for `K` and `V`, because QKV is actually a linear projection of the input. Linear projection can not implement this non-linear operation of masking some of the activations out. There's a sentence that reads `... where Q, K, V ... are positional embeddings ...`. They are different projections of u.

The calculation of the percentage of gap due to AR hits could be better motivated and justified in appendix.

Other minor typos:
 - `u * k = FFT^{-1} (FFT(u) FFT(k))` (should drop the convolution operation in frequency domain).
 - Page 5: Other tokens: ... `1,000` => `1250`.

**Questions:**

(wrote under weaknesses)

---

> ### Author Response · Authors · 2023-11-18
> **Response to Reviewer xdZW**
>
> Thank you for your detailed review! The feedback led to significant improvements in our work. We appreciate that you found the downstream analysis insightful and here we address your questions.
>
> **Theoretical corrections, clarifications, and updates**
>
> Thank you for raising questions about the theoretical pieces. We apologize for the confusion from our original submission and have thoroughly examined the revised submission.
>
> _Page 26. Proof of C.19. QKV matrices and dimensions. – Now page 47, Appendix H.7._
>
> We have added clarifications on how we assumed the input u to be encoded for the proof (we have made this explicit as well as added some justification on why the encoding assumption is reasonable). The proof is updated where Q, K and V are linear projections. While fixing the argument we realized that we needed d=3c so we have made that correction as well.
>
> _Parameters and operations for BaseConv layers._
>
> We had proven that we only need $\tilde{O}(d)$ parameters for the projections to achieve the tight general arithmetic circuit result in Appendix C.4 of our original submission (now Appendix H.5), where the projections use Kaleidoscope matrices for all architectures \[Dao et al., 2020]. In all experiments, the linear projections are dense, but for all the theory results in App H, the linear projections use the restricted poly-log Kaleidoscope matrices. Doing so is what gave the claimed O(Nd) parameter count and runtime on Page 6.
>
> We did not specify this clearly in the main paper and update this in our revision (page 6). We also included an explicit proposition in the appendix (Proposition H.6) that argues the claim on a single BaseConv layer. We also have added explicit statements in the proofs to show that using the near linear parameters Kaleidoscope matrices are sufficient for our proof.
>
> _Clarification on the meaning of N._ We had used N to represent both the sequence length and the number of (key, value, query) tuples. We have updated the number of tuples to read N/3 in the revised copy. We apologize for the confusing notation and thank you for pointing this out.
>
> _Shape of inputs in Proposition 4.3._ As noted by the review, we have corrected proposition 4.3 to read $u \in \\{ 0, 1 \\}^{N \times 3c}$_._
>
> Beyond the comments in the review, we thoroughly checked for consistency with respect to notation and further clarified the assumptions underlying our theoretical results.
>
> **Clarifying Definition 3.1 and Descriptions of MQAR**
>
> We have updated the manuscript in the following ways to address this feedback:
>
> 1. We streamlined definition 3.1 and added a simple example per the reviewer suggestions. We also include a simple example of how AR is formulated in prior work for clear comparison.
> 2. We add Appendix D containing several examples of MQAR that we sourced from real language modeling data.
> 3. We add Appendix E (Algorithm 1) detailing exactly how MQAR synthetic data gets constructed.
>
> We hope these collectively clarify the MQAR contribution.
>
> **Explaining the procedure for measuring the AR gap**
>
> In response to the review, we add Appendix C.1. to justify our method for computing the amount of quality gap between the convolution and attention models that is ascribed to associative recall capability (e.g., in Table 1):
>
> 1. Given an input sequence, we identify recurring bigrams (i.e. bigrams that have already appeared in the sequence at a prior position). Since bigrams that appear frequently during training may be memorized by the model, rather than requiring the model to perform recall at inference-time, we only measure AR log-probabilities with respect to bigrams that are seen fewer than a threshold number of times during training. The threshold used in all the experiments in our submission is $1,250$ training occurrences in the 10B tokens of pretraining data.
> 2. We measure the log-probability assigned to the true bigram completion. This bigram completion is referred to as an AR Hit in our work. This protocol assumes that the model can produce the completion by recalling the prior occurrence of the bigram in the sequence.
> 3. For the model being evaluated $m$, and the attention model $M$, we measure the \% of the quality gap between $m$ and $M$ ascribed to associative recall capability as follows.
> 4. Let the average log-probability for all AR Hits across validation sequences be $l\_{H}^m$ and $l\_{H}^M$ for $m$ and $M$ respectively. Let the average log-probabilities of *all tokens* in the validation sequences be $l^m$ and $l^M$ respectively. Let $p\_{H}$ be the proportion of AR Hit tokens in the validation data. As the final gap ascribed to AR, we report:
>
> $$\min (\frac{(l\_{H}^m - l\_{H}^M) p\_{H}}{l^m - l^M}, 1.0)$$
>
> Shown above, if $m$ is better than attention ($M$) overall and $M$ is better than $m$ at AR, we ascribe 100\% of the gap to AR.

---

> ### Author Response · Authors · 2023-11-18
> **Response to Reviewer xdZW (continued.)**
>
> **We briefly discuss two important decisions in this protocol**.
>
> First, we only measure _explicit bigrams_, i.e. bigrams are identified based on token ids in the sequence. However, intuitively, models may also perform associative recall between related _concepts_. For instance, language may contain bigrams in which one word is swapped by a synonym. We note that our work does not measure recall in higher dimensional recall in favor of a more transparent procedure.
>
> Next, we measure the gap based on _log-probabilities_ rather than perplexity. This is simply because we want to make our metrics independent of the number of tokens in each of the slices of the validation set. Approximately 6.4\% of validation tokens are AR Hits with the threshold set to consider bigrams seen less than $1,250 \times$ during training.
>
>
> **Additional Comments**
>
> Overall we have made clarifications to the submission, which we hope makes the work more understandable.
>
> We include Appendix C and E, which provide several additional details on our experimental protocols for synthetic and downstream training across models. We also include a zip file of our code for reproducing our results with all models. Upon publication, we will also release all the pretrained checkpoints analyzed in our paper. We hope these materials build confidence in the reproducibility and thoroughness of our work.
>
> **References**
>
> \[1] Dao et al., Kaleidoscope: An Efficient, Learnable Representation For All Structured Linear Maps, 2020.

---

> > ### Author Response · Authors · 2023-11-21
> > **Following Up on Responses to Reviewer xdZW**
> >
> > Dear Reviewer xdZW,
> >
> > Thank you again for reviewing our work! As the discussion period ends shortly, we want to check if you have any further questions or found our responses helpful?
> >
> > Please let us know and thank you for your time!

---

### Official Review · Reviewer_9Su6 · 2023-11-07

**Soundness:** 3 good
**Presentation:** 3 good
**Contribution:** 3 good
**Rating:** 8
**Confidence:** 4

**Summary:**

The authors study the performance gap between Gated Convolution Models (GCM) and Transformers on language modeling. They show that the lacking capability of GCM on the Associative Recall (AR) task can explain most of the perplexity gap in the real world scenario. They further demonstrate two architecture modifications to GCM, i.e., adding extra attention layers or selectively looking-up for associative tokens, can bridge the performance gap. A theoretical bound of the number of parameters required for GCM to solve the Multiple Query AR task is also derived.

**Strengths:**

The paper provides a comprehensive study of the associative recall capability of neural language models with different neural architectures, and examines its impact on the next token prediction performance of the models on real world data.

The authors empirically demonstrate that boosting the associative recall capability of GCMs can mostly bridge its performance gap with the attention-based model under the scale of 360M parameters.

The authors derive a theoretical scaling bound for data-independent GCMs to solve AR, and validate it with synthetic data.

**Weaknesses:**

The novelty of the paper is limited. The lack of AR ability of State Space Models (which is a special kind of long convolution model with embedded recurrency) has been analyzed in the H3 paper [1] through synthetic data. The proposed architectural modification of hybridization is a simple replication of the Hybrid-H3.

The scale of the experiments is limited. The authors only empirically examine their hypothesis for models with the size up to 360M number of parameters. It is not clear whether their claims still holds empirically given the shrinking trend of the performance gap that can already be observed under the current setting.

The paper does not provide important technical details for reproducibility. The implementation details of the proposed modification of selectively look-up is missing.

The research problem that the authors are trying to solve has been alleviated with existing [1,2] or emerging solutions [3,4]. The authors do not examine the empirical AR ability of data-dependant convolutions by claiming technical difficulties, but there does exist data-dependant SSMs, such as Liquid-S4 [2], that support causal language modeling. Not to mention the latest GCM, Monarch Mixer [3], that also supports causality. On hybridization, a previous work [4] on dynamic input subset selection for attention modules has also been proposed for efficiently combining SSMs with attention. The authors should consider comparing the proposed architectural modifications with these works to avoid being outdated upon publication.

---
[1] Hungry Hungry Hippos: Towards Language Modeling with State Space Models (ICLR 2023)

[2] Liquid Structural State-Space Models (ICLR 2023)

[3] Monarch Mixer: A Simple Sub-Quadratic GEMM-Based Architecture (NeurIPS 2023)

[4] Sparse Modular Activation for Efficient Sequence Modeling (NeurIPS 2023)

**Questions:**

According to Table 1, it seems that as the number parameters increase the performance gap between GCMs and Transformers is shrinking. Can you explain this phenomenon? Is it possible that GCMs may outperform Transformers on a larger scale such as with 700M or 1B parameter counts?

Can you provide mathematical formulas for calculating the perplexity scores of AR hit tokens mentioned in Table 1 and Table2?

How is the selectively look-up mechanism exactly implemented to produce the numbers in Table 2? Is the attention based look-up trainable or not trainable? How is it added to the GCMs layer? Can you provide a series of formulas to describe a GCM layer that is equipped with the proposed selectively look-up mechanism?

---

> ### Author Response · Authors · 2023-11-18
> **Response to Reviewer 9Su6**
>
> Thank you for your thoughtful review! The experiments you suggested and questions you raised helped us greatly improve the work! We appreciated that you found our work comprehensive across theory, synthetic experiments, and downstream experiments. Below, we address your questions and highlight new experiments on an additional set of 6 baseline architectures (H3, Liquid S4, M2, Sliding window attention as in Mistral, Block attention, and RetNet) and a new analysis of associative recall as model size increases (up to the 7B parameter scale).
>
> **Novelty and differences from prior work (e.g H3)**
>
> Thank you for your thoughtful comments about H3. You are correct that the H3 work studies synthetic associative recall tasks and previously proposes the use of hybrid models. However, we make fundamentally different claims. They argue that gated-convolutions (like H3) can solve AR. In contrast, our claim is that gated-convolutions **cannot** solve AR without scaling the hidden dimension with N. To underscore this point, we have added evaluations of H3 in the main paper (Table 1) and show that the trends we documented for other gated-convolutions also hold for H3 (Figure 2). Further, we find that the synthetic task studied in H3 and Hyena don’t reflect recall in real language, so we develop and release a new synthetic better aligned with actual data.
>
> We use this synthetic to explain why adding attention helps with recall. The H3 paper does not provide this explanation. Finally, in downstream hybrid experiments, we go beyond the application of full quadratic attention layers to show that sparse-attention, localized to potential AR tokens, is sufficient to close the gap to the Transformer baseline. We hope this clarifies our differences and contributions.
>
> Please also see our summary of contributions in the main response to all reviewers.
>
> **Experiments with increased model sizes.**
>
> Thank you for raising this question! Please find our new experimental results in the main response to all reviewers.
>
> **New Related Works Section**
>
> Beyond the three architectures discussed above, we include an extended related works section in Appendix A. In response to your feedback that recent or concurrent architectures could alleviate the problem, the section discusses relevant architectures.
>
> **Details for Reproducibility.**
>
> In response to your feedback on the reproducibility of our work, we provide several additional details on our experimental protocols for synthetic (details in Appendix E.2) and downstream training (details in Appendix C) across models. We also include a zip file of our code for reproducing our results with all models. Upon publication, we will also release all the pretrained checkpoints analyzed in our paper. We hope these materials build confidence in the reproducibility of our work.

---

> > ### Author Response · Authors · 2023-11-18
> > **Response to Reviewer 9Su6 (continued.)**
> >
> > **Comparisons to additional architectures shows the AR gap vs. attention remains large.**
> >
> > To address your questions, we evaluate H3 \[Dao et al.], Liquid state space models \[Hasani et al.], M2 \[Fu et al.], and SeqBoat \[Ren et al.] on the Pile language modeling corpus. We also include a new related works section to contextualize our contributions.
> >
> > _Hungry-hungry hippos (H3)._ We evaluate H3 across parameter scales and find that the trends match those of the Hyena architecture – MQAR explains between 63.2% and 88.4% of the gap. We evaluate purely using the H3 layer, not the hybrid architecture, to focus on the behavior of the gated-convolution components. We further evaluate H3 on the MQAR synthetic task that is used throughout our submission and observe empirically that the trend closely follows Hyena and RWKV. We have updated Table 1 and Figure 2 to reflect the new experiments.
> >
> > We also note that the H3 architecture is closely related to the Hyena architecture. They differ only in the parameterization of the short and long convolutions. Each Hyena/H3 layer contains a short convolution followed by a long convolution that is sandwiched between gating. Given these similarities, we originally included just one of these two architectures, but agree with the suggestion that including the explicit comparison is important for readers.
> >
> > Below, recall that we measure the % Explained using the log-perplexities (justified in Appendix C.1). We report perplexities and log-perplexities in parentheses.
> >
> > |                |              |             |              |             |
> > | -------------- | ------------ | ----------- | ------------ | ----------- |
> > | Model          | Overall PPL  | AR PPL      | Non-AR PPL   | % Explained |
> > | Attention 72M  | 12.99 (2.56) | 2.41 (0.88) | 14.76 (2.69) | -           |
> > | H3 72M         | 15.8 (2.76)  | 13.9 (2.63) | 15.9 (2.77)  | 63.2%       |
> > | Attention 125M | 11.01 (2.40) | 2.16 (0.77) | 12.45 (2.52) | -           |
> > | H3 168M        | 12.1 (2.49)  | 6.75 (1.91) | 12.60 (2.53) | 88.4%       |
> > | Attention 360M | 9.44 (2.25)  | 1.98 (0.69) | 10.62 (2.36) | -           |
> > | H3 357M        | 10.38 (2.34) | 4.81 (1.57) | 11.00 (2.40) | 65.8%       |
> >
> > _Monarch-Mixer (M2, Fu et al.)._ Next, we note that M2 is an implementation of the Hyena architecture that is optimized for higher hardware utilization. M2 uses the exact same sequence mixer as Hyena, but implements the convolutions with monarch matrix multiplications (instead of \`torch.fft\`). Therefore, the analysis in our main paper pertains to M2 as well. Since Hyena lacks sufficient data-dependence for MQAR, so do current M2 architectures.
> >
> > We note that the M2 authors have released code for training BERT-style language models so we train BERT with M2 vs. BERT with attention from scratch at the 130M parameter scale at a 1024 sequence length. As expected, we find that there is an 4.6% overall quality gap in the masked language modeling accuracy and 58.5% of this can be explained by the gap on repeated bigram tokens.
> >
> > We use the exact same training configurations provided in the M2 codebase. However, we note that the M2 paper compares to Devlin et al. BERT numbers. Devlin et al. trains on  Wikipedia and Books pretraining data with a 15% masking probability, while the numbers reported in the M2 paper are with C4 pretraining data and 30% masking. For fair comparison, we train both BERT and M2-BERT models using consistent data and masking probabilities (C4, 30%) for this analysis. Overall, M2 does not address the MQAR problem as it uses Hyena under the hood, supported further by our experiments.
> >
> > _Liquid state space models (Hasani et al.)._ We evaluate Liquid SSM upon suggestion and include this in Table 3 of the revision. Liquid SSM introduces a new convolution filter that is composed of the original input-independent SSM matrices as well as correlation terms between the original input tokens. The correlation terms are included to make the filter data-dependent. We evaluate Liquid SSM on the Pile after training for 5B tokens and find that quality lags attention by 4.4 ppl.
> >
> > Below, recall that we measure the % Explained using the log-perplexities (justified in Appendix C.1). We report perplexities and log-perplexities in parentheses.
> >
> > |                       |              |              |              |             |
> > | --------------------- | ------------ | ------------ | ------------ | ----------- |
> > | Model after 5B Tokens | Overall PPL  | AR PPL       | Non-AR PPL   | % Explained |
> > | Attention (125M)      | 12.37 (2.52) | 2.32 (0.84)  | 14.04 (2.64) | -           |
> > | Liquid SSM  (145M)    | 16.80 (2.82) | 22.75 (3.12) | 16.42 (2.80) | 52.6%       |

---

> ### Author Response · Authors · 2023-11-18
> **Response to Reviewer 9Su6 (continued.)**
>
> _SeqBoat (Ren et al.)._ We thank the reviewer for this pointer! The simple learned selection function we evaluate in Section 5 is similar to SeqBoat. One major difference is that we explicitly limit the number of activated tokens to ensure sub-quadratic scaling, while SeqBoat allows all tokens to activate. This occurs in some of experiments in the SeqBoat (See Figure 3 of their paper). While SeqBoat use the sparse attention at every layer, we simply replace three of the BaseConv layers with the sparse attention mechanism. Also, unlike SeqBoat, we use an auxiliary loss that guides the model to sparsely select attention positions. Based on the reviewer’s pointer, we evaluate SeqBoat’s approach of using no auxiliary loss and find that it increases perplexity by 0.5.
>
> |     |                     |                                  |
> | --- | ------------------- | -------------------------------- |
> |     | With Auxiliary Loss | Without Auxiliary Loss (SeqBoat) |
> | PPL | 11.134              | 11.634                           |
>
>
> We include a discussion of our differences from this work in Appendix A.3 (Related Works). We also note that our objective in Section 5 is to analyze simple, common-sense approaches for closing the gap sub-quadratically. We do not claim that this simple method is novel or that we are the first to propose it, but rather our contribution lies in the explanation of why these mechanisms work.
>
> Finally, please see synthetic MQAR and downstream Pile experiments on additional architectures (RetNet, Sliding window attention like Mistral, and Blocked attention) in Appendix F.

---

> > ### Comment · Reviewer_9Su6 · 2023-11-20
> >
> > Thanks for the detailed clarifications and additional experiments. I have raised my score to 8, given that the authors' response has resolved most of my concerns.

---

### Author Response · Authors · 2023-11-18
**Common Response to Reviewers**

We thank all the reviewers for their time and effort reviewing our work and for their constructive comments, which have made our paper stronger. Reviewers consistently appreciated our insightful and comprehensive analysis \[9Su6, xdZW, 5EW4] and the combination of both theoretical and empirical contributions towards explaining the gaps between architecture classes \[9Su6, 5EW4].

****

In this common response, we address questions from the reviews providing: a short discussion of our contributions, a summary of the changes we made in the revision, and details on important new experiments relevant to all reviewers.  Please find our comments for individual reviewers in their respective threads.


### Summary of Contributions

**Our key contribution is a novel analysis of gated-convolution architectures.** This analysis is timely because interest in gated-convolution models is taking off rapidly – they (1) process sequences asymptotically faster than attention and (2) recent works suggest that they can match attention in language modeling quality \[H3, Hyena, RWKV, GSS, AFT]. In our analysis we:


1. **Compare architectures on real world data and reveal quality gaps that are not covered in the prior work.** We pretrain 15 models spanning 3 parameter scales (70m - 350M) and 5 popular architectures (H3, Hyena, S4, RWKV, attention) (Table 1) on the Pile and observe overall perplexity gaps to attention across sizes, which is surprising since prior work suggest the architectures match in quality.


2. **Provide novel insight into a historically important language modeling sub-task called associative recall (AR).** _As 9Su6 notes, prior work suggests gated-convolutions solve this task_ \[H3, Hyena]_._ In stark contrast, we find gated convolutions are far worse at AR than attention: we find that 82% of average gap to attention can be attributed to poor quality on AR.  We surprisingly find that a 70M parameter attention model outperforms a 1.4 billion parameter Hyena model 20x its size on AR.  We also note we are the first work to study the AR capabilities of gated-convolutions on _real data._


3. **Develop novel analytical tools that improve upon existing formalisms.** A major contribution of our work is _Multi-Query Associative Recall_ (MQAR), a new formalization of AR. Prior work tests recall at a _fixed position in the sequence_ - a model can solve this task using circuits that do not generalize to _all positions in the sequence._ Meanwhile, MQAR uses multiple AR queries per example at varying positions within a sequence (examples in Appendix D). We show that a model’s quality on our MQAR synthetic task correlates with AR quality in real world data (Sec. 4.3, Sec. 5, Appendix F), unlike the prior recall diagnostics.


4. **Produce actionable insights for future architecture design.** We empirically and theoretically show: while attention can solve associative recall at all model dimensions, gated convolutions use model dimension that scales with sequence length. This scaling is undesirable – we theoretically and empirically pinpoint how to efficiently close the MQAR gap by using input-dependent sequence aggregation (whereas standard convolution filters are input-independent). Informed by our analysis, we study minimal architectures that can close the gap: we add three layers of input-dependent sparse attention to a simple gated-convolution architecture and show it can close 97% of the gap to attention on real language modeling.

9Su6 asks how our work relates to H3. Our work builds on prior work like H3, which show the power of full attention hybrids. We go one step further, providing an explanation of _why_ the attention mechanism is necessary and showing how sparse sub-quadratic attention suffices to close most of the gap. **This analysis is the first that can \*_explain_\* the gap between gated-convolutions and attention on associative recall.**&#x20;

---

> ### Author Response · Authors · 2023-11-18
> **Summary of Changes**
>
> ### Summary of Changes
>
> **Revision Summary**. In our revision, we made extensive reviewer-suggested changes that strengthen our work and highlight our contributions. These changes include:
>
>
> 1. **Experiments on larger scale models \[9Su6, 5EW4] (Section 3, Appendix G).** In response to comments on the model sizes used in our experiments,  we extend our empirical analysis to 1.4 and 7 billion parameter language models. We find that the AR gap between attention and gated convolutions persists at these scales.
> 2. **Experiments with new architectures \[9Su6, 5EW4] (Section 3, Appendix F).** We include five new architectures in our analyses including H3, Liquid S4, RetNet, Sliding Window Attention, and Blocked Attention.
> 3. **Improved presentation and reproducibility \[9Su6, 5EW4].** Based on feedback from 9Su6 and 5EW4 on missing technical details, we’ve restructured Sections 2, 3 and 5 for improved clarity. We additionally include code for reproducibility and will publicly release code and pre-trained checkpoints from this work upon publication.
> 4. **Clarifications on novelty \[9Su6].** In response to questions on the difference between our work and H3, we clarify in Section 3.2 how our work (1) makes a fundamentally different claim than H3 and (2) uses a novel formalization of associative recall, MQAR, which remains unsolved by gated-convolutions with constant dimensionality. We also add an **extended related works section** that (1) details the prior use of associative recall in the literature and (2) details the efficient architectures that have been proposed.
> 5. **Updates to the theory \[xdZW].** In response to concerns, typos, and clarity issues in our theoretical analysis raised by xdZW, we’ve made updates to Proposition 4.3 and the corresponding proof H.27, though the overarching claim and proof structure are unchanged. We also provide additional discussion in all proof sections to improve clarity for readers. Finally, we also made a thorough pass through our proofs checking for correctness and typos. We also note that we provide empirical validation of our theoretical statements, using simple synthetics, supporting the correctness.

---

> ### Author Response · Authors · 2023-11-18
> **Common Discussion of New Experiments**
>
> ### Discussion of new experiments
>
> To address questions from 9Su6 and 5EW4 about the size of the MQAR gap as the attention and gated convolution model sizes increase, we include three new sets of experiments:
>
> 1. **Pretraining scaled up.** We pretrain and evaluate new attention and Hyena (gated convolution) models at the 1.4B parameter scale. Even at 1.4B parameters, Hyena still performs worse on AR than attention models 20x smaller (70M parameters).
> 2. **Analyzing off-the-shelf large language models.** We then scale up to 7B pre-trained RWKV-Raven (gated convolution) model downloaded from HuggingFace and Llama 2 (attention) model downloaded from Meta. We note that RWKV is the only gated-convolution architecture that has been scaled up and widely disseminated thus far. Both RWKV 7B and Llama 2 are popular amongst users.
> 3. **Analysis of the AR gap with scale.** Finally, in further analysis of our pretrained checkpoints, we find that for bigrams seen $<1,250\times$ during training the gap between 350M parameter attention and Hyena / RWKV on AR is 1.85 / 1.84 perplexity points respectively (reported in Table 1). However, if we consider bigrams seen $1 \times$, the gaps to attention are 12.0 / 13.2 perplexity points respectively (Appendix G). This suggests that the larger models may get better at memorizing bigrams, but improve less quickly on in-context learning. Overall we find that MQAR remains problematic even as we scale up the gated convolution and attention architectures. We detail these new experiments in Appendix G.
>
> ### **_Pretraining scaled up._**
> We train Hyena and attention models on the Pile at 1.4Bn parameters and observe the following. We observe that the 20x larger 1.4Bn Hyena model still underperforms the 70M parameter attention model on the AR slice. Overall, the % Explained by AR is less than at the smaller scales, but still sizeable. We note that it is also possible for the model to perform AR between higher dimensional concepts (e.g. fuzzy bigram matches) and these are not captured by the bigram heuristic used in our work to measure AR. The heuristic may be less effective for larger models. We include the new results and discussion in Section 3 and Appendix G. Below, recall that we measure the % Explained using the log-perplexities (justified in Appendix C.1). We report perplexities and log-perplexities in parentheses.
> |                |             |             |              |             |
> | -------------- | ----------- | ----------- | ------------ | ----------- |
> | Model          | Overall PPL | AR PPL      | Non-AR PPL   | % Explained |
> | Attention 1.4B | 8.19 (2.10) | 1.91 (0.65) | 9.86 (2.29)  | -           |
> | Hyena 1.4B     | 9.65 (2.26) | 3.43 (1.23) | 11.01 (2.40) | 40.3%       |

---

> ### Author Response · Authors · 2023-11-18
>
> ### **_Analyzing off-the-shelf large language models._**
> We next evaluate the [RWKV](https://huggingface.co/docs/transformers/v4.35.0/en/model_doc/rwkv#transformers.RwkvForCausalLM)-Raven (gated-convolution) and Llama 2 (attention) pre-trained models at the 7B parameter scale. These are both popular models that took a significant amount of effort to train, towards maximizing quality. **We find that there is a large gap between RWKV and attention at the 7B scale on AR tokens and that it _increases as the model needs to conduct more recalls per sequence (P below)_.** On the non-AR tokens, quality is comparable.
>
> |                                              |                            |                               |       |
> | -------------------------------------------- | -------------------------- | ----------------------------- | ----- |
> | P (number of recalls  required per sequence) | RWKV on Associative Recall | Llama 2 on Associative Recall | Gap   |
> | 4                                            | 9.00                       | 14.16                         | -5.16 |
> | 8                                            | 13.16                      | 8.32                          | 4.84  |
> | 16                                           | 22.52                      | 3.73                          | 13.09 |
> | 32                                           | 52.59                      | 3.07                          | 49.52 |
>
> |                                             |                       |                          |      |
> | ------------------------------------------- | --------------------- | ------------------------ | ---- |
> | P (number of recalls required per sequence) | RWKV on Non-AR Tokens | Llama 2 on Non-AR Tokens | Gap  |
> | 4                                           | 1.22                  | 1.21                     | 0.01 |
> | 8                                           | 1.43                  | 1.39                     | 0.04 |
> | 16                                          | 1.95                  | 1.83                     | 0.08 |
> | 32                                          | 3.45                  | 3.05                     | 0.40 |
>
> We summarize the experimental protocol for the 7B scale.
> 1. **Justifying the experimental protocol.** Since frequent bigrams may just be memorized by the model and not require in-context recall, our work measures AR quality on infrequent bigrams in validation sequences (Figure 1). We do not have access to the custom training data mixtures used in training RWKV/Llama 2 to measure bigram frequencies, so we use a synthetic test to fairly measure the AR capabilities.
> 2. **Evaluation dataset.** We construct a synthetic dataset where each sequence contains P token pairs, or “bigrams”. Each bigram, containing a “key” token and a “value” token, appears twice in the sequence. On the second occurrence of a “key” token, the model should look back to the prior occurrence to output the corresponding “value” token. We measure the _AR perplexity_ of the model based on its ability to predict the correct “value” token as the next word for these repeated keys. The synthetic construction procedure is shown in Algorithm 1 (same procedure as Figure 2) and the sequences are constructed using the models’ vocabulary tokens.
> 3. **Evaluation details.** We evaluate (inference only, no training) on sequence lengths of 1024 tokens. We measure the AR perplexity when the sequences contain $P \in \\{4, 8, 16, 32\\}$ key-value pairs, using 1000 samples per P value. The tokens that do not contain a key or value are simply filled with a _fixed_ token id (so this token is repeated frequently in the sequence). We plot perplexity for AR and non-AR tokens (fixed token) vs. P. **We find RWKV quality degrades with P on the AR slice (blue line), while all other lines remain flat. MQAR remains problematic at scale.** We release code to reproduce this experiment.
>
> ### **Analysis of the AR gap with scale**
> In the main paper, we measure the AR perplexity on downstream data based on bigrams that are seen $< 1,250\times$ during pretraining. However, we hypothesize that larger models may memorize a larger number of bigram pairs seen in the training dataset, but do not rapidly gain the capability to perform associative recall as well as attention _in-context_.  In-context learning is defined as learning from examples provided in context [Xie et al., 2021]. Concretely, the gap between the Hyena / RWKV and attention models at the 350m scale is 1.85  / 1.84 perplexity points when we focus on bigrams seen $< 1,250\times$ during pretraining (Table 1). If we instead focus on bigrams seen $1\times$ during pretraining, the gap to attention quality is 12.0 / 13.19 perplexity points respectively. The gated convolutions appear to struggle in the regime of rare bigrams that require the model to use the context.
>
> [1] Xie et al. An Explanation of In-context Learning as Implicit Bayesian Inference, 2021.

---

### Meta-Review · Area_Chair_H3zT · 2023-12-06

**Metareview:**

The paper presents a comprehensive study on the performance gap between Gated Convolution Models (GCM) and Transformers in language modeling tasks. Interestingly, the authors identify that the lack of Associative Recall (AR) capability (output the next token using the prior context) in GCMs accounts for a significant portion of the perplexity gap. They propose architectural modifications to GCMs and provide theoretical bounds for solving AR, which are validated with synthetic data. The authors also present strong baseline models that outperform Transformers at certain parameter scales.

The reviewers appreciated the comprehensive study and the identification of the AR capability as a significant factor in the performance gap. The theoretical bounds and the proposed architectural modifications were also well-received. However, concerns were raised about the novelty of the work, the scale of the experiments, and the lack of important technical details for reproducibility.

I believe the authors' responses clarified most of these concerns. The paper is well-written and the experiments are well-executed, making it a valuable to the conference.

**Justification For Why Not Higher Score:**

The scale of the experiments conducted in the paper is limited to models with up to 360M parameters. It is unclear whether the claims made by the authors would hold true for larger models. The author add new scaled-up results in the responses, but more detailed experiments are still needed.

**Justification For Why Not Lower Score:**

See meta review

---

### Decision · Program_Chairs · 2024-01-16

Accept (poster)